# Generalization in the Face of Adaptivity:
# A Bayesian Perspective

**Moshe Shenfeld**
Department of Computer Science
The Hebrew University of Jerusalem
moshe.shenfeld@mail.huji.ac.il

**Katrina Ligett**
Department of Computer Science
Federmann Center for the Study of Rationality
The Hebrew University of Jerusalem
katrina@cs.huji.ac.il

## Abstract

Repeated use of a data sample via adaptively chosen queries can rapidly lead to overfitting, wherein the empirical evaluation of queries on the sample significantly deviates from their mean with respect to the underlying data distribution. It turns out that simple noise addition algorithms suffice to prevent this issue, and differential privacy-based analysis of these algorithms shows that they can handle an asymptotically optimal number of queries. However, differential privacy's worst-case nature entails scaling such noise to the range of the queries even for highly-concentrated queries, or introducing more complex algorithms.

In this paper, we prove that straightforward noise-addition algorithms already provide variance-dependent guarantees that also extend to unbounded queries. This improvement stems from a novel characterization that illuminates the core problem of adaptive data analysis. We show that the harm of adaptivity results from the covariance between the new query and a Bayes factor-based measure of how much information about the data sample was encoded in the responses given to past queries. We then leverage this characterization to introduce a new data-dependent stability notion that can bound this covariance.

## 1 Introduction

Recent years have seen growing recognition of the role of *adaptivity* in causing overfitting and thereby reducing the accuracy of the conclusions drawn from data. Intuitively, allowing a data analyst to adaptively choose the queries that she issues potentially leads to misleading conclusions, because the results of prior queries might encode information that is specific to the available data sample and not relevant to the underlying distribution from which it was drawn. As a result, future queries might then be chosen to leverage these sample-specific properties, giving answers on the sample that differ wildly from the values of those queries on the data distribution. Such adaptivity has been blamed, in part, for the current reproducibility crisis in the data-driven sciences [Ioannidis, 2005, Gelman and Loken, 2014].

A series of works catalyzed by Dwork et al. [2015] recently established a formal framework for understanding and analyzing adaptivity in data analysis, and introduced a general toolkit for provably preventing the harms of choosing queries adaptively—that is, as a function of the results of previous queries. This line of work has established that enforcing that computations obey a constraint of *differential privacy* [Dwork et al., 2006]—which one can interpret as a robustness or stability requirement on computations—provably limits the extent to which adaptivity can cause overfitting. Practically, for simple numerical computations, these results translate into adding a level of noise to the query results that would be sufficient to ensure *worst-case* stability of the *worst-case* query on the *worst-case* dataset. In particular, when analyzed using differential privacy, simple noise-addition

mechanisms must add noise that scales with the *worst-case* change in query value that could be induced by changing a single input data element, in order to protect against adaptivity.

However, this can be overkill: for the purposes of statistical validity, we do not care about the worst case but the typical case. In the present work, we prove the following new and better generalization guarantees for simple Gaussian noise-addition algorithms:

**Theorem 1.1** (Informal versions of main theorems). *With probability $> 1 - \delta$, the error of the responses produced by a mechanism which only adds Gaussian noise to the empirical values of the queries it receives is bounded by $\epsilon$, even after responding to $k$ adaptively chosen queries, if*

- *the range of the queries is bounded by $\Delta$, their variance is bounded by $\sigma^2$, and the size of the dataset $n = \Omega\left(\max\left\{\frac{\Delta}{\epsilon}, \frac{\sigma^2}{\epsilon^2}\right\}\sqrt{k} \cdot \ln\left(\frac{k\sigma}{\delta\epsilon}\right)\right)$ (Theorem 5.1), or*

- *the queries are $\sigma^2$-sub-Gaussian and the size of the dataset $n = \Omega\left(\frac{\sigma^2\sqrt{k}}{\epsilon^2}\ln\left(\frac{k\sigma}{\delta\epsilon}\right)\right)$ (Theorem 5.3).*

In contrast, analyzing the same mechanism using differential privacy yields a sample size requirement that scales with $n = \Omega\left(\frac{\Delta^2\sqrt{k}}{\epsilon^2}\ln\left(\frac{k\sigma}{\delta\epsilon}\right)\right)$ in the first setting, and provides no guarantee in the latter.

We prove these theorems via a new notion, *pairwise concentration (PC)* (Definition 4.2), which captures the extent to which replacing one dataset by another would be "noticeable," given a particular query-response sequence. This is thus a function of particular differing datasets (instead of worst-case over elements), and it also depends on the actual issued queries. We then build a composition toolkit (Theorem 4.4) that allows us to track PC losses over multiple computations. The PC notion allows for more careful analysis of the information encoded by the query-response sequence than differential privacy does.

In order to leverage this more careful analysis of the information encoded in query-response sequences, we rely on a simple new characterization (Lemma 3.5) that shows explicitly that the harms of adaptivity come from the covariance between the behavior of the future queries and a Bayes factor-based measure of how much information about the data sample was encoded in the responses to the issued queries. Our characterization gives insight into how differential privacy protects against adaptivity, and then allows us to step away from this worst-case approach. What differential privacy accomplishes is that it bounds a worst-case version of the Bayes factor; however, as this new characterization makes clear, it is sufficient to bound the "typical" Bayes factor in order to avoid overfitting.

We measure the harm that past adaptivity causes to a future query by considering the query as evaluated on a posterior data distribution and comparing this with its value on a prior. The prior is the true data distribution, and the posterior is induced by observing the responses to past queries and updating the prior. If the new query behaves similarly on the prior distribution as it does on this posterior (a guarantee we call *Bayes stability*; Definition 3.3), adaptivity has not led us too far astray.[1] If furthermore, the the response given by the mechanism is close to the query result on the posterior, then by a triangle inequality argument, that mechanism is distribution accurate. This type of triangle inequality first appeared as an analysis technique in Jung et al. [2020].

The dependence of our PC notion on the actual adaptively chosen queries places it in the so-called *fully-adaptive* setting [Rogers et al., 2016, Whitehouse et al., 2023], which requires a fairly subtle analysis involving a set of tools and concepts that may be of independent interest. In particular, we establish a series of "dissimilarity" notions in Appendix B, which generalize the notion of divergence, replacing the scalar bound with a function. Our main stability notion (Definition 4.2) can be viewed as an instance-tailored variant of *zero-concentrated differential privacy [Bun and Steinke, 2016]*, and we also make use of a similar extension of the classical max-divergence-based differential privacy definition (B.8).

**Related work** *Differential privacy* [Dwork et al., 2006] is a privacy notion based on a bound on the max divergence between the output distributions induced by any two neighboring input

---

[1]This can be viewed as a generalization of the *Hypothesis Stability* notion of Bousquet and Elisseeff [2002]— which was proven to guarantee on-average generalization [Shalev-Shwartz et al., 2010]—where the hypothesis is a post-processing of the responses to past queries, and the future query is the loss function estimation.

datasets (datasets which differ in one element). One natural way to enforce differential privacy is by directly adding noise to the results of a numeric-valued query, where the noise is calibrated to the *global sensitivity* of the function to be computed—the maximal change in its value between any two neighboring datasets. Dwork et al. [2015] and Bassily et al. [2021] showed that differential privacy is also useful as tool for ensuring generalization in settings where the queries are chosen adaptively.

Differential privacy essentially provides the optimal asymptotic generalization guarantees given adaptive queries [Hardt and Ullman, 2014, Steinke and Ullman, 2015]. However, its optimality is for worst-case adaptive queries, and the guarantees that it offers only beat the naive intervention—of splitting a dataset so that each query gets fresh data—when the input dataset is quite huge [Jung et al., 2020]. A worst-case approach makes sense for privacy, but for statistical guarantees like generalization, we only need statements that hold with high probability with respect to the sampled dataset, and only on the actual queries issued.

One cluster of works that steps away from this worst-case perspective focuses on giving privacy guarantees that are tailored to the dataset at hand [Nissim et al., 2007, Ghosh and Roth, 2011, Ebadi et al., 2015, Wang, 2019]. In Feldman and Zrnic [2021] in particular, the authors elegantly manage to track the individual privacy loss of the elements in the dataset. However, their results do not enjoy a dependence on the standard deviation in place of the range of the queries. Several truncation-based specialized mechanisms have been proposed, both to provide differential privacy guarantees for Gaussian and sub-Gaussian queries even in the case of multivariate distribution with unknown covariance [Karwa and Vadhan, 2018, Ashtiani and Liaw, 2022, Duchi et al., 2023] and, remarkably, design specialized algorithms that achieve adaptive data analysis guarantees that scale like the standard deviation of the queries [Feldman and Steinke, 2017]. Recently, Blanc [2023] proved that randomized rounding followed by sub-sampling provides accuracy guarantees that scale with the queries' variance. But none of these results apply to simple noise addition mechanisms.

Another line of work (e.g., Gehrke et al. [2012], Bassily et al. [2013], Bhaskar et al. [2011]) proposes relaxed privacy definitions that leverage the natural noise introduced by dataset sampling to achieve more average-case notions of privacy. This builds on intuition that average-case privacy can be viewed from a Bayesian perspective, by restricting some distance measure between some prior distribution and some posterior distribution induced by the mechanism's behavior [Dwork et al., 2006, Kasiviswanathan and Smith, 2014]. This perspective was used Shenfeld and Ligett [2019] to propose a stability notion which is both necessary and sufficient for adaptive generalization under several assumptions. Unfortunately, these definitions have at best extremely limited adaptive composition guarantees. Bassily and Freund [2016] connect this Bayesian intuition to statistical validity via *typical stability*, an approach that discards "unlikely" databases that do not obey a differential privacy guarantee, but their results require a sample size that grows linearly with the number of queries even for iid distributions. Triastcyn and Faltings [2020] propose the notion of *Bayesian differential privacy* which leverages the underlying distribution to improve generalization guarantees, but their results still scale with the range in the general case.

An alternative route for avoiding the dependence on worst case queries and datasets was achieved using expectation based stability notions such as *mutual information* and *KL stability* Russo and Zou [2016], Bassily et al. [2021], Steinke and Zakynthinou [2020]. Using these methods Feldman and Steinke [2018] presented a natural noise addition mechanism, which adds noise that scales with the empirical variance when responding to queries with known range and unknown variance. Unfortunately, in the general case, the accuracy guarantees provided by these methods hold only for the expected error rather than with high probability.

A detailed comparison to other lines of work can be found in Appendix F.

## 2 Preliminaries

### 2.1 Setting

We study datasets, each of fixed size $n \in \mathbb{N}$, whose elements are drawn from some domain $\mathcal{X}$. We assume there exists some distribution $D_{\mathcal{X}^n}$ defined over the datasets $s \in \mathcal{X}^n$.[2] We consider a family

---

[2]Throughout the paper, the notation $D(\cdot)$ describing a distribution over a domain, will either represent the probability mass function defined by a discrete probability over a countable sample space or the probability density function (the Radon–Nikodym derivative) in case of a measure over measurable spaces.

of functions (*queries*) $\mathcal{Q}$ of the form $q : \mathcal{X}^n \to \mathcal{R} \subseteq \mathbb{R}$. We refer to the functions' outputs as *responses*.[3]

We denote by $q(D_{\mathcal{X}^n}) \coloneqq \mathbb{E}_{S \sim D_{\mathcal{X}^n}}[q(S)]$ the mean of $q$ with respect to the distribution $D_{\mathcal{X}^n}$, and think of it as the true value of the query $q$, which we wish to estimate. As is the case in many machine learning or data-driven science settings, we do not have direct access to the distribution over datasets, but instead receive a dataset $s$ sampled from $D_{\mathcal{X}^n}$, which can be used to compute $q(s)$—the empirical value of the query.

A *mechanism* $M$ is a (possibly non-deterministic) function $M : \mathcal{X}^n \times \mathcal{Q} \times \Theta \to \mathcal{R}$ which, given a dataset $s \in \mathcal{X}^n$, a query $q \in \mathcal{Q}$, and some auxiliary parameters $\theta \in \Theta$, provides some response $r \in \mathcal{R}$ (we omit $\theta$ when not in use). The most trivial mechanism would simply return $r = q(s)$, but we will consider mechanisms that return a noisy version of $q(s)$. A mechanism induces a distribution over the responses $D_{\mathcal{R}|\mathcal{X}^n}^{q,\theta}(\cdot \,|\, s)$—indicating the probability that each response $r$ will be the output of the mechanism, given $s$, $q$, and $\theta$ as input. Combining this with $D_{\mathcal{X}^n}$ induces the marginal distribution $D_{\mathcal{R}}^{q,\theta}$ over the responses, and the conditional distribution $D_{\mathcal{X}^n|\mathcal{R}}^{q,\theta}(\cdot \,|\, r)$ over sample sets. Considering the uniform distribution over elements in the sample set induces similar distributions over the elements $D_{\mathcal{X}}$ and $D^r \coloneqq D_{\mathcal{X}|\mathcal{R}}^{q,\theta}(\cdot \,|\, r)$. This last distribution, $D^r$, is central to our analysis, and it represents the *posterior distribution* over the sample elements given an observed response—a Bayesian update with respect to a prior of $D_{\mathcal{X}}$. All of the distributions discussed in this subsection are more formally defined in Appendix A.

An *analyst* $A$ is a (possibly non-deterministic) function $A : \mathcal{R}^* \to (\mathcal{Q}, \Theta)$ which, given a sequence of responses, provides a query and parameters to be asked next. An interaction between a mechanism and an analyst— wherein repeatedly (and potentially adaptively) the analyst generates a query, then the mechanism generates a response for the analyst to observe—generates a sequence of $k$ queries and responses for some $k \in \mathbb{N}$, which together we refer to as a *view* $v_k = (r_1, r_2, \ldots, r_k) \in \mathcal{V}_k$ (we omit $k$ when it is clear from context). In the case of a non-deterministic analyst, we add its coin tosses as the first entry in the view, in order to ensure each $(q_i, \theta_i)$ is a deterministic function of $v_{i-1}$, as detailed in Definition A.3; given a view $v \in \mathcal{V}$, we denote by $\bar{q}_v$ the sequence of queries that it induces. Slightly abusing notation, we denote $v = M(s, A)$. The distributions $D_{\mathcal{R}|\mathcal{X}^n}^{q,\theta}$, $D_{\mathcal{R}}^{q,\theta}$, $D_{\mathcal{X}^n|\mathcal{R}}^{q,\theta}$, and $D^r$ naturally extend to versions where a sequence of queries is generated by an analyst: $D_{\mathcal{V}|\mathcal{X}^n}^A$, $D_{\mathcal{V}}^A$, $D_{\mathcal{X}^n|\mathcal{V}}^A$, and $D^v \coloneqq D_{\mathcal{X}|\mathcal{V}}^A(\cdot \,|\, v)$.

Notice that $M$ holds $s$ but has no access to $D$. On the other hand, $A$ might have access to $D$, but her only information regarding $s$ comes from $M$'s responses as represented by $D^v$. This intuitively turns some metric of distance between $D$ and $D^v$ into a measure of potential overfitting, an intuition that we formalize in Definition 3.3 and Lemma 3.5.

## 2.2 Notation

Throughout the paper, calligraphic letters denote domains (e.g., $\mathcal{X}$), lower case letters denote elements of domains (e.g., $x \in \mathcal{X}$), and capital letters denote random variables (e.g., $X \sim D_{\mathcal{X}}$).

We omit most superscripts and subscripts when clear from context (e.g., $D(r \,|\, s) = D_{\mathcal{R}|\mathcal{X}^n}^{q,\theta}(r \,|\, s)$ is the probability to receive $r \in \mathcal{R}$ as a response, conditioned on a particular input dataset $s$, given a query $q$ and parameters $\theta$). Unless specified otherwise, we assume $D_{\mathcal{X}^n}$, $n$, $k$, and $M$ are fixed, and omit them from notations and definitions.

We use $\|\cdot\|$ to denote the Euclidean norm, so $\|\bar{q}_v(x)\| = \sqrt{\sum_{i=1}^k (q_i(x))^2}$ will denote the norm of a concatenated sequence of queries.

## 2.3 Definitions

We introduce terminology to describe the accuracy of responses that a mechanism produces in response to queries.

---

[3]Most of the definitions and claims in this paper can be extended beyond $\mathbb{R}$ and the Euclidean norm. To do so, the divergence in Lemma 3.5 and the $\varphi$ function (Definition 4.2) must be chosen accordingly.

**Definition 2.1** (Accuracy of a mechanism). Given a dataset $s$, an analyst $A$, and a view $v = M(s, A)$, we define three types of output error for the mechanism: *sample error* $\text{err}_S(s, v) \coloneqq \max_{i \in [k]} |r_i - q_i(s)|$, *distribution error* $\text{err}_D(s, v) \coloneqq \max_{i \in [k]} |r_i - q_i(D)|$, and *posterior error* $\text{err}_P(s, v) \coloneqq \max_{i \in [k]} |r_i - q_i(D^v)|$, where $q_i$ is the $i$th query and $r_i$ is the $i$th response in $v$.

These errors can be viewed as random variables with a distribution that is induced by the underlying distribution $D$ and the internal randomness of $M$ and $A$. Given $\epsilon, \delta \geq 0$ we call a mechanism $M$ $(\epsilon, \delta)$-*sample/distribution/posterior accurate* with respect to $A$ if

$$\Pr_{S \sim D^{(n)}, V \sim M(S, A)} [(\text{err}(S, V)) > \epsilon] \leq \delta,$$

for $\text{err} = \text{err}_S / \text{err}_D / \text{err}_P$, respectively.

Notice that if each possible value of $\epsilon$ has a corresponding $\delta$, then $\delta$ is essentially a function of $\epsilon$. Given such a function $\delta : \mathbb{R}^+ \to [0, 1]$, we will say the mechanism is $(\epsilon, \delta(\epsilon))$ accurate, a perspective which will be used in Lemma 3.1.

We start by introducing a particular family of queries known as *linear queries*, which will be used to state the main results in this paper, but it should be noted that many of the claims extend to arbitrary queries as discussed in Section C.2.

**Definition 2.2** (Linear queries). A function $q : \mathcal{X}^n \to \mathbb{R}$ is a *linear query* if it is defined by a function $q_1 : \mathcal{X} \to \mathbb{R}$ such that $q(s) \coloneqq \frac{1}{n} \sum_{i=1}^{n} q_1(s_i)$. For simplicity, we denote $q_1$ as $q$ throughout.

The aforementioned error bounds implicitly assume some known scale of the queries, which is usually chosen to be their range, denoted by $\Delta_q \coloneqq \sup_{x, y \in \mathcal{X}} |q(x) - q(y)|$ (we refer to such a query as $\Delta$-bounded). This poses an issue for concentrated random variables, where the "typical" range can be arbitrarily smaller than the range, which might even be infinite. A natural alternative source of scale we consider in this paper is the query's variance $\sigma_q^2 \coloneqq \mathbb{E}_{X \sim D_\mathcal{X}} \left[ (q(X) - q(D_\mathcal{X}))^2 \right]$, or its variance proxy in the case of a sub-Gaussian query (Definition 5.2). The corresponding definitions for arbitrary queries are presented in Section C.2.

Our main tool for ensuring generalization is the Gaussian mechanism, which simply adds Gaussian noise to the empirical value of a query.

**Definition 2.3** (Gaussian mechanism). Given $\eta > 0$ and a query $q$, the *Gaussian mechanism* with noise parameter $\eta$ returns its empirical mean $q(s)$ after adding a random value, sampled from an unbiased Gaussian distribution with variance $\eta^2$. Formally, $M(s, q) \sim \mathcal{N}(q(s), \eta^2)$.[4]

# 3 Analyzing adaptivity-driven overfitting

In this section, we give a clean, new characterization of the harms of adaptivity. Our goal is to bound the distribution error of a mechanism that responds to queries generated by an adaptive analyst. This bound will be achieved via a triangle inequality, by bounding both the posterior accuracy and the Bayes stability (Definition 3.3). Missing proofs from this section appear in Appendix C.

The simpler part of the argument is posterior accuracy, which we prove can be inherited directly from the sample accuracy of a mechanism. This lemma resembles Lemma 6 in Jung et al. [2020], but has the advantage of being independent of the range of the queries.

**Lemma 3.1** (Sample accuracy implies posterior accuracy). *Given a function $\delta : \mathbb{R} \to [0, 1]$ and an analyst $A$, if a mechanism $M$ is $(\epsilon, \delta(\epsilon))$-sample accurate for all $\epsilon > 0$, then $M$ is $(\epsilon, \delta'(\epsilon))$-posterior accurate for $\delta'(\epsilon) \coloneqq \inf_{\xi \in (0, \epsilon)} \left( \frac{1}{\xi} \int_{\epsilon - \xi}^{\infty} \delta(t) \, dt \right)$.*

We use this lemma to provide accuracy guarantees for the Gaussian mechanism.

---

[4]In the case of an adaptive process, one can also consider the case where $\eta_i$ are adaptively chosen by the analyst and provided to the mechanism as the auxiliary parameter $\theta_i$.

**Lemma 3.2** (Accuracy of Gaussian mechanism). *Given $\eta > 0$, the Gaussian mechanism with noise parameter $\eta$ that receives $k$ queries is $(\epsilon, \delta(\epsilon))$-sample accurate for $\delta(\epsilon) := \frac{2k}{\sqrt{\pi}} e^{-\frac{\epsilon^2}{2\eta^2}}$, and $(\epsilon, \delta(\epsilon))$-posterior accurate for $\delta(\epsilon) := 4k \cdot e^{-\frac{\epsilon^2}{4\eta^2}}$.*

In order to complete the triangle inequality, we have to define the stability of the mechanism. Bayes stability captures the concept that the results returned by a mechanism and the queries selected by the adaptive adversary are such that the queries behave similarly on the true data distribution and on the posterior distribution induced by those results. This notion first appeared in Jung et al. [2020], under the name *Posterior Sensitivity*, as did the following theorem.

**Definition 3.3** (Bayes stability). Given $\epsilon, \delta > 0$ and an analyst $A$, we say $M$ is $(\epsilon, \delta)$-*Bayes stable* with respect to $A$, if

$$\Pr_{\substack{S \sim D^{(n)} \\ V \sim M(S,A), Q \sim A(V)}} \left[ \left| Q\left(D^V\right) - Q\left(D\right) \right| > \epsilon \right] \leq \delta.$$

Bayes stability and sample accuracy (implying posterior accuracy) combine via a triangle inequality to give distribution accuracy.

**Theorem 3.4** (Generalization). *Given two functions $\delta_1 : \mathbb{R} \to [0,1]$, $\delta_2 : \mathbb{R} \to [0,1]$, and an analyst $A$, if a mechanism $M$ is $(\epsilon, \delta_1(\epsilon))$-Bayes stable and $(\epsilon, \delta_2(\epsilon))$-sample accurate with respect to $A$, then $M$ is $(\epsilon, \delta'(\epsilon))$-distribution accurate for $\delta'(\epsilon) := \inf_{\epsilon' \in (0,\epsilon), \xi \in (0,\epsilon-\epsilon')} \left( \delta_1(\epsilon') + \frac{1}{\xi} \int_{\epsilon-\epsilon'-\xi}^{\infty} \delta_2(t) \, dt \right).$*

Since achieving posterior accuracy is relatively straightforward, guaranteeing Bayes stability is the main challenge in leveraging this theorem to achieve distribution accuracy with respect to adaptively chosen queries. The following lemma gives a useful and intuitive characterization of the quantity that the Bayes stability definition requires be bounded. Simply put, the Bayes factor $K(\cdot, \cdot)$ (defined in the lemma below) represents the amount of information leaked about the dataset during the interaction with an analyst, by moving from the prior distribution over data elements to the posterior induced by some view $v$. The degree to which a query $q$ overfits to the dataset is expressed by the correlation between the query and that Bayes factor. This simple lemma is at the heart of the progress that we make in this paper, both in our intuitive understanding of adaptive data analysis, and in the concrete results we show in subsequent sections. Its corresponding version for arbitrary queries are presented in Section C.2.

**Lemma 3.5** (Covariance stability). *Given a view $v \in \mathcal{V}$ and a linear query $q$,*

$$q\left(D^v\right) - q\left(D_{\mathcal{X}}\right) = \operatorname*{Cov}_{X \sim D_{\mathcal{X}}} \left( q(X), K(X, v) \right).$$

*Furthermore, given $\Delta, \sigma > 0$,*

$$\sup_{q \in \mathcal{Q} \text{ s.t. } \Delta_q \leq \Delta} \left| q\left(D^v\right) - q\left(D_{\mathcal{X}}\right) \right| = \Delta \cdot \mathbf{D}_{TV}\left(D^v \| D_{\mathcal{X}}\right)$$

*and*

$$\sup_{q \in \mathcal{Q} \text{ s.t. } \sigma_q^2 \leq \sigma^2} \left| q\left(D^v\right) - q\left(D_{\mathcal{X}}\right) \right| = \sigma \sqrt{\mathbf{D}_{\chi^2}\left(D^v \| D_{\mathcal{X}}\right)},$$

*where $K(x, v) := \frac{D(x \mid v)}{D(x)} = \frac{D(v \mid x)}{D(v)}$ is the Bayes factor of $x$ given $v$ (and vice-versa), $\mathbf{D}_{TV}$ is the total variation distance (Definition B.1), and $\mathbf{D}_{\chi^2}$ is the chi-square divergence (Definition B.2).*

*Proof.* By definition, $q\left(D^v\right) = \operatorname*{\mathbb{E}}_{X \sim D^v}\left[q(X)\right] = \operatorname*{\mathbb{E}}_{X \sim D}\left[K(X, v) \, q(X)\right]$, so

$$q\left(D^v\right) - q\left(D\right) = \operatorname*{\mathbb{E}}_{X \sim D}\left[K(X, v) \, q(X)\right] - \overbrace{\operatorname*{\mathbb{E}}_{X \sim D}\left[K(X, v)\right]}^{=1} \cdot \overbrace{\operatorname*{\mathbb{E}}_{X \sim D}\left[q(X)\right]}^{=q(D)} = \operatorname*{Cov}_{X \sim D}\left(q(X), K(X, v)\right).$$

The second part is a direct result of the known variational representation of total variation distance and $\chi^2$ divergence, which are both $f$-divergences (see Equations 7.88 and 7.91 in Polyanskiy and Wu [2022] for more details). $\qquad\square$

Using the first part of the lemma, we guarantee Bayes stability by bounding the correlation between specific $q$ and $K(\cdot, v)$ as discussed in Section 6. The second part of this Lemma implies that bounding the appropriate divergence is necessary and sufficient for bounding the Bayes stability of the worst query in the corresponding family, which is how the main theorems of this paper are all achieved, using the next corollary.

**Corollary 3.6.** *Given an analyst A, for any $\epsilon > 0$ we have*

$$\Pr_{\substack{S \sim D^{(n)} \\ V \sim M(S,A), Q \sim A(V)}} \left[ \left| Q\left(D^V\right) - Q\left(D\right) \right| > \sigma_Q \cdot \epsilon \right] \leq \Pr_{\substack{S \sim D^{(n)} \\ V \sim M(S,A)}} \left[ \mathbf{D}_{\chi^2} \left(D^v_{\mathcal{X}} \| D_{\mathcal{X}}\right) > \epsilon^2 \right].$$

The corresponding version of this corollary for bounded range queries provides an alternative proof to the generalization guarantees of the the LS stability notion [Shenfeld and Ligett, 2019].

## 4 Pairwise concentration

In order to leverage Lemma 3.5, we need a stability notion that implies Bayes stability of query responses in a manner that depends on the actual datasets and the actual queries (not just the worst case). In this section we propose such a notion and prove several key properties of it. Missing proofs from this section can be found in Appendix D.

We start by introducing a measure of the stability loss of the mechanism.

**Definition 4.1** (Stability loss). Given two sample sets $s, s' \in \mathcal{X}^n$, a query $q$, and a response $r \in \mathcal{R}$, we denote by $\ell_q(s, s'; r) := \ln\left(\frac{D^q(r \mid s)}{D^q(r \mid s')}\right)$ the *stability loss* of $r$ between $s$ and $s'$ for $q$ (we omit $q$ from notation for simplicity).[5] Notice that if $R \sim M(s, q)$, the stability loss defines a random variable. Similarly, given two elements $x, y \in \mathcal{X}$, we denote $\ell(x, y; r) := \ln\left(\frac{D(r \mid x)}{D(r \mid y)}\right)$ and $\ell(x; r) := \ln\left(\frac{D(r \mid x)}{D(r)}\right)$. This definition extends to views as well.

Next we introduce a notion of a bound on the stability loss, where the bound is allowed to depend on the pair of swapped inputs and on the issued queries.

**Definition 4.2** (Pairwise concentration). Given a non-negative function $\varphi : \mathcal{X}^n \times \mathcal{X}^n \times \mathcal{Q} \to \mathbb{R}^+$ which is symmetric in its first two arguments, a mechanism $M$ will be called $\varphi$-*Pairwise Concentrated* (or PC, for short), if for any two datasets $s, s' \in \mathcal{X}^n$ and $q \in \mathcal{Q}$, for any $\alpha \geq 0$,[6]

$$\mathbb{E}_{R \sim M(s,q)} \left[\exp\left((\alpha - 1)\ell(s, s'; R)\right)\right] \leq \exp\left(\alpha(\alpha - 1)\varphi(s, s'; q)\right).$$

Given a non-negative function $\varphi : \mathcal{X}^n \times \mathcal{X}^n \times \mathcal{V} \to \mathbb{R}^+$ which is symmetric in its first two arguments and an analyst $A$, a mechanism $M$ will be called $\varphi$-Pairwise Concentrated with respect to $A$, if for any $s, s' \in \mathcal{X}^n$ and $\alpha \geq 0$,

$$\mathbb{E}_{V \sim M(s,A)} \left[\exp\left((\alpha - 1)\left(\ell(s, s'; V) - \alpha\varphi(s, s'; V)\right)\right)\right] \leq 1.$$

We refer to such a function $\varphi$ as a *similarity function* over responses or views, respectively.

The similarity function serves as a measure of the local sensitivity of the issued queries with respect to the replacement of the two datasets, by quantifying the extent to which they differ from each other with respect to the query $q$. The case of noise addition mechanisms provides a natural intuitive interpretation, where the $\varphi$ scales with the difference between $q(s)$ and $q(s')$, which governs how much observing the response $r$ distinguishes between the two datasets, as stated in the next lemma.

We note that the first part of this definition can be viewed as a refined version of zCDP (Definition B.18), where the bound on the Rényi divergence (Definition B.5) is a function of the sample sets

---

[5]This quantity is sometimes referred to as *privacy loss* in the context of differential privacy, or *information density* in the context of information theory.

[6]The requirement for the range $0 \leq \alpha \leq 1$ might appear somewhat counter-intuitive, and results from the fact that $\varphi$ provides simultaneously a bound on the stability loss random variable's mean (the $-(\alpha - 1)$ coefficient) and variance proxy (the $(\alpha - 1)^2$ coefficient), as discusses in Definitions B.17, B.18.

and the query. As for the second part, since the bound depends on the queries, which themselves are random variables, it should be viewed as a bound on the Rényi dissimilarity notion that we introduce in the appendix (Definition B.9). This kind of extension is not limited to Rényi divergence, as discussed in Appendix B.

**Lemma 4.3** (Gaussian mechanism is PC). *Given $\eta > 0$ and an analyst $A$, if $M$ is a Gaussian mechanism with noise parameter $\eta$, then it is $\varphi$-PC for $\varphi\left(s, s'; q\right) := \frac{\left(q(s) - q\left(s'\right)\right)^2}{2\eta^2}$ and $\varphi$-PC with respect to $A$ for $\varphi\left(s, s'; v\right) := \frac{\left\|\bar{q}_v(s) - \bar{q}_v\left(s'\right)\right\|^2}{2\eta^2}$.*

To leverage this stability notion in an adaptive setting, it must hold under adaptive composition, which we prove in the next theorem.

**Theorem 4.4** (PC composition). *Given $k \in \mathbb{N}$, a similarity function over responses $\varphi$, and an analyst $A$ issuing $k$ queries, if a mechanism $M$ is $\varphi$-PC, then it is $\tilde{\varphi}$-PC with respect to $A$, where $\tilde{\varphi}\left(s, s'; v\right) := \sum_{i=1}^{k} \varphi\left(s, s'; q_i\right)$.*

Before we state the stability properties of PC mechanisms, we first transition to its element-wise version, which while less general than the original definition, is more suited for our use.

**Lemma 4.5** (Element-wise PC). *Given a similarity function $\varphi$ and an analyst $A$, if a mechanism $M$ that is $\varphi$-PC with respect to $A$ receives an iid sample from $\mathcal{X}^n$, then for any two elements $x, y \in \mathcal{X}$ and $\alpha \geq 1$ we have*

$$\mathbb{E}_{V \sim D(\cdot \mid x)} \left[\exp\left((\alpha - 1)\left(\ell\left(x, y; V\right) - \alpha\varphi\left(x, y; V\right)\right)\right)\right] \leq 1$$

*and*

$$\mathbb{E}_{V \sim D(\cdot \mid x)} \left[\exp\left((\alpha - 1)\left(\ell\left(x; V\right) - \alpha\varphi\left(x; V\right)\right)\right)\right] \leq 1,$$

*where $\varphi\left(x, y; v\right) := \sup_{s \in \mathcal{X}^{n-1}} \left(\varphi\left((s, x), (s, y); v\right)\right)$ and $\varphi\left(x; v\right) := \ln\left(\mathbb{E}_{Y \sim D}\left[e^{\varphi(x, Y; v)}\right]\right)$.*

Finally, we bound the Bayes stability of PC mechanisms.

**Theorem 4.6** (PC stability). *Given a similarity function over views $\varphi$ and an analyst $A$, if a mechanism $M$ that is $\varphi$-PC with respect to $A$ receives an iid sample from $\mathcal{X}^n$, then for any $\epsilon, \delta > 0$, $\phi \geq \mathbb{E}_{\substack{X \sim D, S \sim D^{(n)} \\ V \sim M(S, A)}}\left[\varphi\left(X; V\right)\right]$, we have*

$$\Pr_{\substack{S \sim D^{(n)} \\ V \sim M(S, A), Q \sim A(V)}} \left[\left|Q\left(D_{\mathcal{X}}^v\right) - Q\left(D_{\mathcal{X}}\right)\right| > \sigma_Q \epsilon\right]$$

$$\leq \Pr_{\substack{S \sim D^{(n)} \\ V \sim M(S, A)}} \left[\mathbb{E}_{X \sim D}\left[e^{27\ln\left(\frac{1}{\delta}\right)(\varphi(X; V) + \phi)}\right] > 1 + \frac{\epsilon^2}{6}\right] + O\left(\frac{e^\phi \delta}{\phi \epsilon^2}\right),$$

*where $\varphi\left(x; v\right)$ is as defined in Lemma 4.5.*

An exact version of the bound can be found in Theorem D.10.

*Proof outline.* From Corollary 3.6,

$$\Pr_{\substack{S \sim D^{(n)} \\ V \sim M(S, A), Q \sim A(V)}} \left[\left|Q\left(D_{\mathcal{X}}^v\right) - Q\left(D_{\mathcal{X}}\right)\right| > \sigma_Q \epsilon\right] \leq \Pr_{\substack{S \sim D^{(n)} \\ V \sim M(S, A)}} \left[\mathbf{D}_{\chi^2}\left(D_{\mathcal{X}}^v \| D_{\mathcal{X}}\right) > \epsilon^2\right].$$

Denoting $\varepsilon\left(x; v\right) := \varphi\left(x; v\right) + \phi + 4\sqrt{\ln\left(\frac{1}{\delta}\right)\left(\varphi\left(x; v\right) + \phi\right)}$, this can be bounded by

$$\Pr_{\substack{S \sim D^{(n)} \\ V \sim M(S, A)}} \left[\mathbf{D}_{\chi^2}\left(D_{\mathcal{X}}^v \| D_{\mathcal{X}}\right) > \mathbb{E}_{X \sim D}\left[\left(e^{\varepsilon(X; V)} - 1\right)^2\right] + \frac{\epsilon^2}{2}\right] + \Pr_{\substack{S \sim D^{(n)} \\ V \sim M(S, A)}} \left[\mathbb{E}_{X \sim D}\left[\left(e^{\varepsilon(X; V)} - 1\right)^2\right] > \frac{\epsilon^2}{2}\right].$$

We then show via algebraic manipulation that this second term can be bounded using the inequality

$$\mathbb{E}_{X \sim D}\left[\left(e^{\varepsilon(X; v)} - 1\right)^2\right] \leq 3\left(\mathbb{E}_{X \sim D}\left[e^{27\ln\left(\frac{1}{\delta}\right)(\varphi(X; v) + \phi)}\right] - 1\right),$$

and, recalling the definition of the chi square divergence between $P$ and $Q$ (Definition B.2) $\mathbf{D}_{\chi^2}(P\|Q) := \mathop{\mathbb{E}}_{X\sim D_2}\left[\left(\frac{P(X)}{Q(X)}-1\right)^2\right]$, the first term can be bounded using Markov's inequality followed by the Cauchy-Schwarz inequality by the term:

$$\frac{2}{\epsilon^2}\mathop{\mathbb{E}}_{\substack{X\sim D, S\sim D^{(n)}\\ V\sim M(S,A)}}\left[\left(\frac{D(V\mid X)}{D(V)}-1\right)^2\cdot\mathbb{1}_{B(\varepsilon)}((X,V))\right]$$

$$\leq \frac{2}{\epsilon^2}\sqrt{\overbrace{\left(\mathop{\mathbb{E}}_{\substack{X\sim D, S\sim D^{(n)}\\ V\sim M(S,A)}}\left[\left(\frac{D(V\mid X)}{D(V)}\right)^4\right]+1\right)}^{*}\cdot\overbrace{\mathop{\Pr}_{\substack{X\sim D, S\sim D^{(n)}\\ V\sim M(S,A)}}[(X,V)\in B(\varepsilon)]}^{**}}$$

where $B(\varepsilon) := \left\{(x,v)\in\mathcal{X}\times\mathcal{V}\mid \left|\ln\left(\frac{D(v\mid x)}{D(v)}\right)\right|>\varepsilon(x;v)\right\}$.

We then bound the quantity * by $O(e^\phi)$, and we bound ** by $O\left(\frac{\delta^2}{\phi}\right)$, which completes the proof. $\qquad\square$

Combining this theorem with Theorem 3.4 we get that PC and sample accuracy together imply distribution accuracy.

# 5 Variance-based generalization guarantees for the Gaussian mechanism

In this section we leverage the theorems of the previous sections to prove variance-based generalization guarantees for the Gaussian mechanism under adaptive data analysis. All proofs from this section appear in Appendix E.

We first provide generalization guarantees for bounded queries.

**Theorem 5.1** (Generalization guarantees for bounded queries). *Given $k\in\mathbb{N}$; $\Delta,\sigma,\epsilon\geq 0$; $0<\delta\leq\frac{1}{e}$; and an analyst $A$ issuing $k$ $\Delta$-bounded linear queries with variance bounded by $\sigma^2$, if $M$ is a Gaussian mechanism with noise parameter $\eta=\Theta\left(\sigma\sqrt{\frac{\sqrt{k}}{n}}\right)$ that receives an iid dataset of size $n=\Omega\left(\max\left\{\frac{\Delta}{\epsilon},\frac{\sigma^2}{\epsilon^2}\right\}\sqrt{k}\cdot\ln\left(\frac{k\sigma}{\delta\epsilon}\right)\right)$, then $M$ is $(\epsilon,\delta)$-distribution accurate.*

*An exact version of the bound can be found in Theorem E.4.*

This theorem provides significant improvement over similar results that were achieved using differential privacy (see, e.g., Theorem 13 in Jung et al. [2020], which is defined for $\Delta=1$), by managing to replace the $\Delta^2$ term with $\sigma^2$. The remaining dependence on $\Delta$ is inevitable, in the sense that such a dependence is needed even for non-adaptive analysis. This improvement resembles the improvement provided by Bernstein's inequality over Hoeffding's inequality.

This generalization guarantee, which nearly avoids dependence on the range of the queries, begs the question of whether it is possible to extend these results to handle unbounded queries. Clearly such a result would not be true without some bound on the tail distribution for a single query, so we focus in the next theorem on the case of sub-Gaussian queries. Formally, we will consider the case where $q(X)-q(D)$ is a sub-Gaussian random variable, for all queries.

**Definition 5.2** (Sub-Gaussian random variable). *Given $\sigma>0$, a random variable $X\in\mathbb{R}$ will be called $\sigma^2$-sub-Gaussian if for any $\lambda\in\mathbb{R}$ we have $\mathbb{E}\left[e^{\lambda X}\right]\leq e^{\frac{\lambda^2}{2}\nu}$.*

We can now state our result for unbounded queries.

**Theorem 5.3** (Generalization guarantees for sub-Gaussian queries). *Given $k\in\mathbb{N}$; $\sigma,\epsilon\geq 0$; $0<\delta\leq\frac{1}{e}$; and an analyst $A$ issuing $k$ $\sigma^2$-sub-Gaussian linear queries, if $M$ is a Gaussian mechanism with noise parameter $\eta=\Theta\left(\sigma\sqrt{\frac{\sqrt{k}}{n}}\right)$ that receives an iid dataset of size $n=\Omega\left(\frac{\sigma^2}{\epsilon^2}\sqrt{k}\ln\left(\frac{k\sigma}{\delta\epsilon}\right)\right)$, then $M$ is $(\epsilon,\delta)$-distribution accurate.*

*An exact version of the bound can be found in Theorem E.10.*

These results extend to the case where the variance (or variance proxy) of each query $q_i$ is bounded by a unique value $\sigma_i^2$, by simply passing this value to the mechanism as auxiliary information and scaling the added noise $\eta_i$ accordingly. Furthermore, using this approach we can quantify the extent to which incorrect bounds affect the accuracy guarantee. Overestimating the bound on a query's variance would increase the error of the response to this query by a factor of square root of the ratio between the assumed and the correct variance, while the error of the other responses would only decrease. On the other hand, underestimating the bound on a query's variance would only decrease the error of the response to this query, while increasing the error of each subsequent query by a factor of the square root of the ratio between the assumed and the correct variance, divided by the number of subsequent queries. A formal version of this claim can be found in Section E.3.

## 6   Discussion

The contribution of this paper is two-fold. In Section 3, we provide a tight measure of the level of overfitting of some query with respect to previous responses. In Sections 4 and 5, we demonstrate a toolkit to utilize this measure, and use it to prove new generalization properties of fundamental noise-addition mechanisms. The novelty of the PC definition stems replacing the fixed parameters that appear in the differential privacy definition with a function of the datasets and the query. The definition presented in this paper provides a generalization of zero-concentrated differential privacy, and future work could study similar generalizations of other privacy notions, as discussed in Section B.4.

One small extension of the present work would be to consider queries with range $\mathbb{R}^d$. It would also be interesting to extend our results to handle arbitrary normed spaces, using appropriate noise such as perhaps the Laplace mechanism. It might also be possible to relax our assumption that data elements are drawn iid to a weaker independence requirement. Furthermore, it would be interesting to explore an extension from linear queries to general low-sensitivity queries.

We hope that the mathematical toolkit that we establish in Appendix B to analyze our stability notion may find additional applications, perhaps also in context of privacy accounting. Furthermore, the max divergence can be generalized analogously to the "dynamic" generalization of Rényi divergence proposed in this paper (B.9), perhaps suggesting that this approach may be useful in analyzing other mechanisms as well.

Our Covariance Lemma (3.5) shows that there are two possible ways to avoid adaptivity-driven overfitting—by bounding the Bayes factor term, which induces a bound on $|q(D^v) - q(D)|$, as we do in this work, or by bounding the correlation between $q$ and $K(\cdot, v)$. This second option suggests interesting directions for future work. For example, to capture an analyst that is non-worst-case in the sense that she "forgets" some of the information that she has learned about the dataset, both the posterior accuracy and the Bayes stability could be redefined with respect to the internal state of the analyst instead of with respect to the full view. This could allow for improved bounds in the style of Zrnic and Hardt [2019].

## Acknowledgments and Disclosure of Funding

We gratefully acknowledge productive discussions with Etam Benger, Vitaly Feldman, Yosef Rinott, Aaron Roth, and Tomer Shoham. This work was supported in part by a gift to the McCourt School of Public Policy and Georgetown University, Simons Foundation Collaboration 733792, Israel Science Foundation (ISF) grant 2861/20, and a grant from the Israeli Council for Higher Education. Shenfeld's work was also partly supported by the Apple Scholars in AI/ML PhD Fellowship. Part this work was completed while Ligett was visiting Princeton University's Center for Information Technology Policy.

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
