} |q(D^v) - q(D_\mathcal{X})| = \Delta \cdot \mathbf{D}_{TV}(D^v \| D_\mathcal{X})$$

*and*

$$\sup_{q \in \mathcal{Q} \text{ s.t. } \sigma_q^2 \leq \sigma^2} |q(D^v) - q(D_\mathcal{X})| = \sigma \sqrt{\mathbf{D}_{\chi^2}(D^v \| D_\mathcal{X})},$$

*where $K(x, v) := \frac{D(x \mid v)}{D(x)} = \frac{D(v \mid x)}{D(v)}$ is the Bayes factor of $x$ given $v$ (and vice-versa), $\mathbf{D}_{TV}$ is the total variation distance (Definition B.1), and $\mathbf{D}_{\chi^2}$ is the chi-square divergence (Definition B.2).*

*Proof.* By definition, $q(D^v) = \underset{X \sim D^v}{\mathbb{E}} [q(X)] = \underset{X \sim D}{\mathbb{E}} [K(X, v) q(X)]$, so

$$q(D^v) - q(D) = \underset{X \sim D}{\mathbb{E}} [K(X, v) q(X)] - \overbrace{\underset{X \sim D}{\mathbb{E}} [K(X, v)]}^{=1} \cdot \overbrace{\underset{X \sim D}{\mathbb{E}} [q(X)]}^{=q(D)} = \underset{X \sim D}{\mathrm{Cov}} (q(X), K(X, v)).$$

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

*where $\varphi(x, y; v) := \sup_{s \in \mathcal{X}^{n-1}} (\varphi((s, x), (s, y); v))$ and $\varphi(x; v) := \ln\left( \mathbb{E}_{Y \sim D} \left[ e^{\varphi(x, Y; v)} \right] \right)$.*

Finally, we bound the Bayes stability of PC mechanisms.

**Theorem 4.6** (PC stability). *Given a similarity function over views $\varphi$ and an analyst A, if a mechanism M that is $\varphi$-PC with respect to A receives an iid sample from $\mathcal{X}^n$, then for any $\epsilon, \delta > 0$, $\phi \geq \mathbb{E}_{\substack{X \sim D, S \sim D^{(n)} \\ V \sim M(S,A)}} [\varphi(X; V)]$, we have*

$$\Pr_{\substack{S \sim D^{(n)} \\ V \sim M(S,A), Q \sim A(V)}} \left[ |Q(D_{\mathcal{X}}^v) - Q(D_{\mathcal{X}})| > \sigma_Q \epsilon \right]$$

$$\leq \Pr_{\substack{S \sim D^{(n)} \\ V \sim M(S,A)}} \left[ \mathbb{E}_{X \sim D} \left[ e^{27\ln\left(\frac{1}{\delta}\right)(\varphi(X;V) + \phi)} \right] > 1 + \frac{\epsilon^2}{6} \right] + O\left( \frac{e^\phi \delta}{\phi \epsilon^2} \right),$$

*where $\varphi(x; v)$ is as defined in Lemma 4.5.*

An exact version of the bound can be found in Theorem D.10.

*Proof outline.* From Corollary 3.6,

$$\Pr_{\substack{S \sim D^{(n)} \\ V \sim M(S,A), Q \sim A(V)}} \left[ |Q(D_{\mathcal{X}}^v) - Q(D_{\mathcal{X}})| > \sigma_Q \epsilon \right] \leq \Pr_{\substack{S \sim D^{(n)} \\ V \sim M(S,A)}} \left[ \mathbf{D}_{\chi^2}(D_{\mathcal{X}}^v \| D_{\mathcal{X}}) > \epsilon^2 \right].$$

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

# A  Formal definitions

## A.1  Distributions of interest

**Definition A.1** (Distributions over $\mathcal{X}^n$ and $\mathcal{R}$). A distribution $D_{\mathcal{X}^n}$, a query $q$, and a mechanism $M : \mathcal{X}^n \times \mathcal{Q} \to \mathcal{R}$, together induce a set of distributions over $\mathcal{X}^n$, $\mathcal{R}$, and $\mathcal{X}^n \times \mathcal{R}$.

The *conditional distribution* $D_{\mathcal{R}|\mathcal{X}^n}^q$ over $\mathcal{R}$ represents the probability to get $r$ as the output of $M(s,q)$. That is, $\forall s \in \mathcal{X}^n, r \in \mathcal{R}$, if $\mathcal{R}$ is countable,

$$D_{\mathcal{R}|\mathcal{X}^n}^q (r \,|\, s) := \Pr_{R \sim M(s,q)} [R = r \,|\, s],$$

where the probability is taken over the internal randomness of $M$. In case $\mathcal{R}$ is a measurable spaces, it represents the probability density function (the Radon–Nikodym derivative).

The *joint distribution* $D_{(\mathcal{X}^n, \mathcal{R})}^q$ over $\mathcal{X}^n \times \mathcal{R}$ represents the probability to sample a particular $s$ and get $r$ as the output of $M(s,q)$. That is, $\forall s \in \mathcal{X}^n, r \in \mathcal{R}$,

$$D_{(\mathcal{X}^n, \mathcal{R})}^q (s, r) := D_{\mathcal{X}^n} (s) \cdot D_{\mathcal{R}|\mathcal{X}^n}^q (r \,|\, s).$$

The *marginal distribution* $D_{\mathcal{R}}^q$ over $\mathcal{R}$ represents the prior probability to get output $r$ without any knowledge of $s$. That is, $\forall r \in \mathcal{R}$,

$$D_{\mathcal{R}}^q (r) := \int_{s \in \mathcal{X}^n} D_{(\mathcal{X}^n, \mathcal{R})}^q (s, r) \, dr.$$

The *conditional distribution* $D_{\mathcal{X}^n|\mathcal{R}}^q$ over $\mathcal{X}^n$ represents the posterior probability that the input dataset to $M$ was $s$, given that $M(\cdot, q)$ returns $r$. That is, $\forall s \in \mathcal{X}^n, r \in \mathcal{R}$,

$$D_{\mathcal{X}^n|\mathcal{R}}^q (s \,|\, r) := \frac{D_{(\mathcal{X}^n, \mathcal{R})}^q (s, r)}{D_{\mathcal{R}}^q (r)}.$$

**Definition A.2** (Distributions over $\mathcal{X}$ and $\mathcal{R}$). The *marginal distribution* $D_{\mathcal{X}}$ over $\mathcal{X}$ represents the probability to get $x$ by sampling a dataset and then sampling one element from that dataset uniformly at random. That is, $\forall x \in \mathcal{X}$,

$$D_{\mathcal{X}} (x) := \int_{s \in \mathcal{X}^n} D_{\mathcal{X}^n} (s) \cdot D_{\mathcal{X}|\mathcal{X}^n} (x \,|\, s) \, ds,$$

where $D_{\mathcal{X}|\mathcal{X}^n} (x \,|\, s)$ denotes the probability to get $x$ by sampling $s$ uniformly at random.

The *joint distribution* $D_{(\mathcal{X}, \mathcal{R})}^q$ over $\mathcal{X} \times \mathcal{R}$ represents the probability to get $x$ by sampling a dataset uniformly at random and also get $r$ as the output of $M(\cdot, q)$ from the same dataset. That is, $\forall x \in \mathcal{X}, r \in \mathcal{R}$,

$$D_{(\mathcal{X}, \mathcal{R})}^q (x, r) := \int_{s \in \mathcal{X}^n} D_{\mathcal{X}^n} (s) \cdot D_{\mathcal{X}|\mathcal{X}^n} (x \,|\, s) \cdot D_{\mathcal{R}|\mathcal{X}^n}^q (r \,|\, s) \, ds.$$

where $D_{\mathcal{X}|\mathcal{X}^n} (x \,|\, s)$ denotes the probability to get $x$ by sampling $s$ uniformly at random.

The *conditional distribution* $D_{\mathcal{X}|\mathcal{R}}^q$ over $\mathcal{X}$ represents the probability to get $x$ by sampling a dataset uniformly at random, given the fact that we got $r$ as the output of $M(\cdot, q)$ from that dataset. That is, $\forall x \in \mathcal{X}, r \in \mathcal{R}$,

$$D^r (x) := D_{\mathcal{X}|\mathcal{R}}^q (x \,|\, r) := \int_{s \in \mathcal{X}^n} D_{\mathcal{X}^n|\mathcal{R}}^q (s \,|\, r) \cdot D_{\mathcal{X}|\mathcal{X}^n} (x \,|\, s) \, ds.$$

Although all of these definitions depend on $D_{\mathcal{X}^n}$ and $M$, we typically omit these from the notation for simplicity, and usually omit the superscripts and subscripts entirely. We include them only when necessary for clarity.

Though the conditional distributions $D_{\mathcal{R}|\mathcal{X}}^q$ and $D_{\mathcal{X}|\mathcal{R}}^q$ were not defined as the ratio between the joint and marginal distribution, the analogue of Bayes' rule still holds for these distributions. A formal proof of this claim can be found in Appendix A in Shenfeld and Ligett [2019].

## A.2 A Formal treatment of adaptivity

**Definition A.3** (Adaptive mechanism). Illustrated below, the *adaptive mechanism* $\mathrm{Adp}_M : \mathcal{X}^n \times \mathcal{A} \to \mathcal{V}_k$ is a particular type of mechanism, which inputs an analyst as its query and which returns a view as its response and is parametrized by a mechanism $M$ and number of iterations $k$. Given a dataset $s$ and an analyst $A$ as input, the adaptive mechanism iterates $k$ times through the process where $A$ sends a query and auxiliary information to $M$ and receives its response to that query on the dataset. The adaptive mechanism returns the resulting sequence of $k$ responses $v_k$. Naturally, this requires $A$ to match $M$ such that $M$'s output can be $A$'s input, and vice versa.

If $A$ is randomized, we add one more step at the beginning where $\mathrm{Adp}_M$ randomly generates some bits $c$—$A$'s "coin tosses." In this case, $v_k := (c, r_1, \ldots, r_{ik})$ and $A$ receives the coin tosses as an input as well. This addition turns $q_{k+1}$ and $\theta_{k+1}$ into a deterministic function of $v_i$ for any $i \in \mathbb{N}$, a fact that we later rely on. In this situation, the randomness of $\mathrm{Adp}_M$ results both from the randomness of the coin tosses and from that of $M$. We will denote $M(s, A) := \mathrm{Adp}_M(s, A)$ for simplicity.

---

Adaptive mechanism $\mathrm{Adp}_M$

**Input:** $s \in \mathcal{X}^n$, $A \in \mathcal{A}$
**Output:** $v_k \in \mathcal{V}_k$
$v_0 \leftarrow \emptyset$ or $c$
**for** $i \in [k]$ :
   $(q_i, \theta_i) \leftarrow A(v_{i-1})$
   $r_i \leftarrow M(s, q_i, \theta_i)$
   $v_i \leftarrow (v_{i-1}, r_i)$
**return** $v_k$

---

Using the adaptive mechanism we can extend the set of distributions presented in Definitions A.1 and A.2, to distributions over $\mathcal{V}$ and $\mathcal{X}$, with $\mathcal{V}$ taking the place of $\mathcal{R}$ and $A$ replacing $q$.

# B  Divergence, dissimilarity, and stability measures

Given two probability distributions $P, Q$ over some domain $\Omega$, there are many useful measures of their dissimilarity. In this section we develop what is as far as we know a new type of dissimilarity measure between distributions, which generalizes the notion of divergences. We start by recalling several common divergence measures in Section B.1. Next, Section B.2, we present in a generalization of the divergence notion, where a scalar is replaced by a function, yielding dynamic measures which we refer to as dissimilarities. We then prove (Section B.3) several convexity properties for max and Rényi dissimilarities. Finally (Section B.4) we propose several novel stability measures which build on the proposed dissimilarity measures and are used throughout the paper.

## B.1  Divergence measures

Dissimilarity between distributions is typically measured using a divergence, that is, a function receiving two distributions as input, and outputting a non-negative value which is equal to $0$ if and only if the two distributions are identical almost surely. One commonly used family of divergences is the $f$-divergences, defined as $\mathbf{D}_f\left(P\|Q\right) \coloneqq \underset{\omega \sim Q}{\mathbb{E}}\left[f\left(\frac{P(\omega)}{Q(\omega)}\right)\right]$. Here we recall several instances which are used in this paper.

**Definition B.1** (Total variation distance). The *total variation distance* (also known as *statistical distance*) between $P$ and $Q$ is defined as

$$\mathbf{D}_{\text{TV}}\left(P\|Q\right) \coloneqq \frac{1}{2}\underset{\omega \sim Q}{\mathbb{E}}\left[\left|\frac{P(\omega)}{Q(\omega)} - 1\right|\right].$$

**Definition B.2** (Chi-square divergence). The *Chi-square divergence* (also known as Pearson's $\chi^2$ divergence) between $P$ and $Q$ is defined as

$$\mathbf{D}_{\chi^2}\left(P\|Q\right) \coloneqq \underset{\omega \sim Q}{\mathbb{E}}\left[\left(\frac{P(\omega)}{Q(\omega)} - 1\right)^2\right].$$

**Definition B.3** (KL divergence). The *Kullback–Leibler divergence* (or *KL divergence*) between $P$ and $Q$ is defined as

$$\mathbf{D}_{KL}\left(P\|Q\right) \coloneqq \underset{\omega \sim P}{\mathbb{E}}\left[\ln\left(\frac{P(\omega)}{Q(\omega)}\right)\right] = \underset{\omega \sim Q}{\mathbb{E}}\left[\frac{P(\omega)}{Q(\omega)} \cdot \ln\left(\frac{P(\omega)}{Q(\omega)}\right)\right].$$

Not all divergences are $f$-divergences. For example:

**Definition B.4** (($\delta$-Approximate) max divergence). Given $\delta \geq 0$, the *$\delta$-approximate max divergence* between $P$ and $Q$ is defined as

$$\mathbf{D}_\infty^\delta\left(P\|Q\right) \coloneqq \sup_{\boldsymbol{\omega} \subseteq \Omega \,|\, P(\boldsymbol{\omega}) \geq \delta}\left(\ln\left(\frac{P(\omega) - \delta}{Q(\omega)}\right)\right).$$

The case of $\delta = 0$ is simply called the max divergence and is denoted by $\mathbf{D}_\infty$.

These two last definitions are generalized by the following one:

**Definition B.5** (Rényi divergence). Given $\alpha \in [0, \infty]$, the *Rényi divergence* between two probability distributions $P$ and $Q$ is defined as

$$\mathbf{D}_\alpha\left(P\|Q\right) \coloneqq \frac{1}{\alpha - 1}\ln\left(\underset{\omega \sim P}{\mathbb{E}}\left[\left(\frac{P(\omega)}{Q(\omega)}\right)^{\alpha - 1}\right]\right).$$

The cases of $\alpha = 0, 1$, and $\infty$ are defined using the limit and converge to $-\ln\left(\underset{\omega \sim Q}{\Pr}\left[\omega > 0\right]\right)$, the KL divergence, and the max divergence, respectively.[7]

---

[7]This definition generalizes the original one introduced by Alfréd Rényi to include the range $[0, 1)$, which will become useful later in Theorem B.12. For more details, see Van Erven and Harremos [2014]

## B.2 Dissimilarity measures

In this section we present a generalization of the Rényi divergence and the max divergence. We start by recalling two useful identities that are a direct result of the definitions.

**Fact B.6.** *Given $\epsilon, \delta \geq 0$ and two probability distributions $P, Q$, we have $\mathbf{D}_\infty^\delta\left(P\|Q\right) \leq \epsilon$ if and only if for any $\boldsymbol{\omega} \subseteq \Omega$, $\Pr_{\omega \sim P}\left[\omega \in \boldsymbol{\omega}\right] \leq e^\epsilon \Pr_{\omega \sim Q}\left[\omega \in \boldsymbol{\omega}\right] + \delta$.*

**Fact B.7.** *Given $\phi \geq 0$, $\alpha \geq 1$, and two probability distributions $P, Q$, we have $\mathbf{D}_\alpha\left(P\|Q\right) \leq \phi$ if and only if*

$$\mathbb{E}_{\omega \sim P}\left[\exp\left((\alpha - 1)\ln\left(\frac{P\left(\omega\right)}{Q\left(\omega\right)}\right)\right)\right] \leq e^{(\alpha-1)\phi}.$$

Motivated by these two identities, we generalize these divergences by replacing the scalar bounding the divergence by a function, leading to a more "dynamic" measure that can no longer be regarded a divergence, hence the names.

We first generalize the notion of max divergence.

**Definition B.8** (Max dissimilarity). Given $\delta \geq 0$, and a non-negative function $\varepsilon : \Omega \to \mathbb{R}^+$, we say the *($\delta$-approximate) max dissimilarity* between $P$ and $Q$ is bounded by $\varepsilon$ if for any $\boldsymbol{\omega} \subseteq \Omega$,

$$\int_{\omega \in \boldsymbol{\omega}} P\left(\omega\right) d\omega \leq \int_{\omega \in \boldsymbol{\omega}} e^{\varepsilon(\omega)} Q\left(\omega\right) d\omega + \delta.$$

We denote this fact by $P \leq_{(\varepsilon,\delta)}^+ Q$.

Similarly, we denote $P \leq_{(\varepsilon,\delta)}^- Q$ if for any $\boldsymbol{\omega} \subseteq \Omega$,

$$\int_{\omega \in \boldsymbol{\omega}} e^{-\varepsilon(\omega)} P\left(\omega\right) d\omega \leq \int_{\omega \in \boldsymbol{\omega}} Q\left(\omega\right) d\omega + \delta.$$

Next we generalize the notion of Rényi divergence. This dissimilarity measure serves as the mathematical basis for the new stability measure that we introduce (Definition 4.2).

**Definition B.9** (Rényi dissimilarity). Given $\alpha \geq 0$ and a non-negative function $\varphi : \Omega \to \mathbb{R}^+$, we say the $\alpha$-*Rényi dissimilarity* between $P$ and $Q$ is bounded by $\varphi$ if

$$\mathbb{E}_{\omega \sim P}\left[\exp\left((\alpha - 1)\left(\ln\left(\frac{P\left(\omega\right)}{Q\left(\omega\right)}\right) - \varphi\left(\omega\right)\right)\right)\right] \leq 1.$$

The following theorem states several implications between the various max dissimilarity definitions.

**Theorem B.10.** *Given $\delta \geq 0$, a non-negative function $\varepsilon : \Omega \to \mathbb{R}^+$, and two distributions $P, Q$:*

1. $\Pr_{\omega \sim P}\left[\ln\left(\frac{P(\omega)}{Q(\omega)}\right) > \varepsilon\left(\omega\right)\right] \leq \delta$ *implies* $P \leq_{(\varepsilon,\delta)}^+ Q$,

2. $P \leq_{(\varepsilon,\delta)}^+ Q$ *implies* $P \leq_{(\varepsilon,\delta)}^- Q$,

3. $P \leq_{(\varepsilon,\delta)}^+ Q$ *implies* $\Pr_{\omega \sim P}\left[\ln\left(\frac{P(\omega)}{Q(\omega)}\right) > \varepsilon\left(\omega\right) + a\right] < \frac{e^a \delta}{e^a - 1}$ *for any $a > 0$, and*

4. $Q \leq_{(\varepsilon,\delta)}^- P$ *implies* $\Pr_{\omega \sim P}\left[\ln\left(\frac{Q(\omega)}{P(\omega)}\right) > \varepsilon\left(\omega\right) + a\right] < \frac{\delta}{e^a - 1}$ *for any $a > 0$.*

*Proof.* **Part 1**: Denoting $B_\varepsilon\left(P, Q\right) := \left\{\omega \in \Omega \,|\, \ln\left(\frac{P(\omega)}{Q(\omega)}\right) > \varepsilon\left(\omega\right)\right\}$ we have

$$\int_{\omega\in\boldsymbol{\omega}} P\left(\omega\right) d\omega = \int_{\omega\in\boldsymbol{\omega}\setminus B_\varepsilon(P,Q)} P\left(\omega\right) d\omega + \int_{\omega\in\boldsymbol{\omega}\cap B_\varepsilon(P,Q)} P\left(\omega\right) d\omega$$

$$\overset{(a)}{\le} \int_{\omega\in\boldsymbol{\omega}\setminus B_\varepsilon(P,Q)} e^{\varepsilon(\omega)} Q\left(\omega\right) d\omega + \int_{\omega\in B_\varepsilon(P,Q)} P\left(\omega\right) d\omega$$

$$= \int_{\omega\in\boldsymbol{\omega}\setminus B_\varepsilon(P,Q)} e^{\varepsilon(\omega)} Q\left(\omega\right) d\omega + \Pr_{\omega\sim P}\left[\ln\left(\frac{P\left(\omega\right)}{Q\left(\omega\right)}\right) > \varepsilon\left(\omega\right)\right]$$

$$\overset{(b)}{\le} \int_{\omega\in\boldsymbol{\omega}} e^{\varepsilon(\omega)} Q\left(\omega\right) d\omega + \delta,$$

where (a) results from the definition of $B_\varepsilon\left(P,Q\right)$ and (b) from the assumption that $\Pr_{\omega\sim P}\left[\ln\left(\frac{P(\omega)}{Q(\omega)}\right) > \varepsilon\left(\omega\right)\right] \le \delta$.

**Part 2**: Denoting $\delta\left(\omega\right) := \max\left\{P\left(\omega\right) - e^{\varepsilon(\omega)}Q\left(\omega\right), 0\right\}$, which implies $P\left(\omega\right) \le e^{\varepsilon(\omega)}Q\left(\omega\right) + \delta\left(\omega\right)$, we have

$$\int_{\omega\in\boldsymbol{\omega}} e^{-\varepsilon(\omega)} P\left(\omega\right) d\omega \overset{(a)}{\le} \int_{\omega\in\boldsymbol{\omega}} e^{-\varepsilon(\omega)}\left(e^{\varepsilon(\omega)}Q\left(\omega\right) + \delta\left(\omega\right)\right) d\omega$$

$$= \int_{\omega\in\boldsymbol{\omega}}\left(Q\left(\omega\right) + e^{-\varepsilon(\omega)}\cdot\delta\left(\omega\right)\right) d\omega$$

$$\overset{(b)}{\le} \int_{\omega\in\boldsymbol{\omega}} Q\left(\omega\right) d\omega + \int_{\omega\in\boldsymbol{\omega}}\delta\left(\omega\right) d\omega$$

$$\overset{(c)}{\le} \int_{\omega\in\boldsymbol{\omega}} Q\left(\omega\right) d\omega + \delta$$

where (a) results from the fact the inequality holds for any element $\omega\in\Omega$, (b) from the non-negativity of $\varepsilon$, and (c) from the assumption that $P\le^+_{(\varepsilon,\delta)} Q$.

**Part 3**: Denoting $\epsilon^+_Q := \ln\left(\mathbb{E}_{\omega\sim Q}\left[e^{\varepsilon(X)}\right]\right)$ and $Q^+_P\left(\omega\right) := e^{\varepsilon(\omega)-\epsilon^+_Q}Q\left(\omega\right)$, we notice that from the definition $Q^+_P$ is a valid distribution, since

$$\int_{\omega\in\Omega} Q^+_P\left(\omega\right) d\omega = e^{-\epsilon^+_Q}\int_{\omega\in\Omega} e^{\varepsilon(\omega)}Q\left(\omega\right) d\omega = \frac{1}{\mathbb{E}_{\omega\sim Q}\left[e^{\varepsilon(X)}\right]}\mathbb{E}_{\omega\sim Q}\left[e^{\varepsilon(X)}\right] = 1.$$

We first transform the $P\le^+_{(\varepsilon,\delta)} Q$ bound into a bound on the max divergence between $P$ and $Q^+_P$.

$$\Pr_{\omega\sim P}\left[\omega\in\boldsymbol{\omega}\right] \overset{(a)}{\le} \int_{\omega\in\boldsymbol{\omega}} e^{\varepsilon(\omega)}Q\left(\omega\right) d\omega + \delta \overset{(b)}{=} e^{\epsilon^+_Q}\int_{\omega\in\boldsymbol{\omega}} Q^+_P\left(\omega\right) d\omega + \delta = e^{\epsilon^+_Q}\Pr_{\omega\sim Q^+_P}\left[\omega\in\boldsymbol{\omega}\right] + \delta,$$

where (a) results from the assumption that $P\le^+_{(\varepsilon,\delta)} Q$ and (b) from the definition of $Q^+_P\left(\omega\right)$.

Next we denote $B^a_\varepsilon\left(P,Q\right) := \left\{\omega\in\Omega\,|\,\ln\left(\frac{P(\omega)}{Q(\omega)}\right) > \varepsilon\left(\omega\right) + a\right\}$, and notice that for any $\omega\in B^a_\varepsilon\left(P,Q\right)$ we have $P\left(\omega\right) > e^{\varepsilon(\omega)+a}Q\left(\omega\right) = e^{\epsilon^+_Q+a}Q^+_P\left(\omega\right)$, so

$$\Pr_{\omega \sim P} [\omega \in B_\varepsilon^a (P,Q)] \overset{(a)}{\leq} e^{\epsilon_Q^+} \Pr_{\omega \sim Q_P^+} [\omega \in B_\varepsilon^a (P,Q)] + \delta$$

$$= e^{\epsilon_Q^+} \int_{\omega \in B_\varepsilon^a(P,Q)} Q_P^+ (\omega)\, d\omega + \delta$$

$$\overset{(b)}{<} e^{\epsilon_Q^+} \int_{\omega \in B_\varepsilon^a(P,Q)} e^{-\left(\epsilon_Q^+ + a\right)} P(\omega)\, d\omega + \delta$$

$$= e^{-a} \Pr_{\omega \sim P} [\omega \in B_\varepsilon^a (P,Q)] + \delta$$

where (a) results from the previous bound and (b) from the fact the inequality holds for any element $\omega \in \Omega$.

Reordering the terms we get

$$\Pr_{\omega \sim Q} \left[ \ln \left( \frac{P(\omega)}{Q(\omega)} \right) > \varepsilon(\omega) + a \right] = \Pr_{\omega \sim P} [\omega \in B_\varepsilon^a (P,Q)] < \frac{e^a \delta}{e^a - 1}.$$

**Part 4**: Denoting $\epsilon_Q^- := -\ln \left( \mathbb{E}_{\omega \sim Q} \left[ e^{-\varepsilon(\omega)} \right] \right)$ and $Q_P^- (\omega) := e^{-\varepsilon(\omega) + \epsilon_Q^-} Q(\omega)$, we notice that from the definition $Q_P^-$ is a valid distribution, since

$$\int_{\omega \in \Omega} Q_P^- (\omega)\, d\omega = e^{\epsilon_Q^-} \int_{\omega \in \Omega} e^{-\varepsilon(\omega)} Q(\omega)\, d\omega = \frac{1}{\mathbb{E}_{\omega \sim Q} \left[ e^{-\varepsilon(\omega)} \right]} \mathbb{E}_{\omega \sim Q} \left[ e^{-\varepsilon(\omega)} \right] = 1.$$

We first transform the $Q \leq_{(\varepsilon,\delta)}^- P$ bound into a bound on the max divergence between $Q_P^-$ and $P$.

$$\Pr_{\omega \sim Q_P^-} [\omega \in \boldsymbol{\omega}] \overset{(a)}{=} e^{\epsilon_Q^-} \int_{\omega \in \boldsymbol{\omega}} e^{-\varepsilon(\omega)} Q(\omega)\, d\omega \overset{(b)}{\leq} e^{\epsilon_Q^-} \left( \int_{\omega \in \boldsymbol{\omega}} P(\omega) + \delta \right) = e^{\epsilon_Q^-} \Pr_{\omega \sim P} [\omega \in \boldsymbol{\omega}] + e^{\epsilon_Q^-} \delta$$

where (a) results from the definition of $Q_P^-$ and (b) from the assumption that $Q \leq_{(\varepsilon,\delta)}^- P$.

Next we denote $B_\varepsilon^a (Q,P) := \left\{ \omega \in \Omega \,|\, \ln \left( \frac{Q(\omega)}{P(\omega)} \right) > \varepsilon(\omega) + a \right\}$, and notice that for any $\omega \in B_\varepsilon^a (Q,P)$ we have $P(\omega) < e^{-(\varepsilon(\omega)+a)} Q(\omega) = e^{-\left(\epsilon_Q^- + a\right)} Q_P^- (\omega)$, so

$$\Pr_{\omega \sim P} [\omega \in B_\varepsilon^a (Q,P)] = \int_{\omega \in B_\varepsilon^a(Q,P)} P(\omega)\, d\omega$$

$$\overset{(a)}{<} \int_{\omega \in B_\varepsilon^a(Q,P)} e^{-\left(\epsilon_Q^- + a\right)} Q_P^- (\omega)\, d\omega$$

$$= e^{-\left(\epsilon_Q^- + a\right)} \Pr_{\omega \sim Q_P^-} [\omega \in B_\varepsilon^a (Q,P)]$$

$$\overset{(b)}{\leq} e^{-\left(\epsilon_Q^- + a\right)} \left( e^{\epsilon_Q^-} \Pr_{\omega \sim P} [\omega \in B_\varepsilon^a (Q,P)] + e^{\epsilon_Q^-} \delta \right)$$

$$= e^{-a} \left( \Pr_{\omega \sim P} [\omega \in B_\varepsilon^a (Q,P)] + \delta \right)$$

where (a) results from the fact the inequality holds for any element $\omega \in \Omega$ and (b) from the previous bound.

Reordering the terms we get

$$\Pr_{\omega \sim P} \left[ \ln \left( \frac{Q(\omega)}{P(\omega)} \right) > \varepsilon(\omega) + a \right] = \Pr_{\omega \sim P} [\omega \in B_\varepsilon^a (Q,P)] < \frac{\delta}{e^a - 1}.$$

$\square$

Next we prove an important connection between the two dissimilarity measures, starting with a supporting lemma which is a slight variation of Theorem 2.7 in Victor et al. [2007]. This technique is sometimes referred to as the *method of mixtures*.

**Lemma B.11.** *Given two jointly distributed random variables* $(A, B) \sim D$, *if* $\mathop{\mathbb{E}}_{(A,B)\sim D}\left[\exp\left(\lambda A - \frac{\lambda^2 B^2}{2}\right)\right] \leq 1$ *for all* $\lambda \in \mathbb{R}$, *then for any* $\delta, b > 0$,

$$\Pr_{(A,B)\sim D}\left[|A| > 2\sqrt{\ln\left(\frac{1}{\delta}\right)(B^2 + b^2)}\right] \leq \delta\sqrt{\mathop{\mathbb{E}}_{A,B\sim D}\left[\sqrt{\frac{B^2}{b^2} + 1}\right]}.$$

As Victor et al. [2007] note, at first glance this claim might seem trivial, and one can expect to prove it by simply optimizing over $\lambda$ to show $\mathop{\mathbb{E}}_{A,B,\sim D}\left[\exp\left(\frac{A}{2B^2}\right)\right] \leq 1$ and combining with Markov's inequality. The problem is, the optimal $\lambda$ depends on the value of $B$ which is a random variables, while $\lambda$ must be set beforehand. To solve this, we define a distribution over $\lambda$ and use Fubini's theorem.

*Proof.* Consider a Gaussian distribution over $\lambda$ with parameters $\mu = 0$, $\sigma = \frac{1}{b}$.

$$1 \overset{(a)}{\geq} \mathop{\mathbb{E}}_{\lambda \sim \mathcal{N}\left(0, \frac{1}{b^2}\right)}\left[\mathop{\mathbb{E}}_{A,B,\sim D}\left[\exp\left(\lambda A - \frac{\lambda^2 B^2}{2}\right)\right]\right]$$

$$\overset{(b)}{=} \mathop{\mathbb{E}}_{A,B,\sim D}\left[\mathop{\mathbb{E}}_{\lambda \sim \mathcal{N}\left(0, \frac{1}{b^2}\right)}\left[\exp\left(\lambda A - \frac{\lambda^2 B^2}{2}\right)\right]\right]$$

$$= \mathop{\mathbb{E}}_{A,B,\sim D}\left[\frac{b}{\sqrt{2\pi}}\int_{-\infty}^{\infty}\exp\left(\lambda A - \frac{\lambda^2 B^2}{2}\right)\exp\left(-\frac{\lambda^2 b^2}{2}\right)d\lambda\right]$$

$$= \mathop{\mathbb{E}}_{A,B,\sim D}\left[\frac{b}{\sqrt{2\pi}}\int_{-\infty}^{\infty}\exp\left(-\frac{B^2 + b^2}{2}\left(\lambda^2 - 2\lambda\frac{A}{B^2 + b^2}\right)\right)d\lambda\right]$$

$$\overset{(c)}{=} \mathop{\mathbb{E}}_{A,B,\sim D}\left[\frac{b}{\sqrt{B^2 + b^2}}\exp\left(\frac{A^2}{2(B^2 + b^2)}\right)\frac{\sqrt{B^2 + b^2}}{\sqrt{2\pi}}\int_{-\infty}^{\infty}\exp\left(-\frac{B^2 + b^2}{2}\left(\lambda - \frac{A}{B^2 + b^2}\right)^2\right)d\lambda\right]$$

$$\overset{(d)}{=} \mathop{\mathbb{E}}_{A,B,\sim D}\left[\frac{b}{\sqrt{B^2 + b^2}}\exp\left(\frac{A^2}{2(B^2 + b^2)}\right)\underbrace{\mathop{\mathbb{E}}_{\lambda \sim \mathcal{N}\left(\frac{A}{B^2 + b^2}, \frac{1}{B^2 + b^2}\right)}[1]}\right]$$

$$= \mathop{\mathbb{E}}_{A,B,\sim D}\left[\frac{b}{\sqrt{B^2 + b^2}}\exp\left(\frac{A^2}{2(B^2 + b^2)}\right)\right].$$

where (a) results from the fact the inequality $\mathop{\mathbb{E}}_{A,B,\sim D}\left[\exp\left(\lambda A - \frac{\lambda^2 B^2}{2}\right)\right] \leq 1$ holds for all $\lambda$, which implies it holds for the expectation taken with respect to any distribution over $\lambda$ as well, (b) results from Fubini's theorem, (c) is simply completing the square, and (d) from the definition of the Gaussian distribution.

Using this bound we get

$$
\Pr_{A,B,\sim D}\left[|A| > 2\sqrt{\ln\left(\frac{1}{\delta}\right)(B^2+b^2)}\right]
$$

$$
= \Pr_{A,B,\sim D}\left[\exp\left(\frac{A^2}{4(B^2+b^2)}\right) > \frac{1}{\delta}\right]
$$

$$
\overset{(a)}{\leq} \delta\, \mathbb{E}_{A,B,\sim D}\left[\exp\left(\frac{A^2}{4(B^2+b^2)}\right)\right]
$$

$$
= \delta\, \mathbb{E}_{A,B,\sim D}\left[\sqrt{\frac{\sqrt{B^2+b^2}}{b}}\sqrt{\frac{b}{\sqrt{B^2+b^2}}}\exp\left(\frac{A^2}{4(B^2+b^2)}\right)\right]
$$

$$
\overset{(b)}{\leq} \delta\sqrt{\mathbb{E}_{A,B,\sim D}\left[\frac{\sqrt{B^2+b^2}}{b}\right]\mathbb{E}_{A,B,\sim D}\left[\frac{b}{\sqrt{B^2+b^2}}\exp\left(\frac{A^2}{2(B^2+b^2)}\right)\right]}
$$

$$
\overset{(c)}{\leq} \delta\sqrt{\mathbb{E}_{A,B,\sim D}\left[\sqrt{\frac{B^2}{b^2}+1}\right]},
$$

where (a) results from Markov's inequality, (b) from the Cauchy-Schwarz inequality, and (c) from the previous inequality. $\qquad\square$

We can now prove the implication Theorem.

**Theorem B.12.** *Given a non-negative function $\varphi : \Omega \to \mathbb{R}^+$ and two distributions $P, Q$, if the $\alpha$-Rényi dissimilarity between $P$ and $Q$ is bounded by $\alpha\varphi$ for any $\alpha \geq 0$, then*

$$
\Pr_{\omega\sim P}\left[\left|\ln\left(\frac{P(\omega)}{Q(\omega)}\right)\right| > \varepsilon(\omega)\right] \leq \delta\sqrt{\mathbb{E}_{\omega\sim P}\left[\sqrt{\frac{\varphi(\omega)}{\phi}+1}\right]} \textit{ for any } \phi, \delta > 0, \textit{ where } \varepsilon(\omega) := \varphi(\omega) + 2\sqrt{2\ln\left(\frac{1}{\delta}\right)(\varphi(\omega)+\phi)}.
$$

*Proof.* Denoting $A := \ln\left(\frac{P(\omega)}{Q(\omega)}\right) - \varphi(\omega)$ and $B := \sqrt{2\varphi(\omega)}$, for any $\lambda \geq 0$ we denote $\alpha = \lambda + 1$ and from the assumption we get that

$$
\mathbb{E}_{A,B\sim D}\left[\exp\left(\lambda A - \frac{\lambda^2 B^2}{2}\right)\right] \overset{(a)}{\leq} \mathbb{E}_{\omega\sim P}\left[\exp\left((\alpha-1)\left(\ln\left(\frac{P(\omega)}{Q(\omega)}\right) - \alpha\varphi(\omega)\right)\right)\right] \overset{(b)}{\leq} 1
$$

For any $\lambda < 0$ we denote $\alpha = -\lambda$ and from the assumption we get that

$$
\mathbb{E}_{A,B\sim D}\left[\exp\left(\lambda A - \frac{\lambda^2 B^2}{2}\right)\right] \overset{(a)}{\leq} \mathbb{E}_{\omega\sim P}\left[\exp\left(\lambda\left(\ln\left(\frac{P(\omega)}{Q(\omega)}\right) - (\lambda+1)\varphi(\omega)\right)\right)\right]
$$

$$
\overset{(c)}{=} \mathbb{E}_{\omega\sim P}\left[\exp\left(\alpha\left(\ln\left(\frac{Q(\omega)}{P(\omega)}\right) - (\alpha-1)\varphi(\omega)\right)\right)\right]
$$

$$
= \mathbb{E}_{\omega\sim P}\left[\left(\frac{Q(\omega)}{P(\omega)}\right)^{\alpha}\exp\left(-\alpha(\alpha-1)\varphi(\omega)\right)\right]
$$

$$
\overset{(d)}{=} \mathbb{E}_{\omega\sim Q}\left[\left(\frac{Q(\omega)}{P(\omega)}\right)^{\alpha-1}\exp\left(-\alpha(\alpha-1)\varphi(\omega)\right)\right]
$$

$$
= \mathbb{E}_{\omega\sim Q}\left[\exp\left((\alpha-1)\left(\ln\left(\frac{Q(\omega)}{P(\omega)}\right) - \alpha\varphi(\omega)\right)\right)\right]
$$

$$
\overset{(b)}{\leq} 1
$$

where (a) results from the definition of $\lambda$, (b) from the Rényi dissimilarity assumption, (c) from the fact $\ln\left(\frac{a}{b}\right) = -\ln\left(\frac{b}{a}\right)$, and (d) is a renormalization step.

Combining the two with Lemma B.11 and setting $\phi := 2b^2$ we get

$$\Pr_{\omega \sim P}\left[\left|\ln\left(\frac{P(\omega)}{Q(\omega)}\right)\right| > \varepsilon(\omega)\right] = \Pr_{\omega \sim P}\left[\left|\ln\left(\frac{P(\omega)}{Q(\omega)}\right)\right| > \varphi(\omega) + 2\sqrt{2\ln\left(\frac{1}{\delta}\right)(\varphi(\omega) + \phi)}\right]$$

$$\stackrel{(a)}{\leq} \Pr_{\omega \sim P}\left[\left|\ln\left(\frac{P(\omega)}{Q(\omega)}\right) - \varphi(\omega)\right| > +2\sqrt{2\ln\left(\frac{1}{\delta}\right)(\varphi(\omega) + \phi)}\right]$$

$$= \Pr_{(A,B) \sim D}\left[|A| > 2\sqrt{\ln\left(\frac{1}{\delta}\right)(B^2 + b^2)}\right]$$

$$\stackrel{(b)}{\leq} \delta\sqrt{\mathbb{E}_{A,B \sim D}\left[\sqrt{\frac{B^2}{b^2} + 1}\right]}$$

$$= \delta\sqrt{\mathbb{E}_{\omega \sim P}\left[\sqrt{\frac{\varphi(\omega)}{\phi} + 1}\right]},$$

where (a) results from the triangle inequality and (b) from Lemma B.11. $\qquad\square$

## B.3 Convexity

Next we prove some convexity results for max and Rényi dissimilarities.

**Theorem B.13** (Max dissimilarity convexity). *Given $\lambda \in (0,1)$, $\delta_i \geq 0$, distributions $P_i, Q_i$, and non-negative functions $\varepsilon_i : \omega \to \mathbb{R}^+$ for $i \in \{0,1\}$, denote $P_\lambda := \lambda P_0 + (1 - \lambda) P_1$, $Q_\lambda := \lambda Q_0 + (1 - \lambda) Q_1$, $\varepsilon_\lambda(\omega) := \ln\left(\lambda e^{\varepsilon_0(\omega)} + (1 - \lambda) e^{\varepsilon_1(\omega)}\right)$, and $\delta_\lambda := \lambda \delta_0 + (1 - \lambda) \delta_1$.*

1. *Joint quasi-convexity: If $P_i \leq^+_{(\varepsilon_i, \delta_i)} Q_i$ for $i \in \{0, 1\}$, then $P_\lambda \leq^+_{(\varepsilon_{\max}, \delta_\lambda)} Q_\lambda$, where $\varepsilon_{\max}(\omega) := \max\{\varepsilon_0(\omega), \varepsilon_1(\omega)\}$*

2. *Left-hand log-convexity: If $P_i \leq^+_{(\varepsilon_i, \delta_i)} Q$ for $i \in \{0, 1\}$, then $P_\lambda \leq^+_{(\varepsilon_\lambda, \delta_\lambda)} Q$*

3. *Right-hand log-convexity: If $P \leq^-_{(\varepsilon_i, \delta_i)} Q_i$ for $i \in \{0, 1\}$, then $P \leq^-_{(\varepsilon_\lambda, \delta_\lambda)} Q_\lambda$*

*Proof.* **Part 1**: For any $\boldsymbol{\omega} \subseteq \Omega$,

$$\int_{\omega \in \boldsymbol{\omega}} P_\lambda(\omega)\, d\omega \stackrel{(a)}{=} \int_{\omega \in \boldsymbol{\omega}} (\lambda P_0(\omega) + (1 - \lambda) P_1(\omega))\, d\omega$$

$$= \lambda \int_{\omega \in \boldsymbol{\omega}} P_0(\omega)\, d\omega + (1 - \lambda) \int_{\omega \in \boldsymbol{\omega}} P_1(\omega)\, d\omega$$

$$\stackrel{(b)}{\leq} \lambda\left(\int_{\omega \in \boldsymbol{\omega}} e^{\varepsilon_0(\omega)} Q_0(\omega)\, d\omega + \delta_0\right) + (1 - \lambda)\left(\int_{\omega \in \boldsymbol{\omega}} e^{\varepsilon_1(\omega)} Q_1(\omega)\, d\omega + \delta_1\right)$$

$$\stackrel{(c)}{\leq} \lambda\left(\int_{\omega \in \boldsymbol{\omega}} e^{\varepsilon_{\max}(\omega)} Q_0(\omega)\, d\omega + \delta_0\right) + (1 - \lambda)\left(\int_{\omega \in \boldsymbol{\omega}} e^{\varepsilon_{\max}(\omega)} Q_1(\omega)\, d\omega + \delta_1\right)$$

$$\stackrel{(d)}{=} \int_{\omega \in \boldsymbol{\omega}} e^{\varepsilon_{\max}(\omega)} Q_\lambda(\omega)\, d\omega + \delta_\lambda$$

where (a) results from the definition of $P_\lambda$, (b) from the assumption $P_i \leq^+_{(\varepsilon_i, \delta_i)} Q_i$, (c) from the definitions of $\varepsilon_{\max}$ and $\delta_\lambda$, and (d) from the definition of $Q_\lambda$.

**Part 2**: For any $\boldsymbol{\omega} \subseteq \Omega$,

$$
\int_{\omega \in \boldsymbol{\omega}} P_\lambda (\omega) \, d\omega \stackrel{(a)}{=} \int_{\omega \in \boldsymbol{\omega}} \left( \lambda P_0 (\omega) + (1 - \lambda) P_1 (\omega) \right) d\omega
$$

$$
= \lambda \int_{\omega \in \boldsymbol{\omega}} P_0 (\omega) \, d\omega + (1 - \lambda) \int_{\omega \in \boldsymbol{\omega}} P_1 (\omega) \, d\omega
$$

$$
\stackrel{(b)}{\leq} \lambda \left( \int_{\omega \in \boldsymbol{\omega}} e^{\varepsilon_0 (\omega)} Q (\omega) \, d\omega + \delta_0 \right) + (1 - \lambda) \left( \int_{\omega \in \boldsymbol{\omega}} e^{\varepsilon_1 (\omega)} Q (\omega) \, d\omega + \delta_1 \right)
$$

$$
\stackrel{(c)}{=} \int_{\omega \in \boldsymbol{\omega}} \left( \lambda e^{\varepsilon_0 (\omega)} + (1 - \lambda) e^{\varepsilon_1 (\omega)} \right) Q (\omega) \, d\omega + \delta_\lambda
$$

$$
\stackrel{(d)}{=} \int_{\omega \in \boldsymbol{\omega}} e^{\varepsilon_\lambda (\omega)} Q (\omega) \, d\omega + \delta_\lambda
$$

where (a) results from the definition of $P_\lambda$, (b) from the assumption $P_i \leq^+_{(\varepsilon_i, \delta_i)} Q$, (c) from the definition of $\delta_\lambda$, and (d) from the definition of $\varepsilon_\lambda$.

**Part 3**: For any $\boldsymbol{\omega} \subseteq \Omega$,

$$
\int_{\omega \in \boldsymbol{\omega}} e^{-\varepsilon_\lambda (\omega)} P (\omega) \, d\omega \stackrel{(a)}{=} \int_{\omega \in \boldsymbol{\omega}} \frac{1}{\lambda e^{\varepsilon_0 (\omega)} + (1 - \lambda) e^{\varepsilon_1 (\omega)}} P (\omega) \, d\omega
$$

$$
\stackrel{(b)}{\leq} \int_{\omega \in \boldsymbol{\omega}} \left( \lambda e^{-\varepsilon_0 (\omega)} + (1 - \lambda) e^{-\varepsilon_1 (\omega)} \right) P (\omega) \, d\omega
$$

$$
= \lambda \left( \int_{\omega \in \boldsymbol{\omega}} e^{-\varepsilon_0 (\omega)} P (\omega) \, d\omega \right) + (1 - \lambda) \left( \int_{\omega \in \boldsymbol{\omega}} e^{-\varepsilon_1 (\omega)} P (\omega) \, d\omega \right)
$$

$$
\stackrel{(c)}{\leq} \lambda \left( \int_{\omega \in \boldsymbol{\omega}} Q_0 (\omega) \, d\omega + \delta_0 \right) + (1 - \lambda) \left( \int_{\omega \in \boldsymbol{\omega}} Q_1 (\omega) \, d\omega + \delta_1 \right)
$$

$$
\stackrel{(d)}{=} \int_{\omega \in \boldsymbol{\omega}} Q_\lambda (\omega) \, d\omega + \delta_\lambda
$$

where (a) results from the definition of $\varepsilon_\lambda$, (b) from Jensen's inequality for the convex function $\frac{1}{x}$, (c) from the assumption $P \leq^-_{(\varepsilon_i, \delta_i)} Q_i$, and (d) from the definitions of $Q_\lambda$ and $\delta_\lambda$. $\qquad \square$

**Theorem B.14** (Rényi dissimilarity convexity). *Given $\alpha \geq 1$, $\lambda \in (0, 1)$, and non-negative functions $\varphi_i : \omega \to \mathbb{R}^+$ for $i \in \{0, 1\}$, denote $P_\lambda := \lambda P_0 + (1 - \lambda) P_1$, $Q_\lambda := \lambda Q_0 + (1 - \lambda) Q_1$, and $\varphi_\lambda (\omega) := \lambda \varphi_0 (\omega) + (1 - \lambda) \varphi_1 (\omega)$.*

1. *Joint quasi-convexity: If the $\alpha$-Rényi dissimilarity between $P_i$ and $Q_i$ is bounded by $\varphi_i$ for $i \in \{0, 1\}$, then the $\alpha$-Rényi dissimilarity between $P_\lambda$ and $Q_\lambda$ is bounded by $\varphi_{\max}$, where $\varphi_{\max} (\omega) := \max \{ \varphi_0 (\omega), \varphi_1 (\omega) \}$*

2. *Right-hand convexity: If the $\alpha$-Rényi dissimilarity between $P$ and $Q_i$ is bounded by $\varphi_i$ for $i \in \{0, 1\}$, then the $\alpha$-Rényi dissimilarity between $P$ and $Q_\lambda$ is bounded by $\varphi_\lambda$*

*Proof.* The case of $\alpha = 1$ is trivial for both parts, so we focus on the $\alpha > 1$ case.

**Part 1**: We first notice that

$$P_\lambda(\omega) = \lambda P_0(\omega) + (1 - \lambda) P_1(\omega)$$

$$= \lambda (Q_0(\omega))^{\frac{\alpha-1}{\alpha}} \frac{P_0(\omega)}{(Q_0(\omega))^{\frac{\alpha-1}{\alpha}}} + (1 - \lambda)(Q_1(\omega))^{\frac{\alpha-1}{\alpha}} \frac{P_1(\omega)}{(Q_1(\omega))^{\frac{\alpha-1}{\alpha}}}$$

$$\overset{(a)}{\leq} \left( \lambda \left( (Q_0(\omega))^{\frac{\alpha-1}{\alpha}} \right)^{\frac{\alpha}{\alpha-1}} + (1 - \lambda) \left( (Q_1(\omega))^{\frac{\alpha-1}{\alpha}} \right)^{\frac{\alpha}{\alpha-1}} \right)^{\frac{\alpha-1}{\alpha}}$$

$$\cdot \left( \lambda \left( \frac{P_0(\omega)}{(Q_0(\omega))^{\frac{\alpha-1}{\alpha}}} \right)^\alpha + (1 - \lambda) \left( \frac{P_1(\omega)}{(Q_1(\omega))^{\frac{\alpha-1}{\alpha}}} \right)^\alpha \right)^{\frac{1}{\alpha}}$$

$$= (\lambda Q_0(\omega) + (1 - \lambda) Q_1(\omega))^{\frac{\alpha-1}{\alpha}} \cdot \left( \lambda \frac{(P_0(\omega))^\alpha}{(Q_0(\omega))^{\alpha-1}} + (1 - \lambda) \frac{(P_1(\omega))^\alpha}{(Q_1(\omega))^{\alpha-1}} \right)^{\frac{1}{\alpha}}$$

$$= Q_\lambda(\omega) \cdot \left( \lambda \frac{P_0(\omega)}{Q_\lambda(\omega)} \left( \frac{P_0(\omega)}{Q_0(\omega)} \right)^{\alpha-1} + (1 - \lambda) \frac{P_1(\omega)}{Q_\lambda(\omega)} \left( \frac{P_1(\omega)}{Q_1(\omega)} \right)^{\alpha-1} \right)^{\frac{1}{\alpha}}$$

where (a) results from the generalized version of Hölder's inequality, $\mathbb{E}[X \cdot Y] \leq \left( \mathbb{E}[X^p] \right)^{\frac{1}{p}} \cdot \left( \mathbb{E}[Y^q] \right)^{\frac{1}{q}}$ for any $p, q > 0$ such that $\frac{1}{p} + \frac{1}{q} = 1$.

Using this bound we get

$$\underset{\omega \sim P_\lambda}{\mathbb{E}} \left[ \exp\left( (\alpha - 1) \left( \ln\left( \frac{P_\lambda(\omega)}{Q_\lambda(\omega)} \right) - \varphi_{\max}(\omega) \right) \right) \right]$$

$$= \underset{\omega \sim P_\lambda}{\mathbb{E}} \left[ \left( \frac{P_\lambda(\omega)}{Q_\lambda(\omega)} \right)^{\alpha-1} \exp\left( -(\alpha - 1)\varphi_{\max}(\omega) \right) \right]$$

$$= \underset{\omega \sim Q_\lambda}{\mathbb{E}} \left[ \left( \frac{P_\lambda(\omega)}{Q_\lambda(\omega)} \right)^{\alpha} \exp\left( -(\alpha - 1)\varphi_{\max}(\omega) \right) \right]$$

$$\overset{(a)}{\leq} \underset{\omega \sim Q_\lambda}{\mathbb{E}} \left[ \left( \lambda \frac{P_0(\omega)}{Q_\lambda(\omega)} \left( \frac{P_0(\omega)}{Q_0(\omega)} \right)^{\alpha-1} + (1 - \lambda) \frac{P_1(\omega)}{Q_\lambda(\omega)} \left( \frac{P_1(\omega)}{Q_1(\omega)} \right)^{\alpha-1} \right) \exp\left( -(\alpha - 1)\varphi_{\max}(\omega) \right) \right]$$

$$= \lambda \underset{\omega \sim Q_\lambda}{\mathbb{E}} \left[ \frac{P_0(\omega)}{Q_\lambda(\omega)} \left( \frac{P_0(\omega)}{Q_0(\omega)} \right)^{\alpha-1} \exp\left( -(\alpha - 1)\varphi_{\max}(\omega) \right) \right]$$

$$+ (1 - \lambda) \underset{\omega \sim Q_\lambda}{\mathbb{E}} \left[ \frac{P_1(\omega)}{Q_\lambda(\omega)} \left( \frac{P_1(\omega)}{Q_1(\omega)} \right)^{\alpha-1} \exp\left( -(\alpha - 1)\varphi_{\max}(\omega) \right) \right]$$

$$\overset{(b)}{=} \lambda \underset{\omega \sim P_0}{\mathbb{E}} \left[ \exp\left( (\alpha - 1) \left( \ln\left( \frac{P_0(\omega)}{Q_0(\omega)} \right) - \varphi_{\max}(\omega) \right) \right) \right]$$

$$+ (1 - \lambda) \underset{\omega \sim P_1}{\mathbb{E}} \left[ \exp\left( (\alpha - 1) \left( \ln\left( \frac{P_1(\omega)}{Q_1(\omega)} \right) - \varphi_{\max}(\omega) \right) \right) \right]$$

$$\overset{(c)}{\leq} \lambda \underset{\omega \sim P_0}{\mathbb{E}} \left[ \exp\left( (\alpha - 1) \left( \ln\left( \frac{P_0(\omega)}{Q_0(\omega)} \right) - \varphi_0(\omega) \right) \right) \right]$$

$$+ (1 - \lambda) \underset{\omega \sim P_1}{\mathbb{E}} \left[ \exp\left( (\alpha - 1) \left( \ln\left( \frac{P_1(\omega)}{Q_1(\omega)} \right) - \varphi_1(\omega) \right) \right) \right]$$

$$\overset{(d)}{\leq} 1$$

where (a) results from the previous inequality, (b) is a renormalization step, (c) from the the definition of $\varphi_{\max}$, and (d) from the Rényi dissimilarity assumption.

**Part 2**:

$$\mathop{\mathbb{E}}_{\omega \sim P}\left[\exp\left((\alpha - 1)\left(\ln\left(\frac{P(\omega)}{Q_\lambda(\omega)}\right) - \varphi_\lambda(\omega)\right)\right)\right]$$

$$= \mathop{\mathbb{E}}_{\omega \sim P}\left[\exp\left((\alpha - 1)\left(\ln\left(\frac{P(\omega)}{\lambda Q_0(\omega) + (1-\lambda)Q_1(\omega)}\right) - \varphi_\lambda(\omega)\right)\right)\right]$$

$$\stackrel{(a)}{\leq} \mathop{\mathbb{E}}_{\omega \sim P}\left[\exp\left((\alpha - 1)\left(\lambda\ln\left(\frac{P(\omega)}{Q_0(\omega)}\right) + (1-\lambda)\ln\left(\frac{P(\omega)}{Q_1(\omega)}\right) - \varphi_\lambda(\omega)\right)\right)\right]$$

$$\stackrel{(b)}{=} \mathop{\mathbb{E}}_{\omega \sim P}\left[\exp\left(\lambda(\alpha - 1)\left(\ln\left(\frac{P(\omega)}{Q_0(\omega)}\right) - \varphi_0(\omega)\right) + (1-\lambda)(\alpha - 1)\left(\ln\left(\frac{P(\omega)}{Q_1(\omega)}\right) - \varphi_1(\omega)\right)\right)\right]$$

$$\stackrel{(c)}{\leq} \lambda \mathop{\mathbb{E}}_{\omega \sim P}\left[\exp\left((\alpha - 1)\left(\ln\left(\frac{P(\omega)}{Q_0(\omega)}\right) - \varphi_0(\omega)\right)\right)\right]$$

$$+ (1-\lambda)\mathop{\mathbb{E}}_{\omega \sim P}\left[\exp\left((\alpha - 1)\left(\ln\left(\frac{P(\omega)}{Q_1(\omega)}\right) - \varphi_1(\omega)\right)\right)\right]$$

$$\stackrel{(d)}{\leq} 1$$

where (a) results from Jensen's inequality for the convex function $\ln\left(\frac{1}{x}\right)$, (b) from the definition of $\varphi_\lambda$, (c) from Jensen's inequality for the convex function $e^x$ over the $\lambda$ weighted combination, and (d) from the Rényi dissimilarity assumption. $\qquad\square$

## B.4 Stability measures

The various divergence and dissimilarity measures considered in the previous sections were used to define several stability notions; the divergence (dissimilarity) between the distributions over responses induced by two differing input datasets serves as a measure of the stability of the mechanism producing those responses.

One major application of stability notions is in privacy-preserving mechanisms.

**Definition B.15** (Differential privacy [Dwork et al., 2006]). Given $\epsilon, \delta \geq 0$, a mechanism $M$ will be called $(\epsilon, \delta)$-*differentially private* (or DP, for short) if for any two datasets $s, s' \in \mathcal{X}^n$ that differ only in one element, and any query $q \in \mathcal{Q}$, the $\delta$-approximate max divergence between the two distributions defined over $\mathcal{R}$ by $M(s, q)$ and $M(s', q)$ is bounded by $\epsilon$.

As in Definition 2.1, $\delta$ can be viewed as a function of $\epsilon$. In this case, $\delta$ is essentially a tail bound on the distribution of the privacy loss random variable (Definition 4.1). An alternative stability notion can be based on a bound on the moments of the stability loss.

**Definition B.16** (Rényi differential privacy [Mironov, 2017]). Given $\alpha, \phi \geq 0$, a mechanism $M$ will be called $(\alpha, \phi)$-*Rényi differentially private* (or RDP, for short) if for any two datasets $s, s' \in \mathcal{X}^n$ that differ only in one element, and any query $q \in \mathcal{Q}$, the $\alpha$-Rényi divergence between the two distributions defined over $\mathcal{R}$ by $M(s, q)$ and $M(s', q)$ is bounded by $\phi$.

The following two definitions bound the shape of the privacy loss curve.

**Definition B.17** (Concentrated differential privacy [Dwork and Rothblum, 2016]). Given $\mu, \sigma \geq 0$, a mechanism $M$ will be called $(\mu, \sigma)$-*concentrated differentially private* (or CDP, for short) if for any two datasets $s, s' \in \mathcal{X}^n$ that differ only in one element, and any query $q \in \mathcal{Q}$, the stability loss random variable (Definition 4.1) is a $\sigma^2$-sub-Gaussian random variable (Definition 5.2) with expectation bounded by $\mu$. Formally, $\mathop{\mathbb{E}}_{R \sim M(s,q)}[\ell(s, s'; R)] \leq \mu$ and for any $\alpha \geq 1$ we have

$$\mathop{\mathbb{E}}_{R \sim M(s,q)}\left[\exp\left((\alpha - 1)\left(\ell(s, s'; R) - \mathop{\mathbb{E}}_{R' \sim M(s,q)}[\ell(s, s'; R')]\right)\right)\right] \leq e^{\frac{\alpha^2}{2\sigma^2}}.$$

**Definition B.18** (Zero-concentrated differential privacy [Bun and Steinke, 2016]). Given $\phi \geq 0$, a mechanism $M$ will be called $\phi$-*zero concentrated differentially private* (or zCDP, for short) if for any two datasets $s, s' \in \mathcal{X}^n$ that differ only in one element, any query $q \in \mathcal{Q}$, and $\alpha \geq 1$, the $\alpha$-Rényi

divergence between the two distributions defined over $\mathcal{R}$ by $M(s, q)$ and $M(s', q)$ is bounded by $\alpha\phi$ in both directions.[8]

This definition can be viewed as a special case of the CDP definition, with $\mu = \phi$ and $\sigma^2 = 2\phi$. The comparison is formalized in Lemma 24 in [Bun and Steinke, 2016].

The above notions all hold over *all* pairs of neighboring datasets, which can be achieved by scaling the randomness added by the mechanism to the worst-case sensitivity—that is, to the worst-case change in the output that can be induced by replacing one dataset by a neighboring one. As discussed in detail in Appendix F, to avoid the dependence on the range of the queries, our approach instead focuses on a version of the the "local sensitivity," which, in turn, involves transitioning from divergences to dissimilarity notions.

**Definition B.19** (Pairwise concentration; equivalent to Definition 4.2 but restated in terms of divergence and dissimilarity). Given a non-negative function $\varphi : \mathcal{X}^n \times \mathcal{X}^n \times \mathcal{Q} \to \mathbb{R}^+$ which is symmetric in its first two arguments, a mechanism $M$ will be called $\varphi$-*Pairwise Concentrated* (or PC, for short), if for any $s, s' \in \mathcal{X}^n$, query $q \in \mathcal{Q}$, and $\alpha \geq 0$, the $\alpha$-Rényi divergence between the two distributions defined over $\mathcal{R}$ by $M(s, q)$ and $M(s', q)$ is bounded by $\alpha\varphi(s, s'; q)$ in both directions.

Given a non-negative function $\varphi : \mathcal{X}^n \times \mathcal{X}^n \times \mathcal{V} \to \mathbb{R}^+$ which is symmetric in its first two arguments and an analyst $A$, a mechanism $M$ will be called $\varphi$-Pairwise Concentrated with respect to $A$, if for any $s, s' \in \mathcal{X}^n$ and $\alpha \geq 0$, the $\alpha$-Rényi dissimilarity between the two distributions defined over $\mathcal{V}$ by $M(s, A)$ and $M(s', A)$ is bounded by $\alpha\varphi(s, s'; \cdot)$ in both directions.

Analogously to how one can generalize the zCDP stability notion by replacing the divergence by a dissimilarity measure, one can also generalize other stability notions such as differential privacy.

**Definition B.20** (Pairwise indistinguishability). Given $\delta \geq 0$ and a non-negative function $\varepsilon : \mathcal{X}^n \times \mathcal{X}^n \times \mathcal{Q} \to \mathbb{R}^+$ which is symmetric in its first two arguments, a mechanism $M$ will be called $\varepsilon$-*Pairwise Indistinguishable* (or PI, for short), if for any $s, s' \in \mathcal{X}^n$, and any query $q \in \mathcal{Q}$, the max divergence between the two distributions defined over $\mathcal{R}$ by $M(s, q)$ and $M(s', q)$ is bounded by $\varepsilon(s, s'; q)$ in both directions.

Given a non-negative function $\varepsilon : \mathcal{X}^n \times \mathcal{X}^n \times \mathcal{V} \to \mathbb{R}^+$ which is symmetric in its first two arguments and an analyst $A$, a mechanism $M$ will be called $\varepsilon$-Pairwise Indistinguishable with respect to $A$, if for any $s, s' \in \mathcal{X}^n$, the max dissimilarity between the two distributions defined over $\mathcal{V}$ by $M(s, A)$ and $M(s', A)$ is bounded by $\varepsilon(s, s'; \cdot)$ in both directions.

We note that Theorem 4.6 can be stated in terms of pairwise indistinguishability as well.

---

[8]The original definition includes an additional parameter that is equal to 0 in the case of the Gaussian mechanism, so we omit it here for simplicity.

# C  Missing parts from Section 3

## C.1  Missing proofs

**Lemma C.1.** *Given a probability distribution $P$ over a product domain $\mathcal{X} \times \mathcal{Y}$ and a function $f : \mathcal{X} \times \mathcal{Y} \to \mathbb{R}$, for any $\epsilon, \xi > 0$ we have,*

$$\Pr_{X \sim P_\mathcal{X}} \left[ \left| \underset{Y \sim P_{\mathcal{Y}|\mathcal{X}}}{\mathbb{E}} [f(X,Y) \mid X] \right| > \epsilon + \xi \right] \leq \frac{1}{\xi} \int_\epsilon^\infty \Pr_{(X,Y) \sim P} [|f(X,Y)| > t] \, dt.$$

*Proof.*

$$
\begin{aligned}
\Pr_{X \sim P_\mathcal{X}} \left[ \left| \underset{Y \sim P_{\mathcal{Y}|\mathcal{X}}}{\mathbb{E}} [f(X,Y) \mid X] \right| > \epsilon + \xi \right] &\overset{(a)}{\leq} \Pr_{X \sim P_\mathcal{X}} \left[ \left[ \left| \underset{Y \sim P_{\mathcal{Y}|\mathcal{X}}}{\mathbb{E}} [f(X,Y) \mid X] \right| - \epsilon \right]^+ > \xi \right] \\
&\overset{(b)}{\leq} \frac{1}{\xi} \underset{X \sim P_\mathcal{X}}{\mathbb{E}} \left[ \left[ \left| \underset{Y \sim P_{\mathcal{Y}|\mathcal{X}}}{\mathbb{E}} [f(X,Y) \mid X] \right| - \epsilon \right]^+ \right] \\
&\overset{(c)}{\leq} \frac{1}{\xi} \underset{X \sim P_\mathcal{X}}{\mathbb{E}} \left[ \left[ \underset{Y \sim P_{\mathcal{Y}|\mathcal{X}}}{\mathbb{E}} [|f(X,Y)| \mid X] - \epsilon \right]^+ \right] \\
&\overset{(d)}{\leq} \frac{1}{\xi} \underset{X \sim P_\mathcal{X}}{\mathbb{E}} \left[ \underset{Y \sim P_{\mathcal{Y}|\mathcal{X}}}{\mathbb{E}} \left[ [|f(X,Y)| - \epsilon]^+ \mid X \right] \right] \\
&= \frac{1}{\xi} \underset{(X,Y) \sim P}{\mathbb{E}} \left[ [|f(X,Y)| - \epsilon]^+ \right] \\
&\overset{(e)}{=} \frac{1}{\xi} \int_0^\infty \Pr_{(X,Y) \sim P} \left[ [|f(X,Y)| - \epsilon]^+ > t \right] dt \\
&= \frac{1}{\xi} \int_\epsilon^\infty \Pr_{(X,Y) \sim P} [|f(X,Y)| > t] \, dt
\end{aligned}
$$

where (a) $[x]^+ := \max\{x, 0\}$ is the ReLU function, (b) results from Markov's inequality, (c) from the triangle inequality which implies $\left| \mathbb{E}[X] \right| \leq \mathbb{E}[|X|]$ and $\left[ \mathbb{E}[X] \right]^+ \leq \mathbb{E}\left[ [X]^+ \right]$, (d) from Jensen's inequality for the convex ReLU function, and (e) from the fact that the expectation is taken over a non-negative random variable. □

**Lemma C.2** (Lemma 3.1 restated). *Given a function $\delta : \mathbb{R} \to [0,1]$ and an analyst $A$, if a mechanism $M$ is $(\epsilon, \delta(\epsilon))$-sample accurate for all $\epsilon > 0$, then $M$ is $(\epsilon, \delta'(\epsilon))$-posterior accurate for $\delta'(\epsilon) := \inf_{\xi \in (0,\epsilon)} \left( \frac{1}{\xi} \int_{\epsilon-\xi}^\infty \delta(t) \, dt \right).$*

*Proof.* For any $v \in \mathcal{V}$, denote $I(v) := \arg\max_{i \in [k]} |r_i - q_i(D^v)|$, and $r_v := r_{I(v)}, q_v := q_{I(v)}$. When the view is a random variable $V$, the corresponding query $Q_V$ and response $R_V$ are also random

variables, and hence we denote them with capital letters.

$$\Pr_{V \sim D_\mathcal{V}} \left[ \max_{i \in [k]} |R_i - Q_i (D^V)| > \epsilon + \xi \right]$$

$$= \Pr_{V \sim D} \left[ |R_V - Q_V (D^V)| > \epsilon + \xi \right]$$

$$= \Pr_{V \sim D} \left[ \left| \mathbb{E}_{S \sim D(\cdot \,|\, V)} [R_V - Q_V (S)] \right| > \epsilon + \xi \right]$$

$$\overset{(a)}{\leq} \frac{1}{\xi} \int_{\epsilon - \xi}^{\infty} \Pr_{S \sim D^{(n)}, V \sim M(S,A)} [|R_V - Q_V (S)| > u] \, du$$

$$= \frac{1}{\xi} \int_{\epsilon - \xi}^{\infty} \Pr_{S \sim D^{(n)}, V \sim M(S,A)} \left[ \max_{i \in [k]} |R_i - Q_i (S)| > u \right] du$$

$$\overset{(b)}{\leq} \frac{1}{\xi} \int_{\epsilon - \xi}^{\infty} \delta (t) \, dt$$

where (a) results from part one of Lemma C.1 and (b) from the definition of sample accuracy (Definition 2.1). $\qquad\square$

**Lemma C.3** (Lemma 3.2 restated). *Given $\eta > 0$, the Gaussian mechanism with noise parameter $\eta$ that receives $k$ queries is $(\epsilon, \delta(\epsilon))$-sample accurate for $\delta(\epsilon) := \frac{2k}{\sqrt{\pi}} e^{-\frac{\epsilon^2}{2\eta^2}}$, and $(\epsilon, \delta(\epsilon))$-posterior accurate for $\delta(\epsilon) := 4k \cdot e^{-\frac{\epsilon^2}{4\eta^2}}$.*

*Proof.* In the case of a Gaussian distribution, $\delta(\epsilon)$ is known as the Q-function and for any dataset $s$ and query $q$,

$$\Pr_{R \sim M(s,q)} [|R - q(s)| > \epsilon] = \mathrm{erfc} \left( \frac{\epsilon}{\sqrt{2}\eta} \right) \leq \frac{2}{\sqrt{\pi}} e^{-\frac{\epsilon^2}{2\eta^2}}.$$

Combining this with a union bound completes the proof of sample accuracy.

Invoking Lemma 3.1 with $\xi = \epsilon - \sqrt{\epsilon^2 - 2\eta^2}$ implies for all $\epsilon \geq \sqrt{2}\eta$ and any $i \in [k]$,

$$\Pr_{S \sim D^{(n)}, V \sim M(S,A)} [\mathrm{err}_P (S, V, i) > \epsilon] \leq \frac{1}{\xi} \int_{\epsilon - \xi}^{\infty} \mathrm{erfc} \left( \frac{t}{\sqrt{2}\eta} \right) dt$$

$$\overset{\left(u := \frac{t}{\sqrt{2}\eta}\right)}{=} \frac{\sqrt{2}\eta}{\xi} \int_{\frac{\epsilon - \xi}{\sqrt{2}\eta}}^{\infty} \mathrm{erfc} (u) \, du$$

$$\overset{(a)}{<} \frac{\sqrt{2}\eta \cdot e^{-\frac{\epsilon^2 - 2\eta^2}{2\eta^2}}}{\sqrt{\pi}\epsilon \left( 1 - \sqrt{1 - \frac{2\eta^2}{\epsilon^2}} \right)}$$

$$\overset{(b)}{\leq} \frac{\sqrt{2}e\epsilon}{\sqrt{\pi}\eta} e^{-\frac{\epsilon^2}{2\eta^2}}$$

$$\overset{(c)}{\leq} \frac{2e}{\sqrt{\pi}} e^{-\frac{\epsilon^2}{4\eta^2}}$$

$$\leq 4e^{-\frac{\epsilon^2}{4\eta^2}}$$

where (a) results from the fact that $\int_a^{\infty} \mathrm{erfc} (x) \, dx = \frac{e^{-a^2}}{\sqrt{\pi}} - a \cdot \mathrm{erfc} (a) < \frac{e^{-a^2}}{\sqrt{\pi}}$ and the definition of $\xi$, (b) from the bound on $\epsilon$ and the inequality $\sqrt{1 + x} \leq 1 + \frac{x}{2}$, and (c) from the inequality $x \cdot e^{-x^2} \leq e^{-\frac{x^2}{2}}$.

Combining this with a union bound completes the proof of posterior accuracy. $\qquad\square$

**Theorem C.4** (Theorem 3.4 restated). *Given two functions* $\delta_1 : \mathbb{R} \to [0,1]$, $\delta_2 : \mathbb{R} \to [0,1]$, *and an analyst A, if a mechanism M is* $(\epsilon, \delta_1(\epsilon))$-*Bayes stable and* $(\epsilon, \delta_2(\epsilon))$-*sample accurate with respect to A, then M is* $(\epsilon, \delta'(\epsilon))$-*distribution accurate for* $\delta'(\epsilon) \coloneqq$
$$\inf_{\epsilon' \in (0,\epsilon), \xi \in (0, \epsilon - \epsilon')} \left( \delta_1(\epsilon') + \frac{1}{\xi} \int_{\epsilon - \epsilon' - \xi}^{\infty} \delta_2(t)\, dt \right).$$

*Proof.* Consider an analyst $A$, which once completed an interaction of length $k$ resulting in a view $v$, chooses $q_{k+1} \coloneqq \arg\max_{i \in [k]} |q_i(D^v) - q_i(D)|$. In this case,

$$\Pr_{\substack{S \sim D^{(n)} \\ V \sim M(S,A)}} \left[ \max_{i \in [k]} |R_i - Q_i(D)| > \epsilon \right]$$

$$\leq \Pr_{\substack{S \sim D^{(n)} \\ V \sim M(S,A)}} \left[ \max_{i \in [k]} |R_i - Q_i(D^V)| + \max_{i \in [k]} |Q_i(D^V) - Q_i(D)| > \epsilon \right]$$

$$\leq \Pr_{\substack{S \sim D^{(n)} \\ V \sim M(S,A)}} \left[ \max_{i \in [k]} |R_i - Q_i(D^V)| > \epsilon - \epsilon' \right] + \Pr_{\substack{S \sim D^{(n)} \\ V \sim M(S,A), Q \sim A(V)}} \left[ |Q(D^V) - Q(D)| > \epsilon' \right]$$

$$\overset{(a)}{\leq} \frac{1}{\xi} \int_{\epsilon - \epsilon' - \xi}^{\infty} \delta(t)\, dt + \delta(\epsilon'),$$

where (a) results from Lemma 3.1 and the definition of Bayes stability (Definition 3.3). $\qquad\square$

## C.2  Generalized results for arbitrary queries

While the main results in this paper were stated for linear queries, some of the intermediate results extend to arbitrary queries. Denoting by $\bar{\Delta}_q \coloneqq \sup_{s,s' \in \mathcal{X}} |q(s) - q(s)|$ and $\bar{\sigma}_q^2 \coloneqq$ $\mathbb{E}_{X \sim D_{\mathcal{X}}} \left[ (q(X) - q(D_{\mathcal{X}^n}))^2 \right]$ and extending the posterior distribution notations to $D_{\mathcal{X}^n}^v \coloneqq D_{\mathcal{X}^n | V}^A(\cdot \mid v)$, Lemma 3.5 can be restated in terms of arbitrary queries.

**Lemma C.5** (Covariance stability for arbitrary queries). *Given a view* $v \in \mathcal{V}$ *and a query* $q$,
$$q(D_{\mathcal{X}^n}^v) - q(D_{\mathcal{X}^n}) = \underset{S \sim D_{\mathcal{X}^n}}{Cov}(q(S), K(S,v)).$$

*Furthermore, given* $\Delta, \sigma > 0$,
$$\sup_{q \in \mathcal{Q} \text{ s.t. } \bar{\Delta}_q \leq \Delta} |q(D_{\mathcal{X}^n}^v) - q(D_{\mathcal{X}^n})| = 2\Delta \cdot \mathbf{D}_{TV}(D_{\mathcal{X}^n}^v \| D_{\mathcal{X}^n})$$

*and*
$$\sup_{q \in \mathcal{Q} \text{ s.t. } \bar{\sigma}_q^2 \leq \sigma^2} |q(D_{\mathcal{X}}^v) - q(D_{\mathcal{X}})| = \sigma \sqrt{\mathbf{D}_{\chi^2}(D_{\mathcal{X}^n}^v \| D_{\mathcal{X}^n})},$$

*where* $K(s,v) \coloneqq \frac{D(s \mid v)}{D(s)} = \frac{D(v \mid s)}{D(v)}$ *is the Bayes factor of* $s$ *given* $v$ *(and vice-versa).*

Combining this lemma with the PC definition for arbitrary sample sets, we can state a simple version of Theorem 4.6 for arbitrary queries, and achieve results similar to those guaranteed using typical stability [Bassily and Freund, 2016]. Like in the case of typical stability, this method does not achieve optimal accuracy rates for the iid case as discussed by Kontorovich et al. [2022].

# D Missing parts from Section 4

## D.1 PC mechanisms

**Lemma D.1** (Lemma 4.3 restated). *Given $\eta > 0$ and an analyst A, if $M$ is a Gaussian mechanism with noise parameter $\eta$, then it is $\varphi$-PC for $\varphi\left(s, s'; q\right) := \frac{\left(q(s) - q\left(s'\right)\right)^2}{2\eta^2}$ and $\varphi$-PC with respect to A for $\varphi\left(s, s'; v\right) := \frac{\left\|\bar{q}_v(s) - \bar{q}_v\left(s'\right)\right\|^2}{2\eta^2}$.*

*Proof.* From Lemma 16 in [Bun and Steinke, 2016], for any query $q$, we have $\varphi\left(s, s'; q\right) = \frac{\left(q(s) - q\left(s'\right)\right)^2}{2\eta^2}$.[9] Combining this result with the composition theorem (Theorem 4.4) completes the proof. $\qquad\square$

## D.2 Composition

**Theorem D.2** (Theorem 4.4 restated). *Given $k \in \mathbb{N}$, a similarity function over responses $\varphi$, and an analyst A issuing $k$ queries, if a mechanism $M$ is $\varphi$-PC, then it is $\tilde{\varphi}$-PC with respect to A, where $\tilde{\varphi}\left(s, s'; v\right) := \sum_{i=1}^k \varphi\left(s, s'; q_i\right).$*

*Proof.* Notice that

$$\ell\left(s, s'; v_k\right) = \ln\left(\frac{D\left(v_k \mid s\right)}{D\left(v_k \mid s'\right)}\right) = \sum_{i=1}^k \ln\left(\frac{D\left(r_i \mid s, v_{i-1}\right)}{D\left(r_i \mid s', v_{i-1}\right)}\right) \overset{(a)}{=} \sum_{i=1}^k \ln\left(\frac{D^{q_i}\left(r_i \mid s\right)}{D^{q_i}\left(r_i \mid s'\right)}\right) = \sum_{i=1}^k \ell\left(s, s'; r_i\right)$$

where (a) results from the fact that given $s/s'$ the additional information of $v_{i-1}$ does not change the distribution over $r_i$, since it can be encoded in the auxiliary parameter $\theta_i$. In case of a non-deterministic analyst, the sum includes an additional term of the form $\ln\left(\frac{D(c \mid s)}{D(c \mid s')}\right)$ for the coin-tossing step, as discussed in Section A.2. But since $c$ is independent of the dataset, the ratio is equal to 1 and this term has no effect.

Using this fact we can prove the theorem by induction over $k$. The base case $k = 0$ is trivial. For every $k > 0$,

$$\mathbb{E}_{V_k \sim M(s,A)}\left[e^{(\alpha-1)\left(\ell\left(s,s';V_k\right) - \alpha\varphi\left(s,s';V_k\right)\right)}\right]$$

$$= \mathbb{E}_{V_{k-1} \sim M(s,A)}\left[\mathbb{E}_{V_k \sim M(s,A)}\left[e^{(\alpha-1)\left(\ell\left(s,s';V_k\right) - \alpha\varphi\left(s,s';V_k\right)\right)} \mid V_{k-1}\right]\right]$$

$$\overset{(a)}{=} \mathbb{E}_{V_{k-1} \sim M(s,A)}\left[e^{(\alpha-1)\left(\ell\left(s,s';V_{k-1}\right) - \alpha\varphi\left(s,s';V_{k-1}\right)\right)} \cdot \overbrace{\mathbb{E}_{R_k \sim M(s,Q_k)}\left[e^{(\alpha-1)\left(\ell\left(s,s';R_k\right) - \alpha\varphi\left(s,s';Q_k\right)\right)}\right]}^{\leq 1}\right]$$

$$\overset{(b)}{\leq} \mathbb{E}_{V_{k-1} \sim M(s,A)}\left[e^{(\alpha-1)\left(\ell\left(s,s';V_{k-1}\right) - \alpha\varphi\left(s,s';V_{k-1}\right)\right)}\right]$$

$$\overset{(c)}{\leq} 1,$$

where (a) results from the definition of $\varphi\left(\cdot, \cdot; \bar{q}\right)$, the previous equation and the fact that both $\ell\left(s, s'; V_{k-1}\right)$ and $\varphi\left(s, s'; V_{k-1}\right)$ are independent of $R_k$, (b) follows from the PC definition, and (c) follows from the induction assumption. $\qquad\square$

In the next section we prove Lemma 4.5, as well as a bound on the Rényi divergence between the prior and posterior distributions over elements, which will later be used in the proof of Theorem 4.6.

---

[9]The lemma is stated for the range $\alpha \in [1, \infty]$, but holds for the range $[0, 1)$ as well, directly from the integral definition.

## D.3 Expected stability bound

**Lemma D.3** (Lemma 4.5 restated). *Given a similarity function $\varphi$ and an analyst $A$, if a mechanism $M$ that is $\varphi$-PC with respect to $A$ receives an iid sample from $\mathcal{X}^n$, then for any two elements $x, y \in \mathcal{X}$ and $\alpha \geq 1$ we have*

$$\mathbb{E}_{V \sim D(\cdot \mid x)} \left[ \exp \left( (\alpha - 1) \left( \ell(x, y; V) - \alpha \varphi(x, y; V) \right) \right) \right] \leq 1$$

*and*

$$\mathbb{E}_{V \sim D(\cdot \mid x)} \left[ \exp \left( (\alpha - 1) \left( \ell(x; V) - \alpha \varphi(x; V) \right) \right) \right] \leq 1,$$

*where $\varphi(x, y; v) := \sup\limits_{s \in \mathcal{X}^{n-1}} \left( \varphi((s, x), (s, y); v) \right)$ and $\varphi(x; v) := \ln \left( \mathbb{E}_{Y \sim D} \left[ e^{\varphi(x, Y; v)} \right] \right)$.*

*Proof.* The notation used in this proof is defined in Appendix B.

Notice that since the dataset was sampled iid, we have $D(v \mid x) = \mathbb{E}_{S \sim D^{(n-1)}} \left[ D(v \mid (S, x)) \right]$, so the first part is a direct result of Part 1 of Theorem B.14, where $P_\lambda := D(\cdot \mid x)$, $Q_\lambda := D(\cdot \mid y)$, and the convexity is taken over the convex combination of the sample sets of size $n - 1$.

For the second part, notice that $D(v) = \mathbb{E}_{X \sim D} \left[ D(v \mid X) \right]$, so

$$\mathbb{E}_{V \sim D(\cdot \mid x)} \left[ \exp \left( (\alpha - 1) \left( \ell(x; V) - \alpha \varphi(x; V) \right) \right) \right]$$

$$\overset{(a)}{\leq} \mathbb{E}_{V \sim D(\cdot \mid x)} \left[ \exp \left( (\alpha - 1) \left( \ell(x; V) - \alpha \mathbb{E}_{Y \sim D} \left[ \varphi(x, Y; v) \right] \right) \right) \right]$$

$$\overset{(b)}{\leq} 1,$$

where (a) results from Jensen's inequality for the convex function $e^x$ and (b) from combining the first part with Part 2 of Theorem B.14, where $P := D(\cdot \mid x)$, $Q_\lambda := D$, and the convexity is taken over the convex combination of the elements. $\qquad \square$

Next, we bound the Rényi divergence between the prior and posterior distributions for PC mechanisms.

**Lemma D.4.** *Given an element $x \in \mathcal{X}$, a similarity function over views $\varphi$ and an analyst $A$, if a mechanism $M$ that is $\varphi$-PC with respect to $A$ receives an iid sample from $\mathcal{X}^n$, then for any $\alpha \geq 1$ we have*

$$\mathbf{D}_\alpha \left( D_\mathcal{V}(\cdot \mid x) \,\|\, D_\mathcal{V} \right) \leq \frac{1}{2(\alpha - 1)} \ln \left( \mathbb{E}_{V \sim D} \left[ \exp \left( 2\alpha(2\alpha - 1) \varphi(x; V) \right) \right] \right),$$

*where $\varphi(x; v)$ is as was defined in Lemma 4.5.*

*Proof.*

$$\exp\left((\alpha - 1)\mathbf{D}_\alpha\left(D_\mathcal{V}\left(\cdot\,|X\right)\|D_\mathcal{V}\right)\right)$$

$$= \underset{V \sim D(\cdot\,|\,x)}{\mathbb{E}}\left[\left(\frac{D\left(V\,|\,x\right)}{D\left(V\right)}\right)^{\alpha - 1}\right]$$

$$= \underset{V \sim D}{\mathbb{E}}\left[\left(\frac{D\left(V\,|\,x\right)}{D\left(V\right)}\right)^{\alpha}\right]$$

$$= \underset{V \sim D}{\mathbb{E}}\left[\exp\left(\alpha\left(\ell\left(x;V\right) - (2\alpha - 1)\varphi\left(x;V\right)\right)\right) \cdot \exp\left(\alpha\left(2\alpha - 1\right)\varphi\left(x;V\right)\right)\right]$$

$$\overset{(a)}{\leq} \sqrt{\underset{V \sim D}{\mathbb{E}}\left[\exp\left(2\alpha\left(\ell\left(x;V\right) - (2\alpha - 1)\varphi\left(x;V\right)\right)\right)\right] \cdot \underset{V \sim D}{\mathbb{E}}\left[\exp\left(2\alpha\left(2\alpha - 1\right)\varphi\left(x;V\right)\right)\right]}$$

$$= \sqrt{\underset{V \sim D}{\mathbb{E}}\left[\left(\frac{D\left(V\,|\,x\right)}{D\left(V\right)}\right)^{2\alpha}\exp\left(-2\alpha\left(2\alpha - 1\right)\varphi\left(x;V\right)\right)\right] \cdot \underset{V \sim D}{\mathbb{E}}\left[\exp\left(2\alpha\left(2\alpha - 1\right)\varphi\left(x;V\right)\right)\right]}$$

$$= \sqrt{\underset{V \sim D(\cdot\,|\,x)}{\mathbb{E}}\left[\left(\frac{D\left(V\,|\,x\right)}{D\left(V\right)}\right)^{2\alpha - 1}\exp\left(-2\alpha\left(2\alpha - 1\right)\varphi\left(x;V\right)\right)\right] \cdot \underset{V \sim D}{\mathbb{E}}\left[\exp\left(2\alpha\left(2\alpha - 1\right)\varphi\left(x;V\right)\right)\right]}$$

$$= \sqrt{\underset{V \sim D(\cdot\,|\,x)}{\mathbb{E}}\left[\exp\left((2\alpha - 1)\left(\ell\left(x;V\right) - 2\alpha\varphi\left(x;V\right)\right)\right)\right] \cdot \underset{V \sim D}{\mathbb{E}}\left[\exp\left(2\alpha\left(2\alpha - 1\right)\varphi\left(x;V\right)\right)\right]}$$

$$\overset{(b)}{\leq} \sqrt{\underset{V \sim D}{\mathbb{E}}\left[\exp\left(2\alpha\left(2\alpha - 1\right)\varphi\left(x;V\right)\right)\right]}$$

where (a) results from the Cauchy-Schwarz inequality and (b) from the second part of Lemma 4.5.

Taking the log of both sides and dividing by $\alpha - 1$ completes the proof. $\qquad\square$

In the next section we prove a high probability bound on the stability loss in terms of the similarity function, as a second step toward proving Theorem 4.6.

## D.4   High probability stability bound

**Lemma D.5.** *Given a similarity function over views $\varphi$ and an analyst $A$, if a mechanism $M$ that is $\varphi$-PC with respect to $A$ receives an iid sample from $\mathcal{X}^n$, then*

$$\underset{\substack{X \sim D, S \sim D^{(n)} \\ V \sim M(S,A)}}{Pr}\left[|\ell\left(X;V\right)| > \varepsilon\left(X;V\right)\right] \leq \frac{2^{\frac{1}{4}}\left(e^\phi + 1\right)\delta^2}{e^\phi - 1}\left(\underset{\substack{X \sim D \\ V \sim D}}{\mathbb{E}}\left[\exp\left(12\varphi\left(X;V\right)\right)\right]\right)^{\frac{1}{8}}$$

*for any $\delta > 0$, $\phi \geq \underset{\substack{X \sim D, S \sim D^{(n)} \\ V \sim M(S,A)}}{\mathbb{E}}\left[\varphi\left(X;V\right)\right]$, where $\varepsilon\left(x;v\right) \coloneqq \varphi\left(x;v\right) + \phi + 2\sqrt{2\ln\left(\frac{1}{\delta}\right)\left(\varphi\left(x;v\right) + \phi\right)}$ and $\varphi\left(x;v\right)$ is as was defined in Lemma 4.5.*

Before we prove this lemma, we present several supporting claims. Throughout, we will fix a similarity function over views $\varphi$, an analyst $A$, and a mechanism $M$ which is $\varphi$-PC with respect to $A$ receiving an iid sample from $\mathcal{X}^n$. The notation used here is all defined in Appendix B.

**Claim D.6.** *Given $\phi, \delta > 0$ and two sample sets $s, s' \in \mathcal{X}^n$, we have $D\left(\cdot\,|\,s\right) \leq^+_{\left(\varepsilon_{s,s'}, \delta_{s,s'}\right)} D\left(\cdot\,|\,s'\right)$, where $\varepsilon_{s,s'}\left(v\right) \coloneqq \varphi\left(s,s';v\right) + 2\sqrt{2\ln\left(\frac{1}{\delta}\right)\left(\varphi\left(s,s';v\right) + \phi\right)}$ and $\delta_{s,s'} \coloneqq \delta\sqrt{\underset{V \sim D(\cdot\,|\,s)}{\mathbb{E}}\left[\sqrt{\frac{\varphi(s,s';V)}{\phi} + 1}\right]}$.*

*Proof.* Denote $\Omega \coloneqq \mathcal{V}$, $P \coloneqq D\left(\cdot\,|\,s\right)$, and $Q \coloneqq D\left(\cdot\,|\,s'\right)$. From the PC definition, for any $\alpha \geq 0$ the $\alpha$-Rényi dissimilarity between $P$ and $Q$ is bounded by $\alpha \cdot \varphi\left(s, s'; \cdot\right)$.

Invoking Theorem B.12 we get $\Pr_{V\sim P}\left[\left|\ln\left(\frac{P(V)}{Q(V)}\right)\right| > \tilde\varepsilon_{s,s'}(V)\right] \le \delta_{s,s'}$, where

$$\tilde\varepsilon_{s,s'}(v) := \varphi(s,s';v) + 2\sqrt{2\ln\left(\frac{1}{\delta}\right)(\varphi(s,s';v)+\phi)}.$$

Combining this with Part 1 of Theorem B.10 we get that $P \le^+_{(\varepsilon_{s,s'},\delta_{s,s'})} Q$. $\qquad\square$

**Claim D.7.** *Given $\phi,\delta > 0$ and two elements $x,y \in \mathcal{X}$, we have $D(\cdot\,|\,x) \le^+_{(\varepsilon_{x,y},\delta_{x,y})}$ $D(\cdot\,|\,y)$ and $D(\cdot\,|\,x) \le^-_{(\varepsilon_{x,y},\delta_{x,y})} D(\cdot\,|\,y)$, where $\varepsilon_{x,y}(v) := \sup_{s\in\mathcal{X}^{n-1}}\left(\varepsilon_{(s,x),(s,y)}(v)\right)$, $\delta_{x,y} := \mathbb{E}_{S\sim D^{(n-1)}}\left[\delta_{(S,x),(S,y)}\right]$, and $\varepsilon_{s,s'}$, $\delta_{s,s'}$ are defined as in Claim D.6.*

*Proof.* Notice that from the iid assumption, $D(v\,|\,x) = \mathbb{E}_{S\sim D^{(n-1)}}[D(v\,|\,(S,x))]$, so combining Claim D.6 and Part 1 of Theorem B.13 implies $D(\cdot\,|\,x) \le^+_{(\varepsilon_{x,y},\delta_{x,y})} D(\cdot\,|\,y)$. Combining it with Part 2 of Theorem B.10 implies $D(\cdot\,|\,x) \le^-_{(\varepsilon_{x,y},\delta_{x,y})} D(\cdot\,|\,y)$. $\qquad\square$

**Claim D.8.** *Given $\phi,\delta > 0$ and an element $x \in \mathcal{X}$, we have $D_{\mathcal{V}} \le^+_{(\varepsilon_x,\delta_{x,*})} D_{\mathcal{V}|\mathcal{X}}(\cdot\,|x)$ and $D_{\mathcal{V}|\mathcal{X}}(\cdot\,|x) \le^-_{(\varepsilon_x,\delta_{*,x})} D_{\mathcal{V}}$, where $\varepsilon_x(v) := \ln\left(\mathbb{E}_{Y\sim D}\left[e^{\varepsilon(x,Y;v)}\right]\right)$, $\delta_{x,*} := \mathbb{E}_{Y\sim D}[\delta_{x,Y}]$, $\delta_{*,x} := \mathbb{E}_{Y\sim D}[\delta_{Y,x}]$, and $\varepsilon_{x,y}$, $\delta_{x,y}$ are defined as in Claim D.7.*

*Proof.* Notice that $D(v) = \mathbb{E}_{Y\sim D}[D(v\,|\,Y)]$, so combining the first part of Claim D.7 and Part 2 of Theorem B.13 implies $D(\cdot) \le^+_{(\varepsilon_x,\delta_{x,*})} D(\cdot\,|\,x)$.

Similarly, combining the second part of Claim D.7 and Part 3 of Theorem B.13 implies $D(\cdot\,|\,x) \le^-_{(\varepsilon_x,\delta_{*,x})} D(\cdot)$. Notice that the symmetry of $\varphi$ implies $\varepsilon$ is symmetric as well, so the same $\varepsilon$ function bounds both terms. $\qquad\square$

**Claim D.9.** *Given $\phi,\epsilon,\delta > 0$ and an element $x \in \mathcal{X}$, we have $\Pr_{V\sim D}\left[\ln\left(\frac{D(V)}{D(V\,|\,x)}\right) > \varepsilon_x(V) + \epsilon\right] < \frac{e^\epsilon\delta_{x,*}}{e^\epsilon-1}$ and $\Pr_{V\sim D}\left[\ln\left(\frac{D(V\,|\,x)}{D(V)}\right) > \varepsilon_x(\omega) + \epsilon\right] < \frac{\delta_{*,x}}{e^\epsilon-1}$, where $\varepsilon_x$, $\delta_{x,*}$, and $\delta_{*,x}$ are defined as in Claim D.8.*

*Proof.* The first part results from combining the first part of Claim D.8 and Part 3 of Theorem B.10. The second part results from combining the second part of Claim D.8 and Part 4 of Theorem B.10. $\qquad\square$

*Proof of Lemma D.5.* Notice that for any $\phi,\tilde\delta > 0$ we have

$$\varepsilon_{x,y}(v) = \sup_{s\in\mathcal{X}^{n-1}}\left(\varepsilon_{(s,x),(s,y)}(v)\right)$$

$$= \sup_{s\in\mathcal{X}^{n-1}}\left(\varphi((s,x),(s,y);v) + 2\sqrt{2\ln\left(\frac{1}{\tilde\delta}\right)(\varphi((s,x),(s,y);v)+\phi)}\right)$$

$$\le \sup_{s\in\mathcal{X}^{n-1}}(\varphi((s,x),(s,y);v)) + 2\sqrt{2\ln\left(\frac{1}{\tilde\delta}\right)\left(\sup_{s\in\mathcal{X}^{n-1}}(\varphi((s,x),(s,y);v))+\phi\right)}$$

$$= \varphi(x,y;v) + 2\sqrt{2\ln\left(\frac{1}{\tilde\delta}\right)(\varphi(x,y;v)+\phi)}$$

which implies

$$\varepsilon_x(v) = \underset{Y \sim D}{\mathbb{E}}\left[\varepsilon_{x,Y}(v)\right]$$

$$\overset{(a)}{\leq} \underset{Y \sim D}{\mathbb{E}}\left[\varphi(x,Y;v) + 2\sqrt{2\ln\left(\frac{1}{\tilde{\delta}}\right)(\varphi(x,Y;v) + \phi)}\right]$$

$$\overset{(b)}{\leq} \underset{Y \sim D}{\mathbb{E}}\left[\varphi(x,Y;v)\right] + 2\sqrt{2\ln\left(\frac{1}{\delta}\right)\left(\underset{Y \sim D}{\mathbb{E}}\left[\varphi(x,Y;v)\right] + \phi\right)}$$

$$= \varphi(x;v) + 2\sqrt{2\ln\left(\frac{1}{\tilde{\delta}}\right)(\varphi(x;v) + \phi)},$$

where (a) results from the previous inequality and (b) from Jensen's inequality for the concave function $\sqrt{x}$.

Setting $\epsilon := \phi$ and $\tilde{\delta} := \delta^2$ we get

$$\underset{\substack{X \sim D, S \sim D^{(n)} \\ V \sim M(S,A)}}{\Pr}\left[|\ell(X;V)| > \varepsilon(X;V)\right]$$

$$= \underset{X \sim D}{\mathbb{E}}\left[\underset{\substack{S \sim D^{(n)} \\ V \sim M(S,A)}}{\Pr}\left[\left|\ln\left(\frac{D(V \mid X)}{D(V)}\right)\right| > \varphi(X;V) + \phi + 2\sqrt{2\ln\left(\frac{1}{\tilde{\delta}}\right)(\varphi(X;V) + \phi)}\right]\right]$$

$$\overset{(a)}{\leq} \underset{X \sim D}{\mathbb{E}}\left[\underset{\substack{S \sim D^{(n)} \\ V \sim M(S,A)}}{\Pr}\left[\left|\ln\left(\frac{D(V \mid X)}{D(V)}\right)\right| > \varepsilon_X(V) + \epsilon\right]\right]$$

$$\overset{(b)}{\leq} \frac{e^\epsilon}{e^\epsilon - 1}\underset{X \sim D}{\mathbb{E}}\left[\delta_{*,X}\right] + \frac{1}{e^\epsilon - 1}\underset{X \sim D}{\mathbb{E}}\left[\delta_{X,*}\right]$$

$$\overset{(c)}{=} \frac{e^\epsilon + 1}{e^\epsilon - 1}\underset{\substack{X \sim D, Y \sim D \\ S \sim D^{(n-1)}}}{\mathbb{E}}\left[\tilde{\delta}\sqrt{\underset{V \sim M((S,X),A)}{\mathbb{E}}\left[\sqrt{\frac{\varphi((S,X),(S,Y);V)}{\phi}} + 1\right]}\right]$$

$$\overset{(d)}{\leq} \frac{(e^\phi + 1)\delta^2}{e^\phi - 1}\sqrt{\underset{\substack{X \sim D, Y \sim D, S \sim D^{(n-1)} \\ V \sim M((S,X),A)}}{\mathbb{E}}\left[\sqrt{\frac{\varphi((S,X),(S,Y);V)}{\phi}} + 1\right]},$$

where (a) results from the previous bound, (b) from the two parts of Claim D.9, (c) from the definitions of $\delta_{x,*}$ and $\delta_{*,x}$, and (d) from Jensen's inequality for the concave function $\sqrt{x}$.

Analyzing the expectation term we get

$$
\mathop{\mathbb{E}}_{\substack{X\sim D,Y\sim D,S\sim D^{(n-1)}\\ V\sim M((S,X),A)}}\left[\sqrt{\frac{\varphi\left((S,X),(S,Y);V\right)}{\phi}+1}\right]
$$

$$
\overset{(a)}{\le}\mathop{\mathbb{E}}_{\substack{X\sim D,Y\sim D\\ V\sim D(\cdot\,|\,X)}}\left[\sqrt{\frac{\varphi\left(X,Y;V\right)}{\phi}+1}\right]
$$

$$
=\mathop{\mathbb{E}}_{\substack{X\sim D,Y\sim D\\ V\sim D}}\left[\frac{D\left(V\,|\,X\right)}{D\left(V\right)}\sqrt{\frac{\varphi\left(X,Y;V\right)}{\phi}+1}\right]
$$

$$
\overset{(b)}{\le}\sqrt{\mathop{\mathbb{E}}_{\substack{X\sim D\\ V\sim D}}\left[\left(\frac{D\left(V\,|\,X\right)}{D\left(V\right)}\right)^{2}\right]\cdot\mathop{\mathbb{E}}_{\substack{X\sim D,Y\sim D\\ V\sim D}}\left[\frac{\varphi\left(X,Y;V\right)}{\phi}+1\right]}
$$

$$
\overset{(c)}{\le}\sqrt{2\mathop{\mathbb{E}}_{\substack{X\sim D\\ V\sim D(\cdot\,|\,X)}}\left[\frac{D\left(X\,|\,V\right)}{D\left(X\right)}\right]}
$$

$$
=\sqrt{2\mathop{\mathbb{E}}_{X\sim D}\left[\exp\left(\mathbf{D}_{2}\left(D_{\mathcal{V}}\left(\cdot|X\right)\|D_{\mathcal{V}}\right)\right)\right]}
$$

$$
\overset{(d)}{\le}\sqrt{2}\left(\mathop{\mathbb{E}}_{\substack{X\sim D\\ V\sim D}}\left[\exp\left(12\varphi\left(X;V\right)\right)\right]\right)^{\frac{1}{4}},
$$

where (a) results from the definition of $\varphi\left(x,y;v\right)$, (b) from the Cauchy-Schwarz inequality, (c) from the bound on $\phi$, and (d) from Lemma D.4.

Plugging this term back in completes the proof. $\qquad\square$

We can now prove the main stability theorem for PC mechanisms.

### D.5  Bayes stability of PC mechanisms

**Theorem D.10** (Exact version of Theorem 4.6)**.** *Given a similarity function over views $\varphi$ and an analyst A, if a mechanism M that is $\varphi$-PC with respect to A receives an iid sample from $\mathcal{X}^{n}$, then*

$$
\mathop{Pr}_{\substack{S\sim D^{(n)}\\ V\sim M(S,A),Q\sim A(V)}}\left[\left|Q\left(D_{\mathcal{X}}^{v}\right)-Q\left(D_{\mathcal{X}}\right)\right|>\sigma_{Q}\cdot\sqrt{\epsilon^{2}+\xi}\right]
$$

$$
\le\mathop{Pr}_{\substack{S\sim D^{(n)}\\ V\sim M(S,A),Q\sim A(V)}}\left[\mathop{\mathbb{E}}_{X\sim D}\left[e^{27\ln\left(\frac{1}{\delta}\right)(\varphi(X;v)+\phi)}\right]>1+\frac{\epsilon^{2}}{3}\right]
$$

$$
+\frac{\delta}{\xi}\left(4\psi\left(12\right)\right)^{\frac{1}{16}}\sqrt{\left(\sqrt{\psi\left(56\right)}+1\right)\cdot\frac{e^{\phi}+1}{e^{\phi}-1}}
$$

*for any $\epsilon,\xi,\delta>0$, $\phi\ge\mathop{\mathbb{E}}_{\substack{X\sim D,S\sim D^{(n)}\\ V\sim M(S,A)}}\left[\varphi\left(X;V\right)\right]$, where $\varphi\left(x;v\right)$ is as was defined in Lemma 4.5 and $\psi\left(\alpha\right):=\mathop{\mathbb{E}}_{\substack{X\sim D,S\sim D^{(n)}\\ V\sim M(S,A)}}\left[e^{\alpha\varphi(X;V)}\right].$*

We first prove several supporting claims.

**Claim D.11.** *Given a function $f : \mathcal{X} \times \mathcal{V} \to \mathbb{R}$, an analyst $A$, and a mechanism $M$, then for any $\xi > 0$ we have*

$$\Pr_{\substack{S \sim D^{(n)} \\ V \sim M(S,A)}} \left[ \mathbf{D}_{\chi^2} \left( D_{\mathcal{X}}^v \| D_{\mathcal{X}} \right) > \mathbb{E}_{X \sim D} \left[ \left( e^{f(X,V)} - 1 \right)^2 \right] + \xi \right]$$

$$\leq \frac{1}{\xi} \sqrt{\left( \mathbb{E}_{\substack{X \sim D, S \sim D^{(n)} \\ V \sim M(S,A)}} \left[ e^{4\ell(X;V)} \right] + 1 \right) \cdot \Pr_{\substack{X \sim D, S \sim D^{(n)} \\ V \sim M(S,A)}} \left[ |\ell(X;V)| > f(X,V) \right].}$$

*Proof.* Denoting $B(f) := \{(x,v) \in \mathcal{X} \times \mathcal{V} \mid |\ell(x;v)| > \varepsilon(x;v)\}$ and $\bar{B}(f) := (\mathcal{X} \times \mathcal{V}) \setminus B(f)$ we get

$$\mathbf{D}_{\chi^2} \left( D_{\mathcal{X}}^v \| D_{\mathcal{X}} \right) - \mathbb{E}_{X \sim D} \left[ \left( e^{f(X,V)} - 1 \right)^2 \right]$$

$$= \mathbb{E}_{X \sim D} \left[ \left( e^{\ell(X;V)} - 1 \right)^2 \right] - \mathbb{E}_{X \sim D} \left[ \left( e^{f(X,V)} - 1 \right)^2 \right]$$

$$= \mathbb{E}_{X \sim D} \left[ \left( \left( e^{\ell(X;V)} - 1 \right)^2 - \left( e^{f(X,V)} - 1 \right)^2 \right) \cdot \mathbb{1}_{\bar{B}(f)} \left( (X,V) \right) \right]$$

$$+ \mathbb{E}_{X \sim D} \left[ \left( \left( e^{\ell(X;V)} - 1 \right)^2 - \left( e^{f(X,V)} - 1 \right)^2 \right) \cdot \mathbb{1}_{B(f)} \left( (X,V) \right) \right]$$

$$\overset{(a)}{\leq} \mathbb{E}_{X \sim D} \left[ \left( \left( e^{|\ell(X;V)|} - 1 \right)^2 - \left( e^{f(X,V)} - 1 \right)^2 \right) \cdot \mathbb{1}_{\bar{B}(f)} \left( (X,V) \right) \right]$$

$$+ \mathbb{E}_{X \sim D} \left[ \left( \left( e^{\ell(X;V)} - 1 \right)^2 - \left( e^{f(X,V)} - 1 \right)^2 \right) \cdot \mathbb{1}_{B(f)} \left( (X,V) \right) \right]$$

$$\overset{(b)}{\leq} \mathbb{E}_{X \sim D} \left[ \left( e^{\ell(X;V)} - 1 \right)^2 \cdot \mathbb{1}_{B(f)} \left( (X,V) \right) \right]$$

where (a) results from the fact that $(e^x - 1)^2 \leq \left( e^{|x|} - 1 \right)^2$ and (b) from removing negative terms. Using this bound we get

$$\Pr_{\substack{S \sim D^{(n)} \\ V \sim M(S,A)}} \left[ \mathbf{D}_{\chi^2} \left( D_{\mathcal{X}}^v \| D_{\mathcal{X}} \right) > \mathbb{E}_{X \sim D} \left[ \left( e^{f(X,V)} - 1 \right)^2 \right] + \xi \right]$$

$$\overset{(a)}{\leq} \Pr_{\substack{S \sim D^{(n)} \\ V \sim M(S,A)}} \left[ \mathbb{E}_{X \sim D} \left[ \left( e^{\ell(X;V)} - 1 \right)^2 \cdot \mathbb{1}_{B(f)} \left( (X,V) \right) \right] > \xi \right]$$

$$\overset{(b)}{\leq} \frac{1}{\xi} \mathbb{E}_{\substack{X \sim D, S \sim D^{(n)} \\ V \sim M(S,A)}} \left[ \left( e^{\ell(X;V)} - 1 \right)^2 \cdot \mathbb{1}_{B(f)} \left( (X,V) \right) \right]$$

$$\overset{(c)}{\leq} \frac{1}{\xi} \sqrt{\mathbb{E}_{\substack{X \sim D, S \sim D^{(n)} \\ V \sim M(S,A)}} \left[ \left( e^{\ell(X;V)} - 1 \right)^4 \right] \cdot \mathbb{E}_{\substack{X \sim D, S \sim D^{(n)} \\ V \sim M(S,A)}} \left[ \mathbb{1}_{B(f)}^2 \left( (X,V) \right) \right]}$$

$$\overset{(d)}{\leq} \frac{1}{\xi} \sqrt{\left( \mathbb{E}_{\substack{X \sim D, S \sim D^{(n)} \\ V \sim M(S,A)}} \left[ e^{4\ell(X;V)} \right] + 1 \right) \cdot \Pr_{\substack{X \sim D, S \sim D^{(n)} \\ V \sim M(S,A)}} \left[ (X,V) \in B(f) \right]}$$

where (a) results from the previous bound, (b) from Markov's inequality, (c) from the Cauchy-Schwarz inequality, and (d) from the inequality $(x - 1)^4 \leq x^4 + 1$ for any $x \geq 0$. $\qquad\square$

**Claim D.12.** *Given $\phi > 0$, $\delta \in \left(0, \frac{1}{e}\right)$, and a view $v \in \mathcal{V}$, we have*

$$\mathbb{E}_{X \sim D}\left[\left(e^{\varepsilon(X;v)} - 1\right)^2\right] \leq 3\left(\mathbb{E}_{X \sim D}\left[e^{27\ln\left(\frac{1}{\delta}\right)(\varphi(X;v)+\phi)}\right] - 1\right)$$

*where $\varepsilon(x; v)$ was defined in Lemma D.5.*

*Proof.* We first notice that for any $\alpha \geq 1$,

$$\mathbb{E}_{X \sim D}\left[e^{\alpha\varphi(X;v)}\right] \stackrel{(a)}{=} \mathbb{E}_{X \sim D}\left[\left(\mathbb{E}_{Y \sim D}\left[e^{\varphi(X,Y;v)}\right]\right)^\alpha\right] \stackrel{(b)}{\leq} \mathbb{E}_{X,Y \sim D}\left[e^{\alpha\varphi(X,Y;v)}\right] \tag{1}$$

where (a) follows from the definition of $\varphi(x; v)$ and (b) from Jensen's inequality for the convex function $x^\alpha$.

Next we prove a useful identity. For any $a, b \geq 0$,

$$e^{a+b} - 1 = e^a\left(e^b - 1\right) + \left(e^a - 1\right) = e^a\sqrt{\left(e^b - 1\right)^2} + \sqrt{\left(e^a - 1\right)^2} \leq e^a\sqrt{\left(e^{\frac{3}{2}b^2} - 1\right)} + \sqrt{\left(e^{2a} - 1\right)}$$

where the inequality results from the inequalities $\left(e^x - 1\right)^2 \leq e^{\frac{3}{2}x^2} - 1$ for any $x \in \mathbb{R}$ and $\left(e^x - 1\right)^2 \leq e^{2x} - 1$ for any $x \geq 0$.

Using this bound we get

$$\left(e^{a+b} - 1\right)^2 \stackrel{(a)}{\leq} \left(e^a\sqrt{\left(e^{\frac{3}{2}b^2} - 1\right)} + \sqrt{\left(e^{2a} - 1\right)}\right)^2$$

$$= e^{2a}\left(e^{\frac{3}{2}b^2} - 1\right) + 2e^a\sqrt{\left(e^{\frac{3}{2}b^2} - 1\right)\left(e^{2a} - 1\right)} + \left(e^{2a} - 1\right)$$

$$= \left(e^{2a+\frac{3}{2}b^2} - 1\right) + 2e^a\sqrt{\left(e^{\frac{3}{2}b^2} - 1\right)\left(e^{2a} - 1\right)}$$

$$\stackrel{(b)}{\leq} \left(e^{2a+\frac{3}{2}b^2} - 1\right) + 2\left(e^{3a+\frac{3}{2}b^2} - 1\right)$$

$$\leq 3\left(e^{3a+\frac{3}{2}b^2} - 1\right)$$

where (a) results from the previous bound and (b) from the inequalities $\left(e^x - 1\right)\left(e^y - 1\right) \leq \left(e^{x+y} - 1\right)^2$ and $e^x\left(e^y - 1\right) \leq \left(e^{x+y} - 1\right)$ for any $x, y \geq 0$.

Using this inequality we get

$$\mathbb{E}_{X \sim D}\left[\left(e^{\varepsilon(X;v)} - 1\right)^2\right]$$

$$= \mathbb{E}_{X \sim D}\left[\left(\exp\left(\varphi(X;v) + \phi + 4\sqrt{\ln\left(\frac{1}{\delta}\right)(\varphi(X;v) + \phi)}\right) - 1\right)^2\right]$$

$$\stackrel{(a)}{\leq} \mathbb{E}_{X \sim D}\left[3\left(\exp\left(3\left(1 + 8\ln\left(\frac{1}{\delta}\right)\right)(\varphi(X;v) + \phi)\right) - 1\right)\right]$$

$$\stackrel{(b)}{\leq} 3\left(\mathbb{E}_{X \sim D}\left[e^{27\ln\left(\frac{1}{\delta}\right)(\varphi(X;v)+\phi)}\right] - 1\right)$$

where (a) results from the previous inequality and (b) from the bound on $\delta$. $\qquad\square$

*Proof of Theorem D.10.* First notice

$$
\underset{\substack{X\sim D, S\sim D^{(n)} \\ V\sim M(S,A)}}{\mathbb{E}}\left[e^{4\ell(X;V)}\right] = \underset{X\sim D}{\mathbb{E}}\left[\underset{V\sim D}{\mathbb{E}}\left[\left(\frac{D\left(V\mid X\right)}{D\left(V\right)}\right)^{4}\right]\right]
$$

$$
= \underset{X\sim D}{\mathbb{E}}\left[\underset{V\sim D(\cdot\mid X)}{\mathbb{E}}\left[\left(\frac{D\left(V\mid X\right)}{D\left(V\right)}\right)^{3}\right]\right]
$$

$$
= \underset{X\sim D}{\mathbb{E}}\left[\exp\left(\mathbf{D}_{4}\left(D_{\mathcal{V}}\left(\cdot\mid X\right)\|D_{\mathcal{V}}\right)\right)\right]
$$

$$
\overset{(a)}{\leq} \underset{X\sim D}{\mathbb{E}}\left[\sqrt{\underset{V\sim D}{\mathbb{E}}\left[e^{56\varphi(X;V)}\right]}\right]
$$

$$
\overset{(b)}{\leq} \sqrt{\underset{X\sim D, V\sim D}{\mathbb{E}}\left[e^{56\varphi(X;V)}\right]}
$$

$$
\overset{(c)}{=} \sqrt{\psi\left(56\right)}
$$

where (a) results from Lemma D.4, (b) from Jensen's inequality for the concave function $\sqrt{x}$, and (c) from the definition of $\psi$.

Using this bound we get

$$
\underset{\substack{S\sim D^{(n)} \\ V\sim M(S,A), Q\sim A(V)}}{\Pr}\left[|Q\left(D_{\mathcal{X}}^{v}\right) - Q\left(D_{\mathcal{X}}\right)| > \sigma_{Q}\cdot\sqrt{\epsilon^{2}+\xi}\right]
$$

$$
\overset{(a)}{\leq} \underset{\substack{S\sim D^{(n)} \\ V\sim M(S,A), Q\sim A(V)}}{\Pr}\left[\mathbf{D}_{\chi^{2}}\left(D_{\mathcal{X}}^{V}\|D_{\mathcal{X}}\right) > \epsilon^{2}+\xi\right]
$$

$$
\overset{(b)}{\leq} \underset{\substack{S\sim D^{(n)} \\ V\sim M(S,A), Q\sim A(V)}}{\Pr}\left[\mathbf{D}_{\chi^{2}}\left(D_{\mathcal{X}}^{v}\|D_{\mathcal{X}}\right) > \underset{X\sim D}{\mathbb{E}}\left[\left(e^{\varepsilon(X;V)}-1\right)^{2}\right]+\xi\right]
$$

$$
+ \underset{\substack{S\sim D^{(n)} \\ V\sim M(S,A), Q\sim A(V)}}{\Pr}\left[\underset{X\sim D}{\mathbb{E}}\left[\left(e^{\varepsilon(X;V)}-1\right)^{2}\right] > \epsilon^{2}\right]
$$

$$
\overset{(c)}{\leq} \frac{1}{\xi}\sqrt{\left(\underset{\substack{X\sim D, S\sim D^{(n)} \\ V\sim M(S,A)}}{\mathbb{E}}\left[e^{4\ell(X;V)}\right]+1\right)\cdot \underset{\substack{X\sim D, S\sim D^{(n)} \\ V\sim M(S,A)}}{\Pr}\left[|\ell\left(X;V\right)| > \varepsilon\left(X;V\right)\right]}
$$

$$
+ \underset{\substack{S\sim D^{(n)} \\ V\sim M(S,A), Q\sim A(V)}}{\Pr}\left[3\left(\underset{X\sim D}{\mathbb{E}}\left[e^{27\ln\left(\frac{1}{\delta}\right)(\varphi(X;v)+\phi)}\right]-1\right) > \epsilon^{2}\right]
$$

$$
\overset{(d)}{\leq} \frac{1}{\xi}\sqrt{\left(\sqrt{\psi\left(56\right)}+1\right)\cdot \frac{2^{\frac{1}{4}}\left(e^{\phi}+1\right)\delta^{2}}{e^{\phi}-1}\left(\psi\left(12\right)\right)^{\frac{1}{8}}}
$$

$$
+ \underset{\substack{S\sim D^{(n)} \\ V\sim M(S,A), Q\sim A(V)}}{\Pr}\left[\underset{X\sim D}{\mathbb{E}}\left[e^{27\ln\left(\frac{1}{\delta}\right)(\varphi(X;v)+\phi)}\right] > 1+\frac{\epsilon^{2}}{3}\right]
$$

where (a) results from Corollary 3.6, (b) from the inequality $\Pr\left[a > b\right] \leq \Pr\left[a > c \vee c > b\right] \leq \Pr\left[a > c\right] + \Pr\left[c > b\right]$, (c) from Claims D.11 and D.11, and (d) from the previous bound and Lemma D.5. $\qquad\square$

# E    Missing parts from Section 5

Using Theorem D.10, one only has to bound $\underset{X \sim D}{\mathbb{E}} \left[ e^{27\ln\left(\frac{1}{\delta}\right)(\varphi(X;v)+\phi)} \right]$ and $\psi(\alpha)$ to prove Bayes stability. To do so, we first prove a supporting claim and then provide separate bounds for the bounded case (Section E.1) and sub-Gaussian case (Section E.2).

**Claim E.1.** *Given $k \in \mathbb{N}$; $\eta > 0$ and a view $v \in \mathcal{V}$ consisting of responses to $k$ linear queries, produced by a Gaussian mechanism $M$ with noise parameter $\eta$ which receives an iid sampled dataset of size $n$, then for any $\alpha \geq 1$ we have*

$$\underset{X \sim D}{\mathbb{E}} \left[ e^{\alpha \varphi(X;v)} \right] \leq \underset{X,Y \sim D}{\mathbb{E}} \left[ e^{\alpha \varphi(X,Y;v)} \right] = \underset{X,Y \sim D}{\mathbb{E}} \left[ \exp\left( \frac{\alpha}{2n^2\eta^2} \sum_{i=1}^{k} (q_i(X) - q_i(Y))^2 \right) \right],$$

*where $\varphi(x;v)$ and $\phi$ are defined as in Theorem 4.6.*

*Proof.* Notice that

$$
\begin{aligned}
\underset{X \sim D}{\mathbb{E}} \left[ e^{\alpha \varphi(X;v)} \right] &\overset{(a)}{=} \underset{X \sim D}{\mathbb{E}} \left[ \left( \underset{Y \sim D}{\mathbb{E}} \left[ e^{\varphi(X,Y;v)} \right] \right)^{\alpha} \right] \\
&\overset{(b)}{\leq} \underset{X,Y \sim D}{\mathbb{E}} \left[ e^{\alpha \varphi(X,Y;v)} \right] \\
&\overset{(c)}{=} \underset{X,Y \sim D}{\mathbb{E}} \left[ \exp\left( \alpha \sup_{s \in \mathcal{X}^{n-1}} \left( \frac{\|\bar{q}_v((s,X)) - \bar{q}_v((s,Y))\|^2}{2\eta^2} \right) \right) \right] \\
&= \underset{X,Y \sim D}{\mathbb{E}} \left[ \exp\left( \frac{\alpha}{2\eta^2} \sup_{s \in \mathcal{X}^{n-1}} \left( \sum_{i=1}^{k} (q_i((s,X)) - q_i((s,Y)))^2 \right) \right) \right] \\
&= \underset{X,Y \sim D}{\mathbb{E}} \left[ \exp\left( \frac{\alpha}{2n^2\eta^2} \sum_{i=1}^{k} (q_i(X) - q_i(Y))^2 \right) \right]
\end{aligned}
$$

where (a) results from the definition of $\varphi(x;v)$, (b) from Jensen's inequality for the convex function $x^a$ for any $a \geq 1$, and (c) from Theorem 4.3. $\qquad\square$

## E.1    Generalization guarantees, bounded case

**Claim E.2.** *Given $k \in \mathbb{N}$; $\eta, \Delta, \sigma > 0$; $\phi \geq 0$, and a view $v \in \mathcal{V}$ consisting of responses to $k$ $\Delta$-bounded linear queries with variance bounded by $\sigma^2$, produced by a Gaussian mechanism $M$ with noise parameter $\eta$ which receives an iid sampled dataset of size $n$, then for any $1 \leq \alpha \leq \frac{1}{\frac{k\Delta^2}{2n^2\eta^2} + \phi}$ we have*

$$\underset{X \sim D}{\mathbb{E}} \left[ e^{\alpha(\varphi(X;v)+\phi)} \right] \leq \underset{X,Y \sim D}{\mathbb{E}} \left[ e^{\alpha(\varphi(X,Y;v)+\phi)} \right] \leq 1 + 2\alpha \left( \phi + \frac{k\sigma^2}{n^2\eta^2} \right),$$

*where $\varphi(x;v)$ is as was defined in Theorem 4.6.*

*Proof.* First notice that for any linear query $q$ with variance $\sigma^2$,

$$
\begin{aligned}
\underset{X,Y \sim D}{\mathbb{E}} \left[ (q(X) - q(Y))^2 \right] &= \underset{X,Y \sim D}{\mathbb{E}} \left[ ((q(X) - q(D)) + (q(D) - q(Y)))^2 \right] \\
&\overset{(a)}{=} \underset{X \sim D}{\mathbb{E}} \left[ (q(X) - q(D))^2 \right] + \underset{Y \sim D}{\mathbb{E}} \left[ (q(D) - q(Y))^2 \right] \quad (2)\\
&\qquad + 2 \underset{X \sim D}{\mathbb{E}} [q(X) - q(D)] \cdot \underset{Y \sim D}{\mathbb{E}} [q(D) - q(Y)] \\
&\overset{(b)}{=} 2\sigma^2
\end{aligned}
$$

where (a) results from the fact $X, Y$ are independent and (b) from the fact that $q(X)$ is a random variable with expectation $q(D)$ and variance $\sigma^2$.

For any two elements $x, y \in \mathcal{X}$ we have

$$\varphi(x, y; v) = \frac{1}{2n^2\eta^2} \sum_{i=1}^{k} (q_i(X) - q_i(Y))^2 \leq \frac{k\Delta^2}{2n^2\eta^2}. \tag{3}$$

Using this bound we get

$$
\mathop{\mathbb{E}}_{X \sim D} \left[ e^{\alpha(\varphi(X;v)+\phi)} \right] \stackrel{(a)}{\leq} \mathop{\mathbb{E}}_{X,Y \sim D} \left[ e^{\alpha(\varphi(X,Y;v)+\phi)} \right]
$$

$$
= \mathop{\mathbb{E}}_{X,Y \sim D} \left[ \exp\left( \alpha \left( \phi + \frac{1}{2n^2\eta^2} \sum_{i=1}^{k} (q_i(X) - q_i(Y))^2 \right) \right) \right]
$$

$$
\stackrel{(b)}{\leq} \mathop{\mathbb{E}}_{X,Y \sim D} \left[ 1 + 2\alpha \left( \phi + \frac{1}{2n^2\eta^2} \sum_{i=1}^{k} (q_i(X) - q_i(Y))^2 \right) \right]
$$

$$
= 1 + \alpha \left( 2\phi + \frac{1}{n^2\eta^2} \sum_{i=1}^{k} \mathop{\mathbb{E}}_{X,Y \sim D} \left[ (q_i(X) - q_i(Y))^2 \right] \right)
$$

$$
\stackrel{(c)}{\leq} 1 + \alpha \left( 2\phi + \frac{1}{n^2\eta^2} \sum_{i=1}^{k} 2\sigma^2 \right)
$$

$$
= 1 + 2\alpha \left( \phi + \frac{k\sigma^2}{n^2\eta^2} \right)
$$

where (a) results from combining the assumption $\alpha \geq 1$ with Claim E.1, (b) from combining Equation 3, the assumption $\alpha \leq \frac{1}{\frac{k\Delta^2}{2n^2\eta^2} + \phi}$, and the inequality $e^x \leq 1 + 2x$ for any $0 \leq x \leq 1$, and (c) from Equation 2. $\qquad \square$

**Lemma E.3.** *Given $k \in \mathbb{N}$; $\Delta, \sigma, \eta \geq 0$; $0 < \delta \leq \frac{1}{e}$; and an analyst $A$ issuing $k$ $\Delta$-bounded linear queries with variance bounded by $\sigma^2$, if $M$ is a Gaussian mechanism with noise parameter $\eta$ that receives an iid dataset of size $n \geq \frac{9\sqrt{k}\Delta}{\eta}\sqrt{\ln\left(\frac{1}{\delta}\right)}$, then $M$ is $\left( \frac{\sigma^2}{n\eta}\sqrt{488k\ln\left(\frac{1}{\delta}\right)}, \frac{2n^3\eta^3\delta}{k^{\frac{3}{2}}\sigma^3} \right)$-Bayes stable.*

*Proof.* Setting $\phi := \frac{2k\sigma^2}{n^2\eta^2}$ we get

$$
27\ln\left(\frac{1}{\delta}\right) \stackrel{(a)}{<} \frac{2n^2\eta^2}{5k\Delta^2} = \frac{1}{\frac{5k\Delta^2}{2n^2\eta^2}} \stackrel{(b)}{<} \frac{1}{\frac{k\Delta^2}{2n^2\eta^2} + \phi}
$$

where (a) results from the assumption $n \geq \frac{9\sqrt{k}\Delta}{\eta}\sqrt{\ln\left(\frac{1}{\delta}\right)}$ and (b) from the definition of $\phi$.

Using this bound we can invoke Claim E.2, and get that for any view $v \in \mathcal{V}$,

$$
\mathop{\mathbb{E}}_{X \sim D} \left[ e^{27\ln\left(\frac{1}{\delta}\right)(\varphi(X;v)+\phi)} \right] \leq 1 + 54 \left( \frac{k\sigma^2}{n^2\eta^2} + \phi \right) \ln\left(\frac{1}{\delta}\right). \tag{4}
$$

Next, we notice that

$$
\mathop{\mathbb{E}}_{X \sim D, V \sim D} [\varphi(X; V)] = \mathop{\mathbb{E}}_{X \sim D, V \sim D} \left[ \ln\left( \mathop{\mathbb{E}}_{Y \sim D} \left[ e^{\varphi(X,Y;V)} \right] \right) \right]
$$

$$
\stackrel{(a)}{\leq} \ln\left( \mathop{\mathbb{E}}_{X,Y \sim D, V \sim D} \left[ e^{\varphi(X,Y;V)} \right] \right)
$$

$$
\stackrel{(b)}{\leq} \ln\left( 1 + \frac{2k\sigma^2}{n^2\eta^2} \right) \tag{5}
$$

$$
\stackrel{(c)}{\leq} \frac{2k\sigma^2}{n^2\eta^2}
$$

$$
\stackrel{(d)}{=} \phi
$$

where (a) results from Jensen's inequality for the concave function $\ln(x)$, (b) from Claim E.2 where the condition on $\alpha$ is fulfilled by the assumption that $n \geq \frac{9\sqrt{k}\Delta}{\eta}\sqrt{\ln\left(\frac{1}{\delta}\right)}$, (c) from the inequality $\ln(1+x) \leq x$ for any $x \geq 0$, and (d) from the definition of $\phi$.

Similarly, for any $\alpha \leq \frac{2n^2\eta^2}{k\Delta^2}$ we have

$$\psi(\alpha) \overset{(a)}{\leq} 1 + \frac{2\alpha k\sigma^2}{n^2\eta^2} \overset{(b)}{\leq} 1 + \frac{\alpha k\Delta^2}{2n^2\eta^2} \overset{(c)}{\leq} 1 + \frac{\alpha k\Delta^2}{2n^2\eta^2} \overset{(d)}{\leq} 1 + \frac{\alpha}{162} \tag{6}$$

where (a) results from the assumption on $\alpha$ and Claim E.2, (b) from the fact that for any distribution, $\sigma^2 \leq \frac{\Delta^2}{4}$, (c) from the assumption that $\delta \leq \frac{1}{e}$, and (d) from the assumption that $n \geq \frac{9\sqrt{k}\Delta}{\eta}\sqrt{\ln\left(\frac{1}{\delta}\right)}$.

Setting $\xi := \frac{2k\sigma^2}{n^2\eta^2}$ we get

$$\Pr_{\substack{S\sim D^{(n)}\\ V\sim M(S,A),Q\sim A(V)}}\left[|Q(D_{\mathcal{X}}^v) - Q(D_{\mathcal{X}})| > \frac{\sigma^2}{n\eta}\sqrt{488k\ln\left(\frac{1}{\delta}\right)}\right]$$

$$\overset{(a)}{\leq} \Pr_{\substack{S\sim D^{(n)}\\ V\sim M(S,A),Q\sim A(V)}}\left[|Q(D_{\mathcal{X}}^v) - Q(D_{\mathcal{X}})| > \sigma_Q\sqrt{\frac{488k\sigma^2}{n^2\eta^2}\ln\left(\frac{1}{\delta}\right)}\right]$$

$$\overset{(b)}{\leq} \Pr_{\substack{S\sim D^{(n)}\\ V\sim M(S,A),Q\sim A(V)}}\left[|Q(D_{\mathcal{X}}^v) - Q(D_{\mathcal{X}})| > \sigma_Q\sqrt{162\left(\frac{k\sigma^2}{n^2\eta^2} + \frac{2k\sigma^2}{n^2\eta^2}\right)\ln\left(\frac{1}{\delta}\right) + \frac{2k\sigma^2}{n^2\eta^2}}\right]$$

$$\overset{(c)}{=} \Pr_{\substack{S\sim D^{(n)}\\ V\sim M(S,A),Q\sim A(V)}}\left[|Q(D_{\mathcal{X}}^v) - Q(D_{\mathcal{X}})| > \sigma_Q\sqrt{162\left(\frac{k\sigma^2}{n^2\eta^2} + \phi\right)\ln\left(\frac{1}{\delta}\right) + \xi}\right]$$

$$\overset{(d)}{\leq} \Pr_{\substack{S\sim D^{(n)}\\ V\sim M(S,A),Q\sim A(V)}}\left[\mathbb{E}_{X\sim D}\left[e^{27\ln\left(\frac{1}{\delta}\right)(\varphi(X;v)+\phi)}\right] > 1 + 54\left(\frac{k\sigma^2}{n^2\eta^2} + \phi\right)\ln\left(\frac{1}{\delta}\right)\right]$$

$$+ \frac{\delta}{\xi}(4\psi(12))^{\frac{1}{16}}\sqrt{\left(\sqrt{\psi(56)}+1\right)\frac{e^\phi+1}{e^\phi-1}}$$

$$\overset{(e)}{\leq} \frac{\delta}{\xi}(4\psi(12))^{\frac{1}{16}}\sqrt{\left(\sqrt{\psi(56)}+1\right)\frac{e^\phi+1}{e^\phi-1}}$$

$$\overset{(f)}{\leq} \frac{\delta}{\xi}\left(\frac{116}{27}\right)^{\frac{1}{16}}\sqrt{\left(\sqrt{\frac{109}{81}}+1\right)\frac{3}{\phi}}$$

$$\leq \frac{3\delta}{\xi\sqrt{\phi}}$$

$$\overset{(c)}{=} \frac{2n^3\eta^3\delta}{k^{\frac{3}{2}}\sigma^3}$$

where (a) results from fact that the variance of all queries issued by $A$ is bounded by $\sigma^2$, (b) from the assumption that $\delta \leq \frac{1}{e}$, (c) from the definition of $\phi$ and $\xi$, (d) from Theorem D.10 where the condition was proven in Equation 5, (e) from Equation 4, and (f) from Equation 6, and the fact that $\frac{e^x+1}{e^x-1} \leq \frac{3}{x}$ for any $0 < x < 2$. $\qquad\square$

**Theorem E.4** (Exact version of Theorem 5.1). *Given $k \in \mathbb{N}$; $\Delta, \sigma, \epsilon \geq 0$; $0 < \delta \leq \frac{1}{e}$; and an analyst $A$ issuing $k$ $\Delta$-bounded linear queries with variance bounded by $\sigma^2$, if $M$ is a Gaussian mechanism with noise parameter $\eta = (122)^{\frac{1}{4}}\sigma\sqrt{\frac{\sqrt{k}}{n}}$ that receives an iid dataset of size $n \geq \max\left\{\frac{9\Delta}{\epsilon}, \frac{89\sigma^2}{\epsilon^2}\right\}\sqrt{k}\ln\left(\frac{8k}{\delta}\right)$, and $k \geq 2n$, then $M$ is $(\epsilon, \delta)$-distribution accurate.*

*Proof.* From Lemma E.3 setting $\delta' := \frac{\delta}{8k}$ (where $\delta'$ denotes the parameter $\delta$ used in the Lemma) we get that $M$ is $\left(\frac{\sigma^2}{n\eta}\sqrt{488k\ln\left(\frac{4k}{\delta}\right)}, \frac{n^3\eta^3\delta}{k^{\frac{5}{2}}\sigma^3}\right)$-Bayes stable. From Lemma 3.2, $M$ is $\left(2\eta\sqrt{\ln\left(\frac{4k}{\delta}\right)}, \frac{\delta}{2}\right)$-posterior accurate.

Optimizing over $\eta$ we get that $\eta = (122)^{\frac{1}{4}}\sigma\sqrt{\frac{\sqrt{k}}{n}}$. Combining this with the bound on $k$ implies $\frac{n^3\eta^3}{k^{\frac{3}{2}}\sigma^3} = \frac{3(122)^{\frac{1}{4}}n^{\frac{3}{2}}}{\sqrt{2}k^{\frac{3}{4}}} \leq 2k^{\frac{3}{4}} \leq 2k$. Plugging the definition of $\eta$ back in we get that $M$ is $\left(\sigma\sqrt{\frac{\sqrt{488k}}{n}}\ln\left(\frac{4k}{\delta}\right), \frac{\delta}{2}\right)$-Bayes stable and posterior accurate.

Using the bound on $n$ we get that $M$ is $\left(\frac{\epsilon}{2}, \frac{\delta}{2}\right)$-Bayes stable and posterior accurate. Combining this with Theorem 3.4 completes the proof. $\qquad\square$

## E.2 Generalization guarantees, sub-Gaussian case

To prove the generalization guarantees in the sub-Gaussian case, we must first introduce the notion of sub-exponential random variables, and recall several folklore facts about sub-exponential random variables.

**Definition E.5.** Given $\sigma, \alpha \geq 0$, a random variable $X$ is called $(\alpha, \sigma^2)$-sub-exponential if for all $\lambda \in \left[-\frac{1}{\alpha}, \frac{1}{\alpha}\right]$ we have $\mathbb{E}\left[e^{\lambda X}\right] \leq e^{\frac{\lambda^2\sigma^2}{2}}$.

**Fact E.6.** *Given* $\sigma \geq 0$ *and a* $\sigma^2$*-sub-Gaussian random variable* $X$*, the random variable* $Y := X^2 - \mathbb{E}\left[X'^2\right]$ *is* $\left(4\sigma^2, 64\sigma^4\right)$*-sub-exponential.*

**Fact E.7.** *Given* $\sigma_i, \geq 0$ *and independent* $\sigma_i^2$*-sub-Gaussian random variables* $X_i$ *for* $i \in \{0, 1\}$*, the random variable* $Y := X_0 + X_1$ *is* $\left(0, \sigma_0^2 + \sigma_1^2\right)$*-sub-Gaussian.*

**Claim E.8.** *Given* $k \in \mathbb{N}$*;* $\eta, \sigma > 0$ *and a view* $v \in \mathcal{V}$ *consisting of responses to* $k$ $\sigma^2$*-sub-Gaussian linear queries, produced by a Gaussian mechanism* $M$ *with noise parameter* $\eta$ *which receives an iid sampled dataset of size* $n$*, then for any* $1 \leq \alpha \leq \frac{n^2\eta^2}{4k\sigma^2}$ *we have*

$$\mathbb{E}_{X\sim D}\left[e^{\alpha\varphi(X;v)}\right] \leq \mathbb{E}_{X,Y\sim D}\left[e^{\alpha\varphi(X,Y;v)}\right] \leq \exp\left(\frac{9\alpha k\sigma^2}{n^2\eta^2}\right),$$

*where* $\varphi(x; v)$ *is as was defined in Theorem 4.6.*

*Proof.* From Fact E.7 and the assumption, the random variable $q(X) - q(Y)$ is a $\left(0, 2\sigma^2\right)$-sub-Gaussian random variable. Combining this with Fact E.6 we get that the random variable $(q(X) - q(Y))^2 - 2\sigma^2$ is a $\left(8\sigma^2, 256\sigma^4\right)$-sub-exponential random variable, where we rely on the fact that

$$\begin{aligned}
\mathbb{E}_{X,Y\sim D}\left[(q(X) - q(Y))^2\right] &= \mathbb{E}_{X,Y\sim D}\left[((q(X) - q(D)) + (q(D) - q(Y)))^2\right] \\
&\overset{(a)}{=} \mathbb{E}_{X\sim D}\left[(q(X) - q(D))^2\right] + \mathbb{E}_{Y\sim D}\left[(q(D) - q(Y))^2\right] \\
&\quad + 2\mathbb{E}_{X\sim D}\left[q(X) - q(D)\right] \cdot \mathbb{E}_{Y\sim D}\left[q(D) - q(Y)\right] \\
&\overset{(b)}{\leq} 2\sigma^2
\end{aligned}$$

where (a) results from the fact $X, Y$ are independent and (b) from the fact that $q(X)$ is a random variable with expectation $q(D)$ and variance proxy $\sigma^2$ which upper bounds its variance.

Using this we get

$$\mathop{\mathbb{E}}_{X \sim D}\left[e^{\alpha\varphi(X;v)}\right] \overset{(a)}{\leq} \mathop{\mathbb{E}}_{X,Y \sim D}\left[e^{\alpha\varphi(X,Y;v)}\right]$$

$$= \mathop{\mathbb{E}}_{X,Y \sim D}\left[\exp\left(\frac{\alpha}{2n^2\eta^2}\sum_{i=1}^{k}(q_i(X) - q_i(Y))^2\right)\right]$$

$$= e^{\frac{\alpha k\sigma^2}{n^2\eta^2}}\mathop{\mathbb{E}}_{X,Y \sim D}\left[\exp\left(\frac{\alpha}{2n^2\eta^2}\sum_{i=1}^{k}\left((q_i(X) - q_i(Y))^2 - 2\sigma^2\right)\right)\right]$$

$$= e^{\frac{\alpha k\sigma^2}{n^2\eta^2}}\mathop{\mathbb{E}}_{X,Y \sim D}\left[\prod_{i=1}^{k}\exp\left(\frac{\alpha}{2n^2\eta^2}\left((q_i(X) - q_i(Y))^2 - 2\sigma^2\right)\right)\right]$$

$$\overset{(b)}{\leq} e^{\frac{\alpha k\sigma^2}{n^2\eta^2}}\prod_{i=1}^{k}\left(\mathop{\mathbb{E}}_{X,Y \sim D}\left[\exp\left(\frac{\alpha k}{2n^2\eta^2}\left((q_i(X) - q_i(Y))^2 - 2\sigma^2\right)\right)\right]\right)^{\frac{1}{k}}$$

$$\overset{(c)}{\leq} e^{\frac{\alpha k\sigma^2}{n^2\eta^2}}\prod_{i=1}^{k}\left(\exp\left(\frac{32\alpha^2 k^2\sigma^4}{n^4\eta^4}\right)\right)^{\frac{1}{k}}$$

$$\overset{(d)}{\leq} e^{\frac{\alpha k\sigma^2}{n^2\eta^2}}\prod_{i=1}^{k}\left(\exp\left(\frac{8\alpha k\sigma^2}{n^2\eta^2}\right)\right)^{\frac{1}{k}}$$

$$= \exp\left(\frac{9\alpha k\sigma^2}{n^2\eta^2}\right)$$

where (a) results from combining the assumption $\alpha \geq 1$ with the Claim E.1, (b) from the generalized version of Hölder's inequality, (c) from the fact $(q(X) - q(Y))^2 - 2\sigma^2$ is a $(8\sigma^2, 256\sigma^4)$-sub-exponential random variable and the assumption $\alpha \leq \frac{n^2\eta^2}{4k\sigma^2}$, and (d) from the the assumption $\alpha \leq \frac{n^2\eta^2}{4k\sigma^2}$. □

**Lemma E.9.** *Given* $k \in \mathbb{N}$; $\sigma, \eta > 0$; $0 < \delta \leq \frac{1}{e}$; *and an analyst $A$ issuing $k$ $\sigma^2$-sub-Gaussian linear queries, if $M$ is a Gaussian mechanism with noise parameter $\eta$ that receives an iid dataset of size* $n \geq \frac{23\sqrt{k}\sigma}{\eta}\sqrt{\ln\left(\frac{1}{\delta}\right)}$, *then $M$ is* $\left(\frac{\sigma^2}{n\eta}\sqrt{976k\ln\left(\frac{1}{\delta}\right)}, \frac{n^3\eta^3\delta}{k^{\frac{3}{2}}\sigma^3}\right)$-*Bayes stable.*

*Proof.* Setting $\phi := \frac{9k\sigma^2}{n^2\eta^2}$ we get

$$\mathop{\mathbb{E}}_{X \sim D}\left[e^{27\ln\left(\frac{1}{\delta}\right)(\varphi(X;v)+\phi)}\right] \overset{(a)}{=} \exp\left(\frac{243k\sigma^2}{n^2\eta^2}\ln\left(\frac{1}{\delta}\right)\right)\mathop{\mathbb{E}}_{X \sim D}\left[e^{27\ln\left(\frac{1}{\delta}\right)\varphi(X;v)}\right]$$

$$\overset{(b)}{\leq} \exp\left(\frac{243k\sigma^2}{n^2\eta^2}\ln\left(\frac{1}{\delta}\right) + \frac{243k\sigma^2}{n^2\eta^2}\ln\left(\frac{1}{\delta}\right)\right) \qquad (7)$$

$$= \exp\left(\frac{486k\sigma^2}{n^2\eta^2}\ln\left(\frac{1}{\delta}\right)\right)$$

$$\overset{(c)}{\leq} 1 + \frac{972k\sigma^2}{n^2\eta^2}\ln\left(\frac{1}{\delta}\right)$$

where (a) results from the definition of $\phi$, (b) from Claim E.8 where the condition on $\alpha$ is fulfilled by the assumption that $n \geq \frac{23\sqrt{k}\sigma}{\eta}\sqrt{\ln\left(\frac{1}{\delta}\right)}$, and (c) from the inequality $e^x \leq 1 + 2x$ for any $x \leq 1$ where the condition on $x$ is fulfilled by the assumption that $n \geq \frac{23\sqrt{k}\sigma}{\eta}\sqrt{\ln\left(\frac{1}{\delta}\right)}$.

Next, we notice that

$$\mathop{\mathbb{E}}_{X\sim D, V\sim D}\left[\varphi\left(X;V\right)\right] = \mathop{\mathbb{E}}_{X\sim D, V\sim D}\left[\ln\left(\mathop{\mathbb{E}}_{Y\sim D}\left[e^{\varphi(X,Y;V)}\right]\right)\right]$$

$$\overset{(a)}{\leq} \ln\left(\mathop{\mathbb{E}}_{X,Y\sim D, V\sim D}\left[e^{\varphi(X,Y;V)}\right]\right)$$

$$\overset{(b)}{\leq} \ln\left(\exp\left(\frac{9k\sigma^2}{n^2\eta^2}\right)\right) \tag{8}$$

$$= \frac{9k\sigma^2}{n^2\eta^2}$$

$$\overset{(d)}{=} \phi,$$

where (a) results from Jensen's inequality for the concave function $\ln\left(x\right)$, (b) from the assumption $n \geq \frac{23\sqrt{k}\sigma}{\eta}\sqrt{\ln\left(\frac{1}{\delta}\right)}$ and Claim E.8, and (c) from the definition of $\phi$.

Similarly, for any $\alpha \leq \frac{n^2\eta^2}{4k\sigma^2}$ we have

$$\psi\left(\alpha\right) \overset{(a)}{\leq} \exp\left(\frac{9\alpha k\sigma^2}{n^2\eta^2}\right) \overset{(b)}{\leq} \exp\left(\frac{9\alpha}{529}\right), \tag{9}$$

where (a) results from the assumption on $\alpha$ and Claim E.8 and (b) from the fact that $n \geq \frac{23\sqrt{k}\sigma}{\eta}\sqrt{\ln\left(\frac{1}{\delta}\right)}$.

Setting $\xi := \frac{6k\sigma^2}{n^2\eta^2}$ we get

$$\mathop{\Pr}_{\substack{S\sim D^{(n)}\\ V\sim M(S,A), Q\sim A(V)}}\left[\left|Q\left(D_{\mathcal{X}}^v\right) - Q\left(D_{\mathcal{X}}\right)\right| > \frac{\sigma^2}{n\eta}\sqrt{976k\ln\left(\frac{1}{\delta}\right)}\right]$$

$$\overset{(a)}{\leq} \mathop{\Pr}_{\substack{S\sim D^{(n)}\\ V\sim M(S,A), Q\sim A(V)}}\left[\left|Q\left(D_{\mathcal{X}}^v\right) - Q\left(D_{\mathcal{X}}\right)\right| > \sigma_Q\sqrt{\frac{976k\sigma^2}{n^2\eta^2}\ln\left(\frac{1}{\delta}\right)}\right]$$

$$\overset{(b)}{\leq} \mathop{\Pr}_{\substack{S\sim D^{(n)}\\ V\sim M(S,A), Q\sim A(V)}}\left[\left|Q\left(D_{\mathcal{X}}^v\right) - Q\left(D_{\mathcal{X}}\right)\right| > \sigma_Q\sqrt{162\left(\frac{9k\sigma^2}{n^2\eta^2} + \frac{9k\sigma^2}{n^2\eta^2}\right)\ln\left(\frac{1}{\delta}\right) + \frac{6k\sigma^2}{n^2\eta^2}}\right]$$

$$\overset{(c)}{=} \mathop{\Pr}_{\substack{S\sim D^{(n)}\\ V\sim M(S,A), Q\sim A(V)}}\left[\left|Q\left(D_{\mathcal{X}}^v\right) - Q\left(D_{\mathcal{X}}\right)\right| > \sigma_Q\sqrt{162\left(\frac{9k\sigma^2}{n^2\eta^2} + \phi\right)\ln\left(\frac{1}{\delta}\right) + \xi}\right]$$

$$\overset{(d)}{\leq} \mathop{\Pr}_{\substack{S\sim D^{(n)}\\ V\sim M(S,A), Q\sim A(V)}}\left[\mathop{\mathbb{E}}_{X\sim D}\left[e^{27\ln\left(\frac{1}{\delta}\right)(\varphi(X;v)+\phi)}\right] > 1 + 54\left(\frac{9k\sigma^2}{n^2\eta^2} + \phi\right)\ln\left(\frac{1}{\delta}\right)\right]$$

$$+ \frac{\delta}{\xi}\left(4\psi\left(12\right)\right)^{\frac{1}{16}}\sqrt{\left(\sqrt{\psi\left(56\right)} + 1\right)\frac{e^\phi+1}{e^\phi-1}}$$

$$\overset{(e)}{\leq} \frac{\delta}{\xi}\left(4\psi\left(12\right)\right)^{\frac{1}{16}}\sqrt{\left(\sqrt{\psi\left(56\right)} + 1\right)\frac{e^\phi+1}{e^\phi-1}}$$

$$\overset{(f)}{\leq} \frac{\delta}{\xi}e^{\frac{27}{2116}}\sqrt{\left(e^{\frac{252}{529}} + 1\right)\frac{3}{\phi}}$$

$$\leq \frac{\delta}{\xi\sqrt{\phi}}$$

$$\overset{(c)}{=} \frac{n^3\eta^3\delta}{k^{\frac{3}{2}}\sigma^3}$$

where (a) results from fact that the variance of all queries issued by $A$ is bounded by $\sigma^2$, (b) from the assumption that $\delta \leq \frac{1}{e}$, (c) from the definition of $\phi$ and $\xi$, (d) from Theorem D.10 where the

condition was proven in Equation 8, (e) from Equation 7, and (f) from Equation 9, and the fact that $\frac{e^x+1}{e^x-1} \leq \frac{3}{x}$ for any $0 < x < 2$. $\qquad\square$

**Theorem E.10** (Exact version of Theorem 5.3). *Given $k \in \mathbb{N}$; $\sigma, \epsilon \geq 0$; $0 < \delta \leq \frac{1}{e}$; and an analyst $A$ issuing $k$ $\sigma^2$-sub-Gaussian linear queries, if $M$ is a Gaussian mechanism with noise parameter $\eta = (244)^{\frac{1}{4}} \sigma \sqrt{\frac{\sqrt{k}}{n}}$ that receives an iid dataset of size $n \geq \frac{32\sqrt{k}\sigma^2}{\epsilon^2}\ln\left(\frac{2k}{\delta}\right)$, and $k \geq 2n$, then $M$ is $(\epsilon, \delta)$-distribution accurate.*

*Proof.* From Lemma E.9 setting $\delta' := \frac{\delta}{4k}$ (where $\delta'$ denotes the parameter $\delta$ used in the Lemma) we get that $M$ is $\left(\frac{\sigma^2}{n\eta}\sqrt{976k\ln\left(\frac{2k}{\delta}\right)}, \frac{n^3\eta^3\delta}{2k^{\frac{5}{2}}\sigma^3}\right)$-Bayes stable. From Lemma 3.2, $M$ is $\left(2\eta\sqrt{\ln\left(\frac{4k}{\delta}\right)}, \frac{\delta}{2}\right)$ Posterior accurate.

Optimizing over $\eta$ we get that $\eta = (244)^{\frac{1}{4}} \sigma \sqrt{\frac{\sqrt{k}}{n}}$. Combining this with the bound on $k$ implies $\frac{n^3\eta^3}{k^{\frac{3}{2}}\sigma^3} = \frac{(224)^{\frac{1}{4}}n^{\frac{3}{2}}}{k^{\frac{3}{4}}} \leq 2k^{\frac{3}{4}} \leq 2k$. Plugging the definition of $\eta$ back in we get that $M$ is $\left(\sigma\sqrt{\frac{\sqrt{976k}}{n}\ln\left(\frac{4k}{\delta}\right)}, \frac{\delta}{2}\right)$-Bayes stable and Posterior accurate.

Using the bound on $n$ we get that $M$ is $\left(\frac{\epsilon}{2}, \frac{\delta}{2}\right)$-Bayes stable and Posterior accurate. Combining this with Theorem 3.4 completes the proof. $\qquad\square$

### E.3 Different or incorrect variances

In this section we discuss in detail the case of a mechanism receiving queries with different variances, and then extend the discussion to the effects of analyst relying upon an incorrect bound on the variance of one of the queries.

In the case of a Gaussian mechanism with queries $q_i$, each with variance bounded by $\sigma_i^2$, the results we achieved translate to a guarantee that with probability $\geq 1 - \delta$, for any $i \in k$, we have

$$|q_i(D) - r_i| \leq O\left(\eta_i\sqrt{\ln\left(\frac{1}{\delta}\right)} + \sigma_i\sqrt{\ln\left(\frac{1}{\delta}\right)\sum_{j=1}^{k}\frac{\sigma_j^2}{n^2\eta_j^2}}\right),$$

where $\eta_i$ is the noise parameter chosen at each iteration.

The $\eta_i\sqrt{\ln\left(\frac{1}{\delta}\right)}$ term represents the sample / posterior accuracy bound, and the $\sigma_i\sqrt{\ln\left(\frac{1}{\delta}\right)\sum_{j=1}^{k}\frac{\sigma_j^2}{n^2\eta_j^2}}$ represents the Bayes stability. Optimizing over $\eta_i$, we get that the this error is minimized by setting $\eta_i = O\left(\sigma_i\sqrt{\frac{\sqrt{k}}{n}}\right)$, which implies

$$|q_i(D) - r_i| \leq O\left(\sigma_i\sqrt{\frac{\sqrt{k}}{n}\ln\left(\frac{1}{\delta}\right)}\right).$$

Now consider a situation where the correct variances were each bounded by $\sigma_i^2$, but the analyst mistakenly assumed a different bound $\tau_i^2$ which might be higher or lower than $\sigma_i^2$. In this case $\eta_i$ would have been set to $O\left(\tau_i\sqrt{\frac{\sqrt{k}}{n}}\right)$, so

$$|q_i(D) - r_i| \leq O\left(\eta_i\sqrt{\ln\left(\frac{1}{\delta}\right)} + \sigma_i\sqrt{\ln\left(\frac{1}{\delta}\right)\sum_{j=1}^{k}\frac{\sigma_j^2}{n^2\eta_j^2}}\right)$$

$$= O\left(\left(\frac{\tau_i}{\sigma_i} + \sqrt{\frac{1}{k}\sum_{j=1}^{k}\frac{\sigma_j^2}{\tau_j^2}}\right)\sigma_i\sqrt{\frac{\sqrt{k}}{n}\ln\left(\frac{1}{\delta}\right)}\right),$$

which is the same term as in the case of correctly estimated variances, multiplied by the term

$$\frac{\tau_i}{\sigma_i} + \sqrt{\frac{1}{k} \sum_{j=1}^{k} \frac{\sigma_j^2}{\tau_j^2}}.$$

Analyzing this term allows us to precisely understand the implications of an analyst making a mistake in the assumed variance bound.

If $\tau_i \geq \sigma_i$ for a single query $q_i$ (which means we added too much noise to the response to that query), the first term increased only for that query, and the second term decreased for all queries. On the other hand, if $\tau_i < \sigma_i$ (which means we did not add enough noise to the response to that query), the first term decreased for that query, but the second term increased for all queries.

The effect on the first term is proportional to the square root of the ratio of the wrong variance to the correct variance of that query, while the effect on the second term is proportional to the square root of average over all $k$ queries of the ratio of the correct variance to the wrong variances (notice the switch between numerator and denominator).

# F Relationships to other stability notions

Intuitively speaking, ensuring $(\epsilon, \delta)$-differential privacy means that for any query $q$ and any two neighboring datasets $s, s'$, with probability $> 1 - \delta$ over the choice of the response $r$, the log conditional probability ratio $\ln\left(\frac{D(r\,|\,s)}{D(r\,|\,s')}\right)$ is bounded by $\epsilon$ (this quantity is sometimes referred to as the *privacy loss*).[10] The most basic method for ensuring $(\epsilon, \delta)$-DP in the context of numerical queries is by randomizing the response through noise addition, calibrating the added noise to the *global sensitivity*, that is, the worst-case privacy loss over all queries and all pairs of neighboring datasets. A series of works initiated by Dwork et al. [2015] leverages bounds on the privacy loss to provide generalization guarantees for mean estimation under adaptive data analysis. Bassily et al. [2021] asymptotically improved this guarantee and extended it to low-sensitivity queries and optimization queries.

Differential privacy's dependence on the global sensitivity naturally yields generalization guarantees that scale with the range of the query's values. However, a dependence on the range of the queries is not required to protect against overfitting in the non-adaptive setting, where the generalization guarantees achievable via Bernstein's inequality scale with the queries' standard deviation. The worst-case nature of DP is not a bug but a feature, since privacy must be ensured for *all* participants under *any* circumstances against *every* potential attack. On the contrary, the task of statistical estimation focuses only on the *typical* case, even in the non-adaptive setting. To overcome the dependence on the queries' range, we seek to decouple DP's stability-preserving powers from its dependence on global sensitivity. This can be done by, in a sense, considering the distribution over the *local sensitivity* induced by the *underlying distribution* over the data, as we discuss in the next sections.

## F.1 Scaling back the noise

One approach one might consider to attempt to circumvent DP's worst-case nature in the context of adaptive data analysis is by adding noise that scales like the *local sensitivity* of each query, that is, the privacy loss given the specific query, the specific dataset under study, and the worst-case neighbor of that specific dataset. A variant of this notion was first introduced by Nissim et al. [2007], who illustrated that if the magnitude of noise addition depends naïvely on the local sensitivity, then the chosen magnitude "leaks information" about the dataset, resulting in a mechanism that is not differentially private. To address this, they introduce the *smoothed sensitivity* notion, which smooths out the addition of locally-tailored noise, achieving improved DP guarantees for the task of median estimation (or any other quantile estimation), but not for mean estimation.

Several works have leveraged this result to improve private mean estimation as well, using more involved mechanisms. For example, Feldman and Steinke [2017] propose an algorithm that splits a dataset into $m$ subsets, estimates the empirical mean over each one in a non-private manner, and then privately reports the median of the means. They prove that this mechanism is still DP, and provide accuracy guarantees that scale with the queries' standard deviation. Another line of work uses truncation-based specialized mechanisms to provide differential privacy guarantees for Gaussian and sub-Gaussian queries, even in the case of multivariate distribution unknown covariance [Karwa and Vadhan, 2018, Ashtiani and Liaw, 2022, Duchi et al., 2023]. Feldman and Steinke [2018] use a more natural mechanism for the purpose of generalization under adaptivity, which adds noise that scales like the empirical standard deviation of the query. The generalization guarantees they prove scale like the true standard deviation, but apply only to the expected error.

An alternative method for avoiding the privacy loss that results from scaling the added noise addition to parameters of the actual dataset is keeping the noise level constant, and instead using local sensitivity only when analyzing the mechanism's accuracy and privacy/stability guarantees. This line of work was initiated by Ghosh and Roth [2011] and later extended by Ebadi et al. [2015], who use similar notions to provide privacy accounting for each individual in the dataset, but do not improve generalization (or privacy) guarantees in the general case. For a survey of this type of analysis, see Wang [2019]. One notable exception is a work by Feldman and Zrnic [2021], who provide improved guarantees under sparsity assumptions on the data—that is, assuming every sample element is relevant only to a small portion of the issued queries.

---

[10]This intuition is formalized with small degradation in the $\epsilon$ and $\delta$ terms in Lemma 3.3.2 of Kasiviswanathan and Smith [2014].

## F.2 Accounting for the underlying distribution

The degradation in the guaranteed generalization capabilities caused by scaling the added noise to the global sensitivity of a query might be also mitigated by considering the "natural noise" induced by sampling the dataset from some underlying distribution. A variety of relaxations of the DP definition in this vein were introduced by Gehrke et al. [2012] and Bassily et al. [2013]. Bhaskar et al. [2011] even consider an alternative privacy notion that relies solely on noise from sampling without any noise added by the mechanism, but all of these approaches suffer from a similar problem: while the privacy guarantee of a single query might improve due to noise introduced by sampling, these definitions have little to no adaptive composition guarantees. A key step in DP's composition proof is the identity $\ln\left(\frac{D(r_k\,|\,s,v_{k-1})}{D(r_k\,|\,s',v_{k-1})}\right) = \ln\left(\frac{D(r_k\,|\,s)}{D(r_k\,|\,s')}\right)$, which relies on the guarantee that the mechanism's response is stable, given any auxiliary information (as discussed in the proof of Theorem 4.4). This guarantee can hold only if we condition on the full dataset, which one cannot do when using a definition that leverages the sampling of the data.

Failure to compose—or to compose well—is a common issue with alternative notions of stability that attempt to weaken DP. Shenfeld and Ligett [2019] propose a stability notion that is both necessary and sufficient for adaptive generalization (under certain assumptions), but it suffers from limited composition guarantees. Triastcyn and Faltings [2020] propose the notion of *Bayesian differential privacy*, which does not enjoy advanced composition guarantees. Instead they prove that the generalization guarantees implied by this notion do compose, but their generalization guarantees still scale with the queries' range in the general case. A notable exception to these composition limitations is the notion of *Typical stability* introduced by Bassily and Freund [2016]; their approach only requires that the stability notion hold with respect to a subset the of possible datasets, so long as the probability of drawing a dataset outside the set is negligible. Using this notion, they provide generalization guarantees that extend beyond datasets where each element is drawn independently and bounded linear queries, and they obtain guarantees that scale with the variance proxy in case of sub-Gaussian distributions. Unfortunately, this notion only provides accuracy guarantees for a number of queries that scales linearly with the sample size, rather than the quadratic guarantee achieved using DP in the iid case.

Considering the underlying data distribution also paves the way for a Bayesian interpretation of stability. This perspective actually dates back to the work that introduced differential privacy. Dwork et al. [2006] interpret the bound on the privacy loss as a bound on the change in the a posteriori probability belief of an analyst, over the sample set held by the mechanism. This interpretation was later formalized by Kasiviswanathan and Smith [2014], who prove that DP is essentially equivalent to a bound on the Total Variation distance (Definition B.1) between the two possible posteriors. Later, Jung et al. [2020] use the posterior distribution induced by the transcript of the interaction between the mechanism and the analyst as a key component in their analysis of DP's generalization guarantees under adaptivity, which forms the basis for the ideas presented in this paper.

Some works consider a true Bayesian setting, where the analyst and/or the mechanism hold a prior over possible data distributions. Machanavajjhala et al. [2009] and Li et al. [2013] use such an assumption to provide stronger privacy guarantees under the assumption that the analyst's knowledge about the distribution is limited. Yang et al. [2015] use this setting to account for non iid data distributions, where knowing of one element in the dataset changes the probability that another element is included in it, thus degrading the privacy of other elements. In the context of adaptive data analysis, Elder [2016] considered a more realistic setting, which removes the gap between the knowledge of the analyst and the mechanism regarding the underlying data distribution. Unfortunately, any assumption about the analyst's knowledge is hard to justify in many realistic scenarios, where implicit priors might be embedded in her algorithm. The guarantees we provide hold even in the case of a fixed distribution which is known to the analyst.

## F.3 Our approach

In this work we combine these two approaches, leveraging the potential improved stability resulting from low local sensitivity and the properties of the underlying distribution, to avoid the aforementioned potential pitfalls. We do so by introducing a localized stability notion that depends on the specific dataset(s) and on the queries issued during the interaction. Using this notion we can leverage the distribution over the local sensitivity induced by the underlying distribution over the domain. By

considering the local sensitivity only in the analysis stage, we avoid the induced stability loss that might be caused, had we scaled the noise with the local sensitivity. Similarly, by choosing a stability notion that depends on the full dataset(s), saving the underlying distribution for the analysis stage, we guarantee that this stability notion composes well.

By transitioning from a global stability notion to a local one, we provide high-probability generalization guarantees that scale with the queries' standard deviation. Furthermore, by removing the dependency on the range, our guarantees extend to unbounded queries as well, so long as they are well-concentrated. Unlike previous works, both of these advantages are achieved even for simple noise-addition mechanisms.

Moving our focus from an a priori stability guarantee to an a posteriori guarantee provided after the fact is similar to the notion of *ex-post privacy* presented by Ligett et al. [2017]. This approach can also be viewed as a variant of the *metric differential privacy* (mDP) notion [Chatzikokolakis et al., 2013, Imola et al., 2022], for the purpose of adaptive data analysis. Unlike mDP, our notion does not rely on some external fixed metric over elements domain, but rather uses a separate metric at each iteration which is induced by the corresponding query.

The dependence of the stability function on the issued queries also introduces a new challenge. Since these queries are adaptively chosen, the sensitivity is not predefined even by fixing the two datasets, and, in fact, is a random variable. This challenge has been considered with different motivation under the name *full adaptivity*, first by Rogers et al. [2016] and later by Whitehouse et al. [2023]. In their framework, the sensitivity is fixed, but the privacy parameters are adaptively chosen. The dependence of our PC definition on the actual issued queries during the interaction is analogous to their *privacy odometer*.

Although differential privacy is defined with respect to given $\epsilon$ and $\delta$ parameters, most mechanisms in fact establish a relationship between the two, where each $\epsilon$ has a corresponding $\delta$, and vice-versa (see Zhu et al. [2022] for a survey of possible representations of this relationship). [M: Be aware of the order things are presented] Mironov [2017] provided a simple representation of this relation, by introducing Rényi DP (Definition B.16), which replaces max divergence (Definition B.4) by Rényi divergence (Definition B.5) as the stability measure. Building on this notion, Bun and Steinke [2016] introduced a new privacy definition which they call *zero-concentration differential privacy (zCDP)* (Definition B.18), and prove that it implies a relationship between the $\epsilon$ and $\delta$ parameters for DP. This definition can also be viewed as a sub-Gaussian bound on the distribution of the privacy loss (Definition 4.1), in the style of *Concentrated differential privacy* introduced by Dwork and Rothblum [2016] (Definition B.17). Our PC stability notion (Definition 4.2) can be viewed as a "localized" version of zCDP, replacing the divergence by a dissimilarity measure as discussed in Appendix B, but the same technique could be used to creating a "localized" version of DP as well (Definition B.8). Triastcyn and Faltings [2020] also propose a notion similar to PC, but they do not leverage it to achieve variance-dependent accuracy guarantees.

Although we have managed to transition from worst-case to typical-case guarantees by considering the issued queries and chosen sample set, the guarantees we provide still protect against a worst-case analyst. Using the Cauchy-Schwarz inequality in the proof of the Covariance Lemma (Lemma 3.5), we bound the stability loss even for queries that are adversarially correlated with the previous view to increase the chance of overfitting. This might be viewed as overkill for a real-world setting that is typically non-adversarial. One possible extension for future work could be to consider a version of "natural analysts", a notion introduced by Feldman and Zrnic [2021], who improve generalization guarantees for such analysts. Alternatively, Du and Dwork [2022] consider a setting where, in each iteration, the issued query is related to a different covariant of the data elements, which are weakly correlated. Although we do not consider this type of relaxation in the present work, the tools we develop here might be used to give improved guarantees under similar assumptions.