# OpenReview forum: "Generalization in the Face of Adaptivity: A Bayesian Perspective"
_NeurIPS.cc/2023/Conference — NeurIPS 2023 spotlight_

### Official Review · Reviewer_WLLe · 2023-06-29

**Soundness:** 3 good
**Presentation:** 3 good
**Contribution:** 3 good
**Rating:** 7
**Confidence:** 4

**Summary:**

This paper explores the problem of adaptive data analysis - when a single dataset is repeatedly used for adaptively chosen queries, overfitting can occur rapidly. To reduce this bias, a popular approach is to add noise to the output of each query. Intuitively, this prevents the analyst from learning too much about the sample and thus prevents them from choosing a query that overfits the sample.

The most common technique used to analyze this adaptive setting is differential privacy. However, the worst-case nature of DP requires scaling the noise to a worst-case dataset, rather than a typical dataset. The paper's contributions are:

1) Prior work has shown that (roughly speaking), to ensure that the query responses are low bias, it suffices to show the responses have "posterior accuracy." This paper shows that posterior accuracy can be thought of as the correlation between the query asked and a Bayes factor.

2) They introduce a new notion of stability (pairwise concentration). Roughly speaking, stability measures (like DP) measure how much an algorithm's output depends on its input and have long been used to analyze mechanisms for adaptive data analysis. Pairwise concentration crucially depends on the dataset and query, to avoid the worst-case requirements of DP. Using this new notion of stability, they bounded the variance of the Bayes factor introduced in the previous point, which in particular bounds the correlation between it and the query asked. As a result, this new notion of stability can be used to prove generalization guarantees.

3) Using pairwise concentration, they show that it suffices to add noise that scales with the standard deviation of each query.

For 3), a similar result was shown by [Feldman and Steinke 18]. However, [FS 18's] result is weaker than this paper in two ways. First, it only holds in expectation over the error, whereas this paper's guarantee holds with extremely high probability (the difference is equivalent to Markov's  vs Chernoff-style bounds). Second, this paper can handle subgaussian queries with unbounded ranges, whereas it's unclear if [FS 18] can handle unbounded ranges.

**Strengths:**

Within the field of adaptive data analysis, the authors give a refined answer to a simple question: What happens if we scale the noise of our answers to the standard deviation of the query? Prior work of [FS 18] suggested this approach can work, but were only able to prove that the error is bounded in expectation. This paper shows the error is bounded with high probability and also extends to unbounded but subgaussian queries.

I'm also excited by the new notion of stability (pairwise concentration) introduced in this paper and hopeful that it can be applied in other settings (for instant, to analyze other mechanisms).

Along the way, the authors introduce a number of perspectives and new tools that may be more broadly applicable. For example, they introduce a number of "dissimilarity" measures, which provide more expressive ways to measure how dissimilar two distributions are than classical notions of divergence.

**Weaknesses:**

The definition of pairwise concentration and its application (in Theorem 4.5) are technically complex. One way for this work to be impactful is for pairwise concentration to be applied more broadly, and providing a simple interface to it would aid that. For example:
1) In Theorem 4.5, I believe you only ever need to bound the pairwise concentration by datasets that differ in one point (via the function $\psi(x,y;v)$ defined on line 291). If so, I would explicitly state that (ideally early) as this would make it easier for someone else to apply your framework.
2) Also in Theorem 4.5, is there ever a reason (up to a small constant) to not set $\xi = \epsilon^2$?  It seems to me the probability bound is monotonically decreasing in both $\xi$ and $\epsilon$, in which case setting both $\xi$ and $\epsilon^2$ to the maximum of the two will only improve the error probability, and affect the deviation bounded by at most a factor of $\sqrt{2}$. If so, my (personal) preference is that it's better to state a simpler version losing that $\sqrt{2}$ and defer more detailed versions to the appendix.

Another minor point: This paper's mechanism assumes it knows the standard deviation of each query. The work of [FS 18] is able to estimate the standard deviation of each query and scale the noise on the fly. Can your analysis handle a similar approach?

**Questions:**

I'm a bit confused about the regime where $\alpha < 1$ in Definition 4.2 and Fact B.7. In Fact B.7, I believe the $\leq$ should be converted to a $\geq$ in this setting because the righthand side becomes decreasing with $\phi$. For Definition 4.2, do you similarly want the direction to depend on whether $(\alpha - 1)$ is positive? That would align with the intuition that $\varphi$ is an upper bound on some notion of ``distance" between $s$ and $s'$. The way it's written, we need $\varphi$ to be an upper bound on this distance when $\alpha > 1$ and a lower bound when $0 \leq \alpha < 1$.

Line 731 typo "ration"-> "ratio"

See also the questions from the Weaknesses section.

**Limitations:**

Yes - all assumptions/theorem statements are clear, and the discussion includes some future directions that the present work does not address.

---

> ### Author Rebuttal · Authors · 2023-08-09
>
> We thank the reviewer for the useful feedback.
>
> We appreciate the suggestions on how to make the presentation of pairwise concentration more accessible. We felt a tension between simplicity versus presenting the more general notion, which may have broader consequences and applications beyond the usage in this paper (for example, group stability becomes a direct result of the definition, not requiring any proof). Given your comments, we will more prominently highlight that this paper uses only the $\varphi(x, y;v)$ notion, where datasets differ in only one point. We'll also take your suggestion to set $\xi = \epsilon^2$ to simplify the statement of Theorem 4.5.
>
> The case of unknown variance is not covered in this work, but we hope that the pairwise concentration stability notion will also offer new insights for this case, in future work.
>
> The sign for the $0 \le \alpha \le 1$ regime is not wrong, though it is somewhat counter-intuitive. It may help to note that $\varphi$ serves two roles in this setting: (1) a bound on the stability loss random variable's mean (the $-\alpha$ part), and (2) the variance proxy of the stability loss's distribution (the $\alpha^{2}$ part). In the CDP representation [1], these two parts are denoted by $\mu$ and $\sigma^{2}$, respectively. Using this notation, we see that the bound is indeed monotonically tighter as $\sigma^{2}$ decreases, as expected.
>
> There was a typo in Fact B.7: it holds only for the range $\alpha \ge 1$ (we only need it for that range).
>
> ~
>
> [1] Dwork, Cynthia, and Guy N. Rothblum. "Concentrated differential privacy."

---

> > ### Comment · Reviewer_WLLe · 2023-08-11
> > **Response to rebuttal + additional question**
> >
> > I thank the authors for this response and am satisfied with the answers.
> >
> > One followup question: Suppose the analyst gives what the mechanism gives what they think is an upper bound on the query's variance, but they are wrong and the variance is actually larger. How does this affect the answer to future queries? Can I think of one underestimated variance (say by a factor of 2) contributing the same amount of stability loss as a constant number of queries with the correct variance? In that case, the occasional underestimated variance seems ok. Or, does it become difficult to give bounds when a single query has bad variance?

---

> > > ### Author Response · Authors · 2023-08-12
> > >
> > > Thanks for the great question! Yes, our results allow us to reason precisely about the impact of an analyst using an incorrect variance bound.
> > >
> > > First we note that our results can be extended to the case of varying variances, by adding noise at each iteration that scales with the corresponding variance, as we mentioned in the response to reviewer KJvw.
> > >
> > > In the case of a Gaussian mechanism with queries of variance bounded by $\sigma_{i}^{2}$, our results translate to a guarantee that with probability $\ge1-\delta$, for any $i\in\left[k\right]$, the quantity
> > >
> > > $$
> > > \left|q_{i}\left(D\right)-r_{i}\right|\le O\left(\eta_{i}\sqrt{\ln\left(1/\delta\right)}+\sigma_{i}\sqrt{\ln\left(1/\delta\right)\sum_{j=1}^{k}\frac{\sigma_{j}^{2}}{n^{2}\eta_{j}^{2}}}\right),
> > > $$
> > > where $\eta_{i}$ is the noise parameter chosen at each iteration. The first term $\left(\eta_{i}\sqrt{\ln\left(1/\delta\right)}\right)$ represents the sample / posterior accuracy bound, and the second $\left(\sigma_{i}\sqrt{\ln\left(1/\delta\right)\sum_{j=1}^{k}\frac{\sigma_{j}^{2}}{n^{2}\eta_{j}^{2}}}\right)$ represents the the Bayes stability. Optimizing over $\eta_{i}$ we get that this term is minimized by $\eta_{i}=O\left(\sigma_{i}\sqrt{\frac{\sqrt{k}}{n}}\right)$, which implies
> > > $$
> > > \left|q_{i}\left(D\right)-r_{i}\right|\le O\left(\sigma_{i}\sqrt{\frac{\sqrt{k}}{n}\ln\left(1/\delta\right)}\right).
> > > $$
> > >
> > > Now consider a situation where the true variances were bounded by $\sigma_{i}^{2}$, but an analyst mistakenly assumed different bounds
> > > $\tau_{i}^{2}$ which might be higher or lower than $\sigma_{i}^{2}$. In this case the analyst would have used noise parameters $\eta_{i}=O\left(\tau_{i}\sqrt{\frac{\sqrt{k}}{n}}\right)$, so
> > > \begin{align*}
> > > \left|q_{i}\left(D\right)-r_{i}\right|\le & O\left(\tau_{i}\sqrt{\frac{\sqrt{k}}{n}\ln\left(1/\delta\right)}+\sigma_{i}\sqrt{\ln\left(1/\delta\right)\sum_{j=1}^{k}\frac{\sigma_{j}^{2}}{n\sqrt{k}\tau_{j}^{2}}}\right)\\\\
> > > = & O\left(\left(\frac{\tau_{i}}{\sigma_{i}}+\sqrt{\frac{1}{k}\sum_{j=1}^{k}\frac{\sigma_{j}^{2}}{\tau_{j}^{2}}}\right)\sigma_{i}\sqrt{\frac{\sqrt{k}}{n}\ln\left(1/\delta\right)}\right),
> > > \end{align*}
> > > which is the same term as in the case of correctly estimated variances, multiplied by the term $\frac{\tau_{i}}{\sigma_{i}}+\sqrt{\frac{1}{k}\sum_{j=1}^{k}\frac{\sigma_{j}^{2}}{\tau_{j}^{2}}}$.
> > >
> > > Analyzing this term allows us to precisely understand the implications of an analyst making a mistake in the assumed variance bound.
> > >
> > > If $\tau_{i}\ge\sigma_{i}$ for a single query $q_{I}$ (which means we added too much noise to the response to that query), the first term increased only for that query, and the second term decreased for all queries.
> > > On the other hand, if $\tau_{i}<\sigma_{i}$ (which means we did not add enough noise to the response to that query), the first term decreased for that query, but the second term increased
> > > for all queries.
> > >
> > > The effect on the first term is proportional to the square root of the ratio of wrong to correct variance of that query, while the effect on the second term is proportional to the square root of average over all $k$ queries of the ratio of correct to wrong variances (notice the switch between numerator and denominator).

---

### Official Review · Reviewer_wAxx · 2023-07-04

**Soundness:** 3 good
**Presentation:** 3 good
**Contribution:** 3 good
**Rating:** 7
**Confidence:** 3

**Summary:**

This paper makes progress in adaptive data analysis by providing a better analysis of the Gaussian mechanism (for answering statistical/linear/counting queries), and showing that adding Gaussian noise ensures generalization error that scales with the **variance** of the queries.

Previously, the differential-privacy-based analysis achieved a similar conclusion. However, the generalization error might scale with the *range* of the queries. In practice, we might expect the queries to exhibit some "nice" behavior so that the worst-case analysis might be too pessimistic. In this sense, the contribution of this paper is valuable.

This paper is not the first one to study this question. Previously, [Feldman and Steinke, 2017] and [Feldman and Steinke 2018] considered the same question and got some results that were quantitatively weaker than the one presented in the current paper. The algorithm presented in this work is simpler than prior ones (i.e., simply adding Gaussian noises suffices), and the bound sharper.

**Strengths:**

* The algorithm is a natural and simple one. It's valuable that we can not understand the role of Gaussian noise in ADA better.
* The bound is sharp.
* The technical analysis seems novel (the so-called "covariance between the new query and a Bayes factor-based measure"), which might help advance the field further (though the reviewer did not have a chance to verify all the claims).

**Weaknesses:**

* The current algorithm only works for linear queries. i.e., queries of the form q(D) = 1/n \sum_i q(x_i). The DP-based technique can offer bounds even for low-sensitivity queries.
* I believe the "covariance" and "Bayes factor" notions are technically interesting. However, it seems that the current write-up does not fully convey an intuitive interpretation of these quantities. In particular, I feel like the claim "how much information about the data sample was encoded in the responses given to past queries" (quote from the abstract) could be made clearer.

**Questions:**

See weaknesses.

**Limitations:**

Limitations and future directions are discussed in the submission.

---

> ### Author Rebuttal · Authors · 2023-08-09
>
> We thank the reviewer for the useful feedback.
>
> The restriction to linear queries is indeed a limitation of the current work, and we hope to extend the results beyond linear queries in future work, as we briefly mention in the discussion.
>
> Thanks for your comment about wanting more discussion/intuition for the Covariance Lemma and surrounding concepts! We agree and will add it in the final version. Here are some comments in that direction:
>
> The Bayes factor $K(x, v)$ represents the likelihood ratio of seeing a particular view $v$ (a history of queries and responses) when conditioning on a particular datapoint $x$ being in the dataset, relative to when the dataset is entirely drawn at random. This captures the extent to which the presence of $x$ affects our probability of seeing $v$.
>
> The first part of the Covariance Lemma (3.5) shows that the correlation that the view induces between a query $q$ and the Bayes factor $K(\cdot, v)$ is an important quantity; it precisely controls the difference between the expectation of $q$ on the underlying data distribution versus its expectation on the posterior distribution (a distribution reflecting how a prior of the true data distribution would be updated after seeing $v$). When $q$ behaves very differently on this prior and this posterior, $q$ ``encodes'' information from the dataset, information that it received via $v$.
>
> For arbitrary analysts with unlimited prior knowledge, the inequality in the Lemma is tight. This can be observed via the variational representation of the Chi-square divergence, which implies that $\chi^{2}\left(D_{\mathcal{X}}^{v} \Vert D_{\mathcal{X}}\right) = \underset{q \in \mathcal{Q}}{\sup} \left(\frac{\left(q \left(D_{\mathcal{X}}^{v} \right) - q \left(D_{\mathcal{X}} \right) \right)^{2}}{\sigma_{q}^{2}} \right)$. In this case, the only way to avoid overfitting is by bounding the Bayes factor term, so that the prior and posterior remain close in Chi-square divergence.
>
> This Lemma offers the tantalizing suggestion that it may also be possible to obtain improved guarantees for adaptive generalization under stricter assumptions on the analyst. Such an improvement would not contradict known lower bounds, and might help us better understand the existence of algorithms whose generalization properties in practice seem to beat the known theoretical guarantees.

---

> > ### Comment · Reviewer_wAxx · 2023-08-18
> > **thank you**
> >
> > Thanks to the authors for answering my questions! I want to keep my current score.

---

### Official Review · Reviewer_KJvw · 2023-07-05

**Soundness:** 3 good
**Presentation:** 4 excellent
**Contribution:** 2 fair
**Rating:** 6
**Confidence:** 4

**Summary:**

Overfitting can occur when a single dataset is used for several statistical tasks. It is known the application of additive noise techniques from differential privacy can prevent overfitting in a variety of statistical settings. The issue is that traditional mechanisms from differential privacy, which are inherently worst-case in nature, are overly pessimistic in adapting to the true variance of queries being issued. This work shows that additive noise mechanisms, such as the Gaussian mechanism, can provide guarantees that are inherently adaptive to the variance of queries issued by a data analyst. This departs from previous works, in which the guarantees are calibrated to worst-case (sensitivity-based) characterizations of the queries. To facilitate analysis, the authors introduce several novel analytic devices based on the realized Bayes factors of the queries.

**Strengths:**

-	The paper is very well written, and the related work section is good at framing the problem considered in this paper with respect to the general research landscape.
-	This work, unlike most existing work, circumvents many aspects of worst-case analysis involved in differential privacy. Namely, through introducing several sophisticated analytical objects, the authors are able to present a more refined utility guarantee for the commonly used Gaussian mechanism.


**Weaknesses:**

-	I think the results in the paper would appear stronger if the guarantees allowed the variance of queries to differ across rounds, as will likely be the case in practical data-analytic settings. From the current wording it appears that the query variance is fixed across rounds, but perhaps I am mistaken? Perhaps in practice many queries are of low variance, but only a few are of high variance.
-	Along this line, I think experiments would be very important in demonstrating the impact in preventing overfitting. Even something along lines of a toy regression problem could make the message of the paper stronger.


**Questions:**

NA

**Limitations:**

-	The authors fairly discuss limitations as well as directions for future work in the final section of the paper.

---

> ### Author Rebuttal · Authors · 2023-08-09
>
> We thank the reviewer for the useful feedback.
>
> Regarding the first point under ``weaknesses'': Our results can handle queries of differing variances, as long as the mechanism has access to bounds on their variances. In such a case, the mechanism can scale the added noise at each iteration $\eta_{i}$ to the corresponding bound on variance $\sigma_{i}$, and the guarantees will still hold. One way to think of this is as simply rescaling all queries according to their variance bounds, adding a fixed level of noise, and then re-scaling the responses. We will clarify this  in the final version of the paper. (The case of entirely unknown variances is not covered in this work, but is an exciting direction for future work.)
>
> We did not include empirical evaluation because it is straightforward to make such an empirical evaluation look arbitrarily good, by choosing a setting where the range is much larger than the standard deviation. One plot we could consider adding would have the range-to-standard deviation ratio as the x-axis, and plot the accuracy on the y-axis for both our analysis and the standard DP analysis. We somewhat prefer to use the space for additional intuition and other details, but are happy to defer to the reviewers on this matter.

---

### Official Review · Reviewer_GhwK · 2023-07-06

**Soundness:** 3 good
**Presentation:** 3 good
**Contribution:** 2 fair
**Rating:** 5
**Confidence:** 3

**Summary:**

In this paper, the authors generalize and improve the sample complexity that scales with variance compared to the sensitivity/range in the state-of-art DP result for adaptivity guarantee based on a novel definition of pairwise concentration (PC). The paper demonstrates such generalization in both bounded queries and sub-Gaussian queries.

**Strengths:**

1. The analysis of the main results for both bounded and unbounded queries using the pairwise concentration (PC) stability is novel and has potential to analyze similar questions in the literature.

2. The main results get rid of the range $\Delta$ dataset size required for the adaptivity guarantee only depend on variance.

**Weaknesses:**

1. In theorem 5.1, the data size n can still be significantly larger than the baseline in DP due to max term and the fact that $\alpha$ is arbitrary. Specifically, when $\alpha$ increases, the dataset size n in Theorem 5.1 does not necessarily improve over the DP result, even if it does not depend on $\Delta$.

2. The paper does not provide any empirical evaluation to compare with the DP result.

**Questions:**

Can the author please provide any explanations and details on the concerns in the weakness part?

**Limitations:**

The paper does not have experiments compared to the DP baseline result. And due to the fact that the adaptivity parameter is arbitrary, I think the dataset size in the main theorem 5.1 can scale beyond the DP result when $\alpha$ is large.

---

> ### Author Rebuttal · Authors · 2023-08-09
>
> We thank the reviewer for the useful feedback.
>
> Regarding the first point under ``weaknesses'': note that the case of $\alpha \ge \frac{\Delta}{2}$ is trivial, since it can trivially be achieved by $r = \frac{1}{2} \left(\underset{x \in \mathcal{X}}{\min} \left(q(x) \right) + \underset{x \in \mathcal{X}}{\max} \left(q(x) \right) \right)$ (the mid point of the query's range), which is independent of the data. For any $\alpha < \frac{\Delta}{2}$, we obtain $\max \left( \frac{\Delta}{\alpha}, \frac{\sigma^{2}}{\alpha^{2}} \right) \le \frac{\Delta^{2}}{2 \alpha^{2}}$, and so the new bound provides improvement over the standard DP analysis for any non-trivial $\alpha$.
>
> We did not include empirical evaluation because it is straightforward to make such an empirical evaluation look arbitrarily good, by choosing a setting where the range is much larger than the standard deviation. One plot we could consider adding would have the range-to-standard deviation ratio as the x-axis, and plot the accuracy on the y-axis for both our analysis and the standard DP analysis. We somewhat prefer to use the space for additional intuition and other details, but are happy to defer to the reviewers on this matter.

---

> > ### Comment · Reviewer_GhwK · 2023-08-14
> >
> > Thank you for your response and I have updated my score.

---

### Decision · Program_Chairs · 2023-09-21

**Decision:**

Accept (spotlight)

**Comment:**

This paper studies generalization in adaptive data analysis. It presents a novel analysis of Gaussian noise addition as a query answering mechanism. It shows that the noise can be scaled to the standard deviation of the queries, rather than their worst-case sensitivity, while achieving high-probability accuracy guarantees.

The main technical tool introduced is "pairwise concentration" which is a refined stability notion. Prior work has used either differential privacy or variants of mutual information.
Differential privacy gives high-probability guarantees, but the noise added must scale with the worst-case sensitivity.
Mutual information based approaches allow the noise to scale with the standard deviation of the queries, but they only yield in-expectation guarantees, rather than high-probability guarantees.
This paper provides the best of both worlds.

One limitation of the paper is that it assumes the queries' variance is known a priori. The prior works do not have this requirement. Thus there is room for further work.

Overall the reviewers are excited by this work and support acceptance at NeurIPS.